# Understanding Incremental Learning of Gradient Descent: A Fine-grained Analysis of Matrix Sensing

## Abstract

The implicit bias of optimization algorithms such as gradient descent (GD) is believed to play an important role in generalization of modern machine learning methods such as deep learning. This paper provides a fine-grained analysis of the dynamics of GD for the matrix sensing problem, whose goal is to recover a low-rank ground-truth matrix from near-isotropic linear measurements. With small initialization, we that GD behaves similarly to the greedy low-rank learning heuristics (Li et al., 2020) and follows an incremental learning procedure (Gissin et al., 2019). That is, GD sequentially learns solutions with increasing ranks until it recovers the ground-truth matrix. Compared to existing works which only analyze the first learning phase for rank-1 solutions, our result is stronger because it characterizes the whole learning process. Moreover, our analysis of the incremental learning procedure applies to the *under-parameterized* regime as well. As a key ingredient of our analysis, we observe that GD always follows an approximately low-rank trajectory and develops novel landscape properties for matrix sensing with low-rank parameterization. Finally, we conduct numerical experiments which confirm our theoretical findings.

## 1 Introduction

Understanding the optimization and generalization properties of optimization algorithms is one of the central topics in deep learning theory (Zhang et al., 2021; Sun, 2019). It has long been a mystery why simple algorithms such as Gradient Descent (GD) or Stochastic Gradient Descent (SGD) can find global minima even for highly non-convex functions (Du et al., 2019), and why the global minima being found can generalize well (Hardt et al., 2016).

One influential line of works provides theoretical analysis of the *implicit bias* of GD/SGD. These results typically exhibit theoretical settings where the low-loss solutions found by GD/SGD attain certain optimality conditions of a particular generalization metric, *e.g.*, the parameter norm (or the classifier margin) (Soudry et al., 2018; Gunasekar et al., 2018; Nacson et al., 2019; Lyu & Li, 2020; Ji & Telgarsky, 2020), the sharpness of local loss landscape (Blanc et al., 2020; Damian et al., 2021; Li et al., 2022; Lyu et al., 2022).

Among these works, a line of works seek to characterize the implicit bias even when the training is away from convergence. Kalimeris et al. (2019) empirically observed that SGD learns model from simple ones, such as linear classifiers, to more complex ones. This behavior, usually referred to as the *simplicity bias/incremental learning* of GD/SGD, can help prevent overfitting for highly over-parameterized models since it tries to fit the training data with minimal complexity. Hu et al. (2020); Lyu et al. (2021); Frei et al. (2021) theoretically establish that GD on two-layer nets learns linear classifiers first.

The goal of this paper is to demonstrate this simplicity bias/incremental learning in the *matrix sensing* problem, a non-convex optimization problem that arises in a wide range of real-world applications, *e.g.*, image reconstruction (Zhao et al., 2010; Peng et al., 2014), object detection (Shen & Wu, 2012; Zou et al., 2013) and array processing systems (Kalogerias & Petropulu, 2013). Moreover, this problem serves as a standard test-bed of the implicit bias of GD/SGD in deep learning theory, since it retains many of the key phenomena in deep learning while being simpler to analyze.

Formally, the matrix sensing problem asks for recovering a ground-truth matrix $\boldsymbol{Z}^* \in \mathbb{R}^{d \times d}$ given $m$ observations $y_1, \ldots, y_m$. Each observation $y_i$ here is resulted from a linear measurement $y_i = \langle \boldsymbol{A}_i, \boldsymbol{Z}^* \rangle$, where $\{\boldsymbol{A}_i\}_{1 \le i \le m}$ is a collection of symmetric measurement matrices. In this paper, we focus on the case where $\boldsymbol{Z}^*$ is positive semi-definite (PSD) and is of low-rank: $\boldsymbol{Z}^* \succeq \boldsymbol{0}$ and $\text{rank}(\boldsymbol{Z}^*) = r_* \ll d$.

An intriguing approach to solve this matrix sensing problem is to use the Burer-Monteiro type decomposition $\boldsymbol{Z}^* = \boldsymbol{U}\boldsymbol{U}^\top$ with $\boldsymbol{U} \in \mathbb{R}^{d \times \hat{r}}$, and minimize the squared loss with GD:

$$\min_{\boldsymbol{U} \in \mathbb{R}^{d \times \hat{r}}} \quad f(\boldsymbol{U}) := \frac{1}{4m} \sum_{i=1}^m \left( y_i - \langle \boldsymbol{A}_i, \boldsymbol{U}\boldsymbol{U}^\top \rangle \right)^2. \tag{1}$$

In the ideal case, the number of columns of $\boldsymbol{U}$, denoted as $\hat{r}$ above, should be set to $r_*$, but $r_*$ may not be known in advance. This leads to two training regimes that are more likely to happen: the under-parameterized regime where $\hat{r} < r_*$, and the over-parameterized regime where $\hat{r} > r_*$.

The over-parameterized regime may lead to overfitting at first glance, but surprisingly, with small initialization, GD induces a good implicit bias towards solutions with exact or approximate recovery of the ground truth. It was first conjectured in Gunasekar et al. (2017) that GD with small initialization finds the matrix with minimum nuclear norm. However, a series of works point out that this nuclear norm minimization view cannot capture the simplicity bias/incremental learning behavior of GD. In the matrix sensing setting, this term particularly refers to the phenomenon that GD tends learn solutions with rank gradually increasing with training steps. Arora et al. (2019) exhibits this phenomenon when there is only one observation ($m = 1$). Gissin et al. (2019); Jiang et al. (2022) study the full-observation case, where every entry of the ground truth is measured independently $f(\boldsymbol{U}) = \frac{1}{4d^2} \|\boldsymbol{Z}^* - \boldsymbol{U}\boldsymbol{U}^\top\|_\text{F}^2$, and GD is shown to sequentially recover singular components of the ground truth from the largest singular value to the smallest one. Li et al. (2020) provide theoretical evidence that the incremental learning behavior generally occurs for matrix sensing. They also give a concrete counterexample for Gunasekar et al. (2017)'s conjecture, where the simplicity bias drives GD to a rank-1 solution that has a large nuclear norm.

In spite of these progresses, theoretical understanding of the simplicity bias of GD remains limited. Indeed, a vast majority of existing analysis only shows that GD is initially biased towards learning a rank-1 solution, but their analysis cannot be generalized to higher ranks, unless additional assumptions on the GD dynamics are made (Li et al., 2020, Appendix H), (Belabbas, 2020; Jacot et al., 2021; Razin et al., 2021; 2022).

## 1.1 OUR CONTRIBUTIONS

In this paper, we take a step towards understanding the generalization of GD with small initialization by firmly demonstrating the simplicity bias/incremental learning behavior in the matrix sensing setting, assuming the Restricted Isometry Property (RIP). Our main result is informally stated below. See Theorem 4.1 for the formal version.

**Definition 1.1 (Best Rank-$s$ Solution)** *We define the best rank-$s$ solution as the unique global minimizer $\boldsymbol{Z}_s^*$ of the following constrained optimization problem:*

$$\min_{\boldsymbol{Z} \in \mathbb{R}^{d \times d}} \quad \frac{1}{4m} \sum_{i=1}^m (y_i - \langle \boldsymbol{A}_i, \boldsymbol{Z} \rangle)^2 \quad s.t. \quad \boldsymbol{Z} \succeq \boldsymbol{0}, \quad \text{rank}(\boldsymbol{Z}) \le s. \tag{2}$$

**Theorem 1.1 (Informal version of Theorem 4.1)** *Consider the matrix sensing problem (1) with rank-$r_*$ ground-truth matrix $\boldsymbol{Z}^*$ and measurements $\{\boldsymbol{A}_i\}_{i=1}^m$. Assume that the measurements satisfy the RIP condition (Definition 3.2). With small learning rate $\mu > 0$ and small initialization $\boldsymbol{U}_{\alpha,0} = \alpha \boldsymbol{U} \in \mathbb{R}^{d \times \hat{r}}$, the trajectory of $\boldsymbol{U}_{\alpha,t}\boldsymbol{U}_{\alpha,t}^\top$ during GD training enters an $o(1)$-neighbourhood of each of the best rank-$s$ solutions in the order of $s = 1, 2, \ldots, \hat{r} \wedge r_*$ when $\alpha \to 0$.*

It is shown in Li et al. (2018); Stöger & Soltanolkotabi (2021) that GD exactly recovers the ground truth under the RIP condition, but our theorem goes beyond this result in a number of ways. First, in the over-parameterized regime (i.e., $\hat{r} \ge r_*$), it implies that the trajectory of GD exhibits an *incremental learning* phenomenon: learning solutions with increasing ranks until it finds the ground

truth. Second, this result also shows that in the under-parameterized regime (i.e., $\hat{r} < r_*$), GD exhibits the same implicit bias, but finally it converges to the best low-rank solution of the matrix sensing loss. By contrast, to the best of our knowledge, only the over-parameterized setting is analyzed in existing literature.

Theorem 1.1 can also be considered as a generalization of previous results in Gissin et al. (2019); Jiang et al. (2022) which show that $\boldsymbol{U}_{\alpha,t}\boldsymbol{U}_{\alpha,t}^{\top}$ passes by the best low-rank solutions one by one in the full observation case of matrix sensing $f(\boldsymbol{U}) = \frac{1}{4d^2}\|\boldsymbol{Z}^* - \boldsymbol{U}\boldsymbol{U}^{\top}\|_{\mathrm{F}}^2$. However, our setting has two major challenges which significantly complicate our analysis. First, since our setting only gives partial measurements, the decomposition of signal and error terms in Gissin et al. (2019); Jiang et al. (2022) cannot be applied. Instead, we adopt a different approach which is motivated by Stöger & Soltanolkotabi (2021); intuitive explanations of our approach is discussed in Appendix B. Second, it is well-known that the optimal rank-$s$ solution of matrix factorization is $\boldsymbol{X}_s$ (defined in Section 3), but little is known for $\boldsymbol{Z}_s^*$. In Section 5 we analyze the landscape of (2), establishing the uniqueness of $\boldsymbol{Z}_s^*$ and local landscape properties under the RIP condition. We find that when $\boldsymbol{U}_{\alpha,t}\boldsymbol{U}_{\alpha,t}^{\top} \approx \boldsymbol{Z}_s^*$, GD follows an approximate low-rank trajectory, so that it behaves similarly to GD in the under-parameterized regime. Using our landscape results, we can finally prove Theorem 1.1.

**Organization.** We review additional related works in Section 2. In Section 3, we provide an overview of necessary background and notations. We then present our main results in Section 4 where we also give a proof sketch. In Section 5, we outline the key landscape results that we use to prove Theorem 4.1. Experimental results are presented in Section 6 which verify our theoretical findings. Finally, in Section 7, we summarize our main contributions and discuss some promising future directions. Complete proofs of all results in this paper are given in the Appendix.

## 2  RELATED WORK

**Low-rank matrix recovery.** The goal of low-rank matrix recovery is to recover an unknown low-rank matrix from a finite number of (possibly noisy) measurements. Examples include matrix sensing (Recht et al., 2010), matrix completion (Candès & Recht, 2009; Candes & Plan, 2010) and robust PCA (Xu et al., 2010; Candès et al., 2011). Fornasier et al. (2011); Ngo & Saad (2012); Wei et al. (2016); Tong et al. (2021) study efficient optimization algorithms with convergence guarantees. Interested readers can refer to Davenport & Romberg (2016) for an overview of low rank matrix recovery.

**Simplicity bias/incremental learning of gradient descent.** Besides the works mentioned in the introduction, other works study the simplicity bias/incremental learning of GD/SGD on tensor factorization (Razin et al., 2021; 2022), deep linear networks (Gidel et al., 2019), two-layer nets with orthogonal inputs (Boursier et al., 2022).

**Landscape analysis of non-convex low-rank problems.** The strict saddle property (Ge et al., 2016; 2015; Lee et al., 2016) was established for non-convex low-rank problems in a unified framework by Ge et al. (2017). Tu et al. (2016) proved a local PL property for matrix sensing with exact parameterization (i.e. the rank of parameterization and ground-truth matrix are the same). The optimization geometry of general objective function with Burer-Monteiro type factorization is studied in Zhu et al. (2018); Li et al. (2019); Zhu et al. (2021). We provide a comprehensive analysis in this regime for matrix factorization as well as matrix sensing that improves over their results.

## 3  PRELIMINARIES

In this section, we first list the notations used in this paper, and then provide details of our theoretical setup and necessary preliminary results.

### 3.1  NOTATIONS

We write $\min\{a,b\}$ as $a \wedge b$ for short. For any matrix $\boldsymbol{A}$, we use $\|\boldsymbol{A}\|_{\mathrm{F}}$ to denote the Frobenius norm of $\boldsymbol{A}$, use $\|\boldsymbol{A}\|$ to denote the spectral norm $\|\boldsymbol{A}\|_2$, and use $\sigma_{\min}(\boldsymbol{A})$ to denote the smallest singular value of $\boldsymbol{A}$. We use the following notation for Singular Value Decomposition (SVD):

**Definition 3.1 (Singular Value Decomposition)** *For any matrix $A \in \mathbb{R}^{d_1 \times d_2}$ of rank $r$, we use $A = V_A \Sigma_A W_A^\top$ to denote a Singular Value Decomposition (SVD) of $A$, where $V_A \in \mathbb{R}^{d_1 \times r}, W_A \in \mathbb{R}^{d_2 \times r}$ satisfy $V_A^\top V_A = I, W_A W_A^\top = I$, and $\Sigma_A \in \mathbb{R}^{r \times r}$ is diagonal.*

For the matrix sensing problem (1), we write the ground-truth matrix as $Z^* = XX^\top$ for some $X = [v_1, v_2, \cdots, v_{r_*}] \in \mathbb{R}^{d \times r_*}$ with orthogonal columns. We denote the singular values of $X$ as $\sigma_1, \sigma_2, \ldots, \sigma_{r_*}$, then the singular values of $Z^*$ are $\sigma_1^2, \sigma_2^2, \ldots, \sigma_{r_*}^2$. We set $\sigma_{r_*+1} := 0$ for convenience. For simplicity, we only consider the case where $Z^*$ has distinct singular values, i.e., $\sigma_1^2 > \sigma_2^2 > \cdots > \sigma_{r_*}^2 > 0$. We use $\kappa := \frac{\sigma_1^2}{\min_{1 \le s \le r_*} \{\sigma_s^2 - \sigma_{s+1}^2\}}$ to quantify the degeneracy of the singular values of $Z^*$. We also use the notatiowrite $X_s = [v_1, v_2, \cdots, v_s]$ for the matrix consisting of the first $s$ columns of $X$. Note that $Z_s^*$ (Definition 1.1) does not equal $X_s X_s^\top$ in general.

We write the results of the measurements $\{A_i\}_{i=1}^m$ as a linear mapping $\mathcal{A} : \mathbb{R}^{d \times d} \mapsto \mathbb{R}^m$, where $[\mathcal{A}(Z)]_i = \frac{1}{\sqrt{m}} \langle A_i, Z \rangle$ for all $1 \le i \le m$. We use $\mathcal{A}^* : \mathbb{R}^m \to \mathbb{R}^{d \times d}, \mathcal{A}^*(w) = \frac{1}{\sqrt{m}} \sum_{i=1}^m w_i A_i$ to denote the adjoint operator of $\mathcal{A}$. Our loss function (1) can then be written as $f(U) = \frac{1}{4} \left\| \mathcal{A} \left( Z^* - UU^\top \right) \right\|_2^2$. The gradient is given by $\nabla f(U) = \mathcal{A}^* \left( y - \mathcal{A}(UU^\top) \right) U = \mathcal{A}^* \mathcal{A} \left( XX^\top - UU^\top \right) U_t$.

In this paper, we consider GD with learning rate $\mu > 0$ starting from $U_0$. The update rule is

$$U_{t+1} = U_t - \mu \nabla f(U_t) =: (I + \mu M_t) U_t, \tag{3}$$

where $M_t = \mathcal{A}^* \mathcal{A} \left( XX^T - U_t U_t^T \right)$. We specifically focus on GD with small initialization: letting $U_0 = \alpha \bar{U}$ for some matrix $\bar{U} \in \mathbb{R}^{d \times r}$ with $\|\bar{U}\| = 1$, we are interested in the trajectory of GD when $\alpha \to 0$. Sometimes we write $U_t$ as $U_{\alpha,t}$ to highlight the dependence of the trajectory on $\alpha$.

## 3.2 ASSUMPTIONS

For our theoretical analysis of the matrix sensing problem, we make the following standard assumption in the matrix sensing literature:

**Definition 3.2 (Restricted Isometry Property)** *We say that a measurement operator $\mathcal{A}$ satisfies the $(\delta, r)$-RIP condition if $(1 - \delta)\|Z\|_F^2 \le \|\mathcal{A}(Z)\|_2^2 \le (1 + \delta)\|Z\|_F^2$ for all matrices $Z \in \mathbb{R}^{d \times d}$ with $\mathrm{rank}(Z) \le r$.*

**Assumption 3.1** *The measurement operator $\mathcal{A}$ satisfies the $(2r_* + 1, \delta)$-RIP property, where $r_* = \mathrm{rank}(Z^*)$ and $\delta \le 10^{-7} \kappa^{-4} r_*^{-1}$.*

The RIP condition is the key to ensure the ground truth to be recoverable with partial observations. An important consequence of RIP is that it guarantees $\mathcal{A}^* \mathcal{A}(Z) = \frac{1}{m} \sum_{i=1}^m \langle A_i, Z \rangle A_i \approx Z$ when $Z$ is low-rank. This is made rigorous in the following proposition.

**Proposition 3.1** *(Stöger & Soltanolkotabi, 2021, Lemma 7.3) Suppose that $\mathcal{A}$ satisfies $(r, \delta)$-RIP with $r \ge 2$, then for all symmetric $Z$,*

*(1). if $\mathrm{rank}(Z) \le r - 1$, we have $\|(\mathcal{A}^* \mathcal{A} - I)Z\|_2 \le \sqrt{r}\delta\|Z\|$.*

*(2). $\|(\mathcal{A}^* \mathcal{A} - I)Z\|_2 \le \delta\|Z\|_*$, where $\|\cdot\|_*$ is the nuclear norm.*

We also need the following regularity condition on the initialization.

**Assumption 3.2** *For all $1 \le s \le \hat{r} \wedge r_*$, $\sigma_{\min}\left(V_{X_s}^\top \bar{U}\right) \ge \rho$ for some positive constant $\rho$, where $V_{X_s}$ is defined as Definition 3.1.*

The following proposition implies that Assumption 3.2 is satisfied with high probability with a Gaussian initialization.

**Proposition 3.2** *Suppose that all entries of $U \in \mathbb{R}^{d \times \hat{r}}$ are independently drawn from $\mathcal{N}\left(0, \frac{1}{\sqrt{\hat{r}}}\right)$ and $\rho = \epsilon \frac{\sqrt{\hat{r}} - \sqrt{\hat{r} \wedge r_* - 1}}{\sqrt{\hat{r}}} \ge \frac{\epsilon}{2r_*}$, then $\sigma_{\min}\left(V_{X_s}^\top U\right) \ge \rho$ holds for all $1 \le s \le \hat{r} \wedge r_*$ with probability at least $1 - \hat{r}\left(C\epsilon + e^{-c\hat{r}}\right)$, where $c, C > 0$ are universal constants.*

## 4 MAIN RESULTS

In this section, we present our main theorem, following the theoretical setup in Section 3.

**Theorem 4.1** *Under Assumptions 3.1 and 3.2, consider GD* (3) *with learning rate* $\mu \leq \frac{1}{10^3 \|\boldsymbol{Z}^*\|}$
*and initialization* $\boldsymbol{U}_{\alpha,0} = \alpha \bar{\boldsymbol{U}}$ *for solving the matrix sensing problem* (1). *There exists universal
constants* $c, M > 0$, *a constant* $C$ *(depending on* $\hat{r}$ *and* $\kappa$*) and a sequence of time points* $T_\alpha^1 < T_\alpha^2 <$
$\cdots < T_\alpha^{\hat{r} \wedge r_*}$ *such that for all* $1 \leq s \leq \hat{r} \wedge r_*$, *the following holds when* $\alpha = \mathcal{O}\left((\rho r_*)^{-c\kappa}\right)$:

$$\left\| \boldsymbol{U}_{\alpha,T_\alpha^s} \boldsymbol{U}_{\alpha,T_\alpha^s}^\top - \boldsymbol{Z}_s^* \right\|_{\mathrm{F}} \leq C \alpha^{\frac{1}{M\kappa^2}}, \tag{4}$$

*where we recall that* $\boldsymbol{Z}_s^*$ *is the best rank-s solution defined in Definition 1.1. Moreover, GD follows
an incremental learning procedure: we have* $\lim_{\alpha \to 0} \max_{1 \leq t \leq T_\alpha^s} \sigma_{s+1}(\boldsymbol{U}_{\alpha,t}) = 0$ *for all* $1 \leq s \leq$
$\hat{r} \wedge r_*$, *where* $\sigma_i(\boldsymbol{A})$ *denotes the* $i$-*th largest singular value of a matrix* $\boldsymbol{A}$.

Compared with existing works (Li et al., 2018; Stöger & Soltanolkotabi, 2021) in the same setting,
our result characterizes the complete learning dynamics of GD and reveals an incremental learning mechanism, *i.e.*, GD starts from learning simple solutions and then gradually increasing the
complexity of search space until it finds the ground truth.

Now we outline the proof of our main theorem. In the following, we fix an integer $1 \leq s \leq \hat{r} \wedge r_*$
and show the existence of $T_\alpha^s > 0$ satisfying that $\lim_{\alpha \to 0} \left\| \boldsymbol{U}_{\alpha,T_\alpha^s} \boldsymbol{U}_{\alpha,T_\alpha^s}^\top - \boldsymbol{Z}_s^* \right\|_{\mathrm{F}} = 0$. We then
show that $T_\alpha^s$ is monotone increasing in $s$ for any fixed $\alpha$.

Our first result states that with small initialization, GD can get into a small neighbourhood of $\boldsymbol{Z}_s^*$.

**Lemma 4.1** *Under Assumptions 3.1 and 3.2, there exists* $\hat{T}_\alpha^s > 0$ *for all* $\alpha > 0$ *and* $1 \leq s \leq \hat{r} \wedge r_*$
*such that* $\lim_{\alpha \to 0} \max_{1 \leq t \leq \hat{T}_\alpha^s} \sigma_{s+1}(\boldsymbol{U}_{\alpha,t}) = 0$. *Furthermore, it holds that* $\left\| \boldsymbol{U}_{\hat{T}_\alpha^s} \boldsymbol{U}_{\hat{T}_\alpha^s}^\top - \boldsymbol{Z}_s^* \right\|_F =$
$\mathcal{O}\left(\kappa^3 \sqrt{r_*} \delta \|\boldsymbol{X}\|^2\right)$.

***Proof sketch***: The proof is motivated by the three-phase analysis in Stöger & Soltanolkotabi (2021)
but has some technical modifications. Starting from a small initialization, GD initially behaves similarly to power iteration since $\boldsymbol{U}_{t+1} = (\boldsymbol{I} + \mu \boldsymbol{M}_t)\boldsymbol{U}_t \approx (\boldsymbol{I} + \mu \boldsymbol{M})\boldsymbol{U}_t$, where $\boldsymbol{M} := \mathcal{A}^*\mathcal{A}(\boldsymbol{X}\boldsymbol{X}^\top)$
is a symmetric matrix. Let $\boldsymbol{M} = \sum_{k=1}^d \hat{\sigma}_k^2 \hat{\boldsymbol{v}}_k \hat{\boldsymbol{v}}_k^\top$ be the eigendecomposition of $\boldsymbol{M}$. Then we have

$$\boldsymbol{U}_T \approx (\boldsymbol{I} + \mu \boldsymbol{M})^T \boldsymbol{U}_0 = \sum_{i=1}^d (1 + \mu \hat{\sigma}_i^2)^T \hat{\boldsymbol{v}}_i \hat{\boldsymbol{v}}_i^\top \boldsymbol{U}_0 \approx \sum_{i=1}^s (1 + \mu \hat{\sigma}_i^2)^T \hat{\boldsymbol{v}}_i \hat{\boldsymbol{v}}_i^\top \boldsymbol{U}_0 \tag{5}$$

where the last step holds because it can be shown that $1 + \mu \hat{\sigma}_s > 1 + \mu \hat{\sigma}_{s+1}$, causing an exponential
separation between the magnitude of the top-$s$ and the remaining components.

When (5) no longer holds, we enter a new phase which we call the *parallel improvement phase*. We
consider the decomposition $\boldsymbol{U}_t = \boldsymbol{U}_t \boldsymbol{W}_t \boldsymbol{W}_t^\top + \boldsymbol{U}_t \boldsymbol{W}_{t,\perp} \boldsymbol{W}_{t,\perp}^\top$, where $\boldsymbol{W}_t := \boldsymbol{W}_{\boldsymbol{V}_{\boldsymbol{X}_s}^\top \boldsymbol{U}_t} \in \mathbb{R}^{\hat{r} \times s}$ is
the matrix consisting of the right singular vectors of $\boldsymbol{V}_{\boldsymbol{X}_s}^\top \boldsymbol{U}_t$ (Definition 3.1) and $\boldsymbol{W}_{t,\perp} \in \mathbb{R}^{\hat{r} \times (\hat{r}-s)}$
is an orthogonal complement of $\boldsymbol{W}_t$.

Assume $\boldsymbol{X}_s = [\sigma_1 \boldsymbol{e}_1, \cdots, \sigma_s \boldsymbol{e}_s]$ without loss of generality. The columns of $\boldsymbol{W}_t$ is an orthogonal
basis of the subspace spanned by the first $s$ rows of $\boldsymbol{U}_t$. Each vector in $\mathbb{R}^r$ can be decomposed as a
parallel component and an orthogonal component w.r.t. this subspace. Intuitively, the row vectors of
$\boldsymbol{U}_t \boldsymbol{W}_t$ are the parallel components of the row vectors in $\boldsymbol{U}_t$; we call $\boldsymbol{U}_t \boldsymbol{W}_t$ the *parallel component*.
For similar reasons, $\boldsymbol{U}_t \boldsymbol{W}_{t,\perp}$ will be referred to as the *orthogonal component*. More discussions for
this decomposition are given in Appendix B.

By the end of the spectral phase we have $\sigma_{\min}(\boldsymbol{U}_t \boldsymbol{W}_t) \gg \|\boldsymbol{U}_t \boldsymbol{W}_{t,\perp}\|$. We show in Appendix C.2
that afterwards, $\frac{\sigma_{\min}(\boldsymbol{U}_t \boldsymbol{W}_t)}{\|\boldsymbol{U}_t \boldsymbol{W}_{t,\perp}\|}$ grows exponentially in $t$, until the former reaches a constant scale, while
the latter stays $o(1)$ $(\alpha \to 0)$.

After $\sigma_{\min}(\boldsymbol{U}_t \boldsymbol{W}_t) = \Theta(1)$, we enter the *refinement phase* where we show in Appendix C.4
that $\left\| \boldsymbol{X}_s \boldsymbol{X}_s^\top - \boldsymbol{U}_t \boldsymbol{U}_t^\top \right\|_{\mathrm{F}}$ keeps decreasing until it is $\mathcal{O}\left(\delta \kappa^2 \sqrt{r_*} \|\boldsymbol{X}\|^2\right)$ (see Lemma 5.1). On

the other hand, we can show that best rank-$s$ solution is close to the matrix factorization minimizer *i.e.* $\left\|\boldsymbol{Z}_s^* - \boldsymbol{X}_s\boldsymbol{X}_s^\top\right\|_F = \mathcal{O}\left(\delta\kappa\sqrt{r_*}\|\boldsymbol{X}\|^2\right)$. We thus obtain that $\left\|\boldsymbol{Z}_s^* - \boldsymbol{U}_t\boldsymbol{U}_t^\top\right\|_F = \mathcal{O}\left(\delta\kappa^2\sqrt{r_*}\|\boldsymbol{X}\|^2\right)$. Finally, since $\operatorname{rank}(\boldsymbol{U}_t\boldsymbol{W}_t) \leq s$, we have $\sigma_{s+1}(\boldsymbol{U}_t) \leq \|\boldsymbol{U}_t\boldsymbol{W}_{t,\perp}\| = o(1)$, as desired. □

Lemma 4.1 shows that $\boldsymbol{U}_t\boldsymbol{U}_t^\top$ would enter a neighbourhood of $\boldsymbol{Z}_s^*$ with *constant* radius. However, there is still a gap between Lemma 4.1 and Theorem 4.1, since the latter states that $\boldsymbol{U}_t\boldsymbol{U}_t^\top$ would actually get $o(1)$-close to $\boldsymbol{Z}_s^*$. To illustrate our proof idea of this result, we first consider the simpler setting where the model is *under-parameterized*, *i.e.*, $\hat{r} \leq r^*$ and show that $\boldsymbol{U}_t\boldsymbol{U}_t^\top$ would eventually converge to $\boldsymbol{Z}_{\hat{r}}^*$.

**Proposition 4.1 (Convergence in the under-parameterized regime)** *Suppose that $\hat{r} \leq r^*$, then there exists a constant $c = c(\hat{r}, \kappa) > 0$ such that when $\alpha < c$, we have $\lim_{t\to+\infty} \boldsymbol{U}_{\alpha,t}\boldsymbol{U}_{\alpha,t}^\top = \boldsymbol{Z}_{\hat{r}}^*$.*

***Proof sketch***: We can deduce from Lemma 4.1 by taking $s = \hat{r}$ that there exists a global minimizer $\boldsymbol{U}_{\hat{r}}^*$ of (1) (equivalently, a matrix in $\mathbb{R}^{d\times s}$ satisfying $\boldsymbol{U}_{\hat{r}}^*\boldsymbol{U}_{\hat{r}}^{*\top} = \boldsymbol{Z}^*$), such that $\left\|\boldsymbol{U}_{\alpha,\hat{T}_\alpha^{\hat{r}}} - \boldsymbol{U}_{\hat{r}}^*\right\| = \mathcal{O}(\kappa^3\sqrt{r_*}\delta\|\boldsymbol{X}\|)$ (cf. Corollary F.1). On the other hand, by taking $s = \hat{r}$ in Theorem 5.1, we can see that within a neighbourhood of $\boldsymbol{U}_{\hat{r}}^*$ with *constant* radius, $f$ satisfies a Polyak-Łojasiewicz (PL) type condition with respect to the procrutes distance defined in Definition 5.1, and the global minimizer of $f$ is unique up to orthogonal transformation. When $\delta$ is sufficiently small, $\boldsymbol{U}_{\alpha,T_\alpha^{\hat{r}}}$ lies in this neighbourhood, and the PL condition implies that GD converges linearly to the set of global minimizers, which yields the desired conclusion. □

Now we turn to the *over-parameterized* regime, where $f$ is not necessarily local PL, and thus we cannot directly derive convergence as in Proposition 4.1. To proceed, we use a low-rank approximation for $\boldsymbol{U}_t$ and associate the dynamics in this neighborhood with the GD dynamics of the following under-parameterized matrix sensing loss:

$$f_s(\boldsymbol{U}) = \frac{1}{4}\left\|\mathcal{A}(\boldsymbol{Z}^* - \boldsymbol{U}\boldsymbol{U}^\top)\right\|_2^2, \quad \boldsymbol{U} \in \mathbb{R}^{d\times s}, \tag{6}$$

It can be shown that when $\delta$ is sufficiently small, the global minimizer of $f_s$ is unique up to rotation, i.e., if $\hat{\boldsymbol{U}}_s^*$ is a global minimizer of $f_s$, then any other global minimizer can be written as $\hat{\boldsymbol{U}}_s^*\boldsymbol{R}$ for some orthogonal matrix $\boldsymbol{R} \in \mathbb{R}^{s\times s}$. The main observation is that GD follows an approximate low-rank trajectory until it gets into a small neighbourhood of $\boldsymbol{Z}_s^*$, so that we can still use our landscape results in the low-rank regime.

***Proof sketch of Theorem 4.1:*** For all $t \geq -\hat{T}_\alpha^s$, define $\hat{\boldsymbol{U}}_{\alpha,t} = \boldsymbol{U}_{\alpha,t+\hat{T}_\alpha^s}\boldsymbol{W}_{\alpha,t+\hat{T}_\alpha^s} \in \mathbb{R}^{d\times s}$, where $\boldsymbol{W}_{\alpha,t} = \boldsymbol{W}_{\boldsymbol{V}_{\boldsymbol{X}_s}^\top\boldsymbol{U}_t}$ as defined in the proof sketch of Lemma 4.1. Note that Lemma 4.1 implies that $\boldsymbol{U}_{\alpha,T_\alpha^s}$ is approximately rank-$s$, so within a time period after $\hat{T}_\alpha^s$, the GD trajectory remains close to another GD initialized at $\hat{\boldsymbol{U}}_{\alpha,0}$ for the rank-$s$ matrix sensing loss (6) until $t = \Theta\left(\log\frac{1}{\alpha}\right)$, *i.e.*

$$\hat{\boldsymbol{U}}_{\alpha,t}\hat{\boldsymbol{U}}_{\alpha,t}^\top \approx \boldsymbol{U}_{\alpha,\hat{T}_\alpha^s+t}\boldsymbol{U}_{\alpha,\hat{T}_\alpha^s+t}^\top. \tag{7}$$

Again by Lemma 4.1, the initialization $\hat{\boldsymbol{U}}_{\alpha,0}$ is within a small neighbourhood of the global minima of $f_s(\boldsymbol{U})$. Furthermore, Theorem 5.1 implies that $f_s(\boldsymbol{U})$ satisfies a local PL-like condition, so that GD with good initialization would converge linearly to its global minima (Karimi et al., 2016). We need to choose a time $t$ such that (7) remains true while this linear convergence takes place for sufficiently many steps. We can show that there always exists some $t = t_\alpha^s$ such that both $\left\|\hat{\boldsymbol{U}}_{\alpha,t}\hat{\boldsymbol{U}}_{\alpha,t}^\top - \boldsymbol{U}_{\alpha,\hat{T}_\alpha^s+t}\boldsymbol{U}_{\alpha,\hat{T}_\alpha^s+t}^\top\right\|_F$ and $\left\|\hat{\boldsymbol{U}}_{\alpha,t} - \boldsymbol{U}_s^*\right\|_F$ are $\mathcal{O}\left(\alpha^{\frac{1}{M\kappa^2}}\right)$. Hence $\left\|\boldsymbol{U}_{\alpha,t}\boldsymbol{U}_{\alpha,t}^\top - \boldsymbol{Z}_s^*\right\|_F \lesssim \alpha^{\frac{1}{M\kappa^2}}$ for $t = T_\alpha^s := \hat{T}_\alpha^s + t_\alpha^s$.

For all $1 \leq s < \hat{r} \wedge r_*$, since (7) always holds for $t \leq t_\alpha^s$ and $\operatorname{rank}\left(\hat{\boldsymbol{U}}_{\alpha,t}\right) \leq s$, we must have $\max_{1\leq t\leq T_\alpha^s} \sigma_{s+1}(\boldsymbol{U}_{\alpha,t}) = o(1)$ as $\alpha \to 0$. Finally, $\left\|\boldsymbol{Z}_{s+1}^* - \boldsymbol{X}_{s+1}\boldsymbol{X}_{s+1}^\top\right\| = \mathcal{O}(\delta)$ (cf. Lemma E.5), so $\sigma_{s+1}(\boldsymbol{Z}_{s+1}^*) = \Theta(1)$. Therefore, $\boldsymbol{U}_{\alpha,t}\boldsymbol{U}_{\alpha,t}^\top$ cannot be close to $\boldsymbol{Z}_{s+1}^*$ when $t \leq T_\alpha^s$, so we must have $T_\alpha^{s+1} > T_\alpha^s$. This completes the proof of Theorem 4.1.

## 5 Convergence in Under-Parameterized Matrix Sensing

In this section, we analyze the properties of the landscape of the matrix sensing loss (1), which plays a crucial role in proving results in Section 4. While the landscape of over-parameterized matrix sensing (i.e. $\hat{r} \geq r^*$) is well-studied (Tu et al., 2016; Ge et al., 2017), few results are known for the under-parameterized case. Our results provide useful tools for analyzing the convergence of gradient-based algorithms for solving problems like low-rank matrix approximation and might have independent interests.

To prove convergence results like Proposition 4.1, we first study the landscape of $f_s$ defined in (6) near the set of global minimizers. One major difficulty here is that $f_s$ is not locally strongly-convex, because if $\boldsymbol{U} \in \mathbb{R}^{d \times s}$ is a global minimizer of $f_s$, then $\boldsymbol{U R}$ is also a global minimizer, for any orthogonal matrix $\boldsymbol{R} \in \mathbb{R}^{s \times s}$. Nonetheless, we can establish a Polyak-Łojasiewicz (PL) like condition, Theorem 5.1, which is the main result of this section.

**Definition 5.1 (Procrutes distance)** *For any $d, s \in \mathbb{N}^+$ and $\boldsymbol{U}_1, \boldsymbol{U}_2 \in \mathbb{R}^{d \times s}$, we define* $\text{dist}(\boldsymbol{U}_1, \boldsymbol{U}_2) = \min \{ \|\boldsymbol{U}_1 - \boldsymbol{U}_2 \boldsymbol{R}\|_F : \boldsymbol{R} \in \mathbb{R}^{s \times s} \text{ is orthogonal} \}$.

We note that the procrutes distance is well-defined because the $s \times s$ orthogonal matrices are a compact set and thus $\|\boldsymbol{U}_1 - \boldsymbol{U}_2 \boldsymbol{R}\|_F$ is continuous in $\boldsymbol{R}$. It can be verified that procrutes distance is a pseudo metric, *i.e.*, it is symmetric and satisfies the triangle inequality.

**Theorem 5.1 (Landscape of under-parameterized matrix sensing)** *The global minimizer of $f_s$ is unique up to an orthogonal transformation,* i.e. *the set of global minimizers of $f_s$ is $\{\boldsymbol{U}_s^* \boldsymbol{R} : \boldsymbol{R} \in \mathbb{R}^{s \times s} \text{ is orthogonal}\}$ where $\boldsymbol{U}_s^*$ is an arbitrary global minimizer. Moreover, let $\boldsymbol{U} \in \mathbb{R}^{d \times s}$ and $\boldsymbol{U}_s^*$ be a global minimizer of $f_s$ such that $\|\boldsymbol{U} - \boldsymbol{U}_s^*\|_F = \text{dist}(\boldsymbol{U}, \boldsymbol{U}_s^*)$. Suppose that Assumption 3.1 holds and $\|\boldsymbol{U} - \boldsymbol{U}_s^*\| \leq 10^{-2} \kappa^{-1} \|\boldsymbol{X}\|$, then $\langle \nabla f_s(\boldsymbol{U}), \boldsymbol{U} - \boldsymbol{U}_s^* \rangle \geq 0.1\tau \text{dist}^2(\boldsymbol{U}, \boldsymbol{U}_s^*)$.*

**Remark 5.1** *Recall PL condition means there exists some constant $\mu > 0$, such that $\|\nabla g(x)\|^2 \geq 2\mu (g(x) - \min_{y \in \mathbb{R}^n} g(y))$ holds for all $x$. Since $f_s$ is locally smooth around $\boldsymbol{U}_s^*$, there exists a constant $c_1 > 0$ such that $f_s(\boldsymbol{U}) - f_s(\boldsymbol{U}_s^*) \leq c_1 \|\boldsymbol{U} - \boldsymbol{U}_s^*\|_F^2$. Moreover, Theorem 5.1 implies that $\|\boldsymbol{U} - \boldsymbol{U}_s^*\|_F^2 \leq 100\tau^{-2} \|\nabla f_s(\boldsymbol{U})\|_F^2$, so we have $\|\nabla f_s(\boldsymbol{U})\|_F^2 \geq 10^{-2}\tau^2 c_1^{-1} (f_s(\boldsymbol{U}) - f_s(\boldsymbol{U}_s^*))$. In other words, Theorem 5.1 implies that the matrix sensing loss (1) is locally PL.*

When $\delta = 0$, $f_s$ reduces to the matrix factorization loss $F_s : \mathbb{R}^{d \times s} \to \mathbb{R}, F_s(\boldsymbol{U}) = \frac{1}{4d^2} \|\boldsymbol{U U}^\top - \boldsymbol{Z}^*\|_F^2$. The following corollary immediately follows from Theorem 5.1.

**Corollary 5.1 (Landscape of under-parameterized matrix factorization)** *The set of global minimizers of $F_s$ is $\{\boldsymbol{X}_s \boldsymbol{R} : \boldsymbol{R} \in \mathbb{R}^{s \times s} \text{ is orthogonal}\}$. Moreover, under Assumption 3.1, given $\boldsymbol{U} \in \mathbb{R}^{d \times s}$ and let $\boldsymbol{R}$ be an orthogonal matrix such that $\|\boldsymbol{U} - \boldsymbol{X}_s \boldsymbol{R}\|_F = \text{dist}(\boldsymbol{U}, \boldsymbol{X}_s)$. If $\text{dist}(\boldsymbol{U}, \boldsymbol{X}_s) \leq 10^{-2} \kappa^{-1} \|\boldsymbol{X}\|$, then $\langle \nabla F(\boldsymbol{U}), \boldsymbol{U} - \boldsymbol{X}_s \boldsymbol{R} \rangle \geq 0.1\tau \text{dist}^2(\boldsymbol{U}, \boldsymbol{X}_s)$.*

We end this section with the following lemma that formalizes the intuition that all global minimizers of the $f_s$ must be close to $\boldsymbol{X}_s$ under the procrutes distance, which is used in the proof sketch of Theorem 4.1 in Section 4.

**Lemma 5.1** *Under Assumption 3.1, we have $\text{dist}(\boldsymbol{U}_s^*, \boldsymbol{X}_s) \leq 40\delta\kappa\|X\|_F$ for any global minimizer $\boldsymbol{U}_s^*$ of $f_s$. Moreover, $\|\boldsymbol{Z}_s^* - \boldsymbol{X}_s \boldsymbol{X}_s^\top\|_F \leq 80\delta\kappa\sqrt{r_*}\|\boldsymbol{X}\|^2$.*

## 6 Experiments

In this section, we perform some numerical experiments to illustrate our theoretical findings.

**Experimental setup.** Consistent with our theory, we consider the matrix sensing problem (1) with $d = 50, r_* = 5, \alpha \in \{1, 0.1, 0.01, 0.001\}, m \in \{1000, 2000, 5000\}$. We will consider different choices for $\hat{r}$ in the experiments. The ground-truth $\boldsymbol{Z}^* = \boldsymbol{X X}^\top$ is generated such that the entries

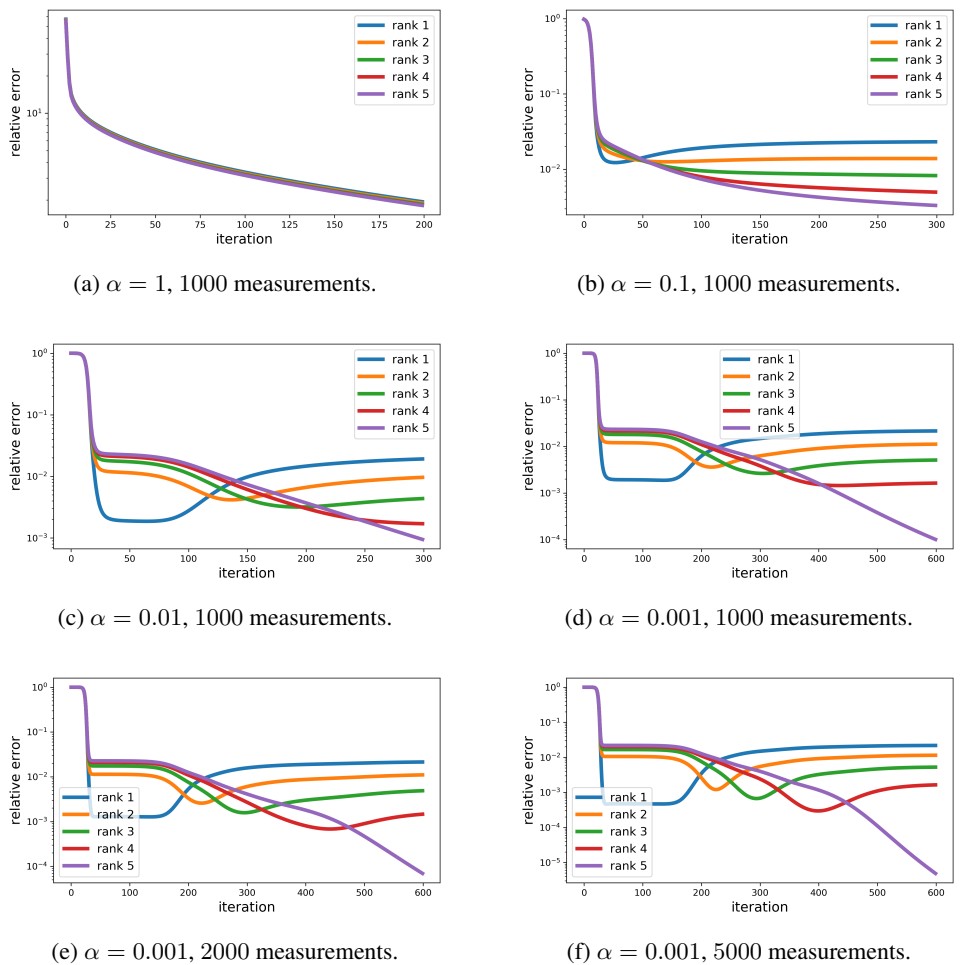

Figure 1: The evolution of relative error against the best solution of different ranks over time.

of $\boldsymbol{X}$ are i.i.d. standard Gaussian variables. Throughout our experiments, we use the same ground-truth to compare different choices of parameters. For $i = 1, 2, \cdots, m$, all entries of the measurement $\boldsymbol{A}_i \in \mathbb{R}^{d \times d}$ are chosen i.i.d. from the standard Gaussian $\mathcal{N}(0, 1)$. When $m \gtrsim dr_* \delta^{-2}$, this set of measurements satisfies the RIP with high probability (Recht et al., 2010, Theorem 4.2).

We solve the problem (1) via running GD for $T = 10^4$ iterations starting with small initialization with scale $\alpha$. Specifically, we choose $\boldsymbol{U}_0 = \alpha \boldsymbol{U}$ where the entries of $\boldsymbol{U} \in \mathbb{R}^{d \times \hat{r}}$ are drawn i.i.d. from standard Gaussian distribution. We consider both the over-parameterized and the exact/under-parameterized regime. The learning rate of GD is set to be $\mu = 0.005$.

### 6.1 IMPLICIT LOW-RANK BIAS

In this subsection, we consider the over-parameterized setting with $r = 50$. For each iteration $t \in [T]$ and rank $s \in [r_*]$, we define the relative error $\mathcal{E}_s(t) = \frac{\left\| \boldsymbol{U}_t \boldsymbol{U}_t^\top - \boldsymbol{X}_s \boldsymbol{X}_s^\top \right\|_F^2}{\left\| \boldsymbol{X}_s \boldsymbol{X}_s^\top \right\|_F^2}$ to measure the proximity of the GD iterates to $\boldsymbol{X}_s$. We plot the relative error in Figure 1 for different choices of $\alpha$ and $m$ (which affects the measurement error $\delta$).

**Small initialization.** The implicit low-rank bias of GD is evident when the initialization scale $\alpha$ is small. Indeed, one can observe that GD first visits a small neighbourhood of $\boldsymbol{X}_1$, spends a long period of time near it, and then moves towards $\boldsymbol{X}_2$. It then proceeds to learn $\boldsymbol{X}_3, \boldsymbol{X}_4, \cdots$ in a similar way, until it finally fits the ground truth. This is in align with Theorem 4.1. By contrast, for large initialization we do not have this implicit bias.

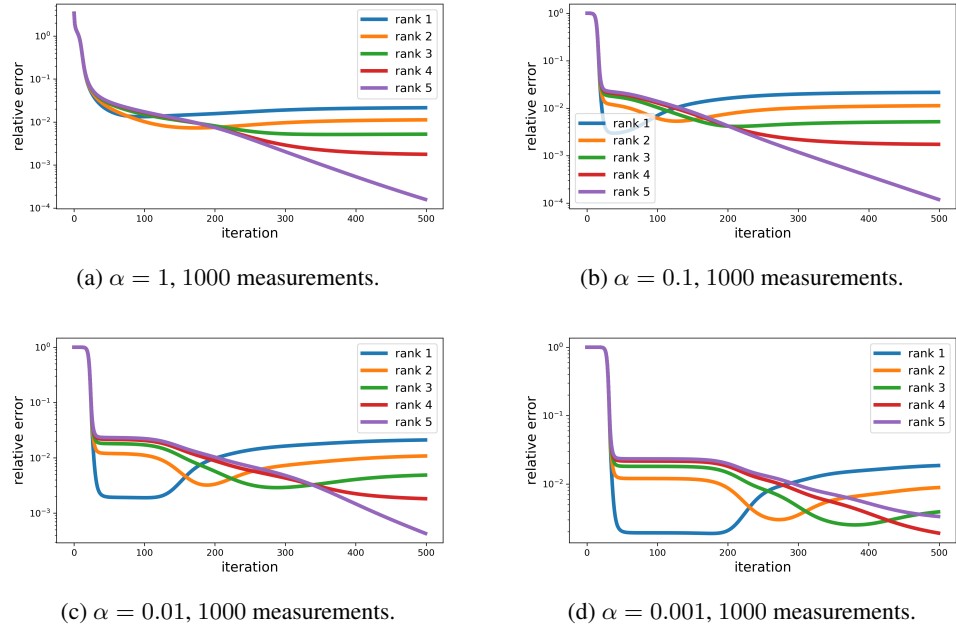

(a) $\alpha = 1$, 1000 measurements.

(b) $\alpha = 0.1$, 1000 measurements.

(c) $\alpha = 0.01$, 1000 measurements.

(d) $\alpha = 0.001$, 1000 measurements.

Figure 2: The evolution of the loss and relative error against best solution of different ranks in the exact-parameterized case $r = 5$.

**The effect of measurement error.** For fixed $\alpha$, one can observe the the relative error becomes smaller when the number of measurement increases. This is in align with Lemma 4.1 in which the bound depends on $\delta$. In particular, for the case $s = r_*$, although GD with fixed initialization does not converge to global minima, but the distance to the set of global minima scales as $\text{poly}(\alpha)$.

### 6.2 MATRIX SENSING WITH EXACT PARAMETERIZATION

Now we study the behavior of GD in the exact parameterization regime ($r = r_*$). We fix $m = 1000$ and $r = r_* = 5$ and run GD for $T = 500$ iterations. We plot the relative error in Figure 2. We can see that GD exhibits an implicit low-rank bias when $\alpha$ is small. However, choosing a very small $\alpha$ would slow down the convergence speed. This is because GD would get into a $\text{poly}(\alpha)$-neighbourhood of the saddle point $\boldsymbol{Z}_s$ and spend a long time escaping the saddle. Also, convergence to global minimizers is guaranteed as long as $\alpha$ is below a certain threshold (see Proposition 4.1).

## 7 CONCLUSION

In this paper, we study the matrix sensing problem with RIP measurements and show that GD with small initialization follows an incremental learning procedure, where GD finds near-optimal solutions with increasing ranks until it finds the ground-truth. We take a step towards understanding the optimization and generalization aspects of simple optimization methods, thereby providing insights into their success in modern applications such as deep learning (Goodfellow et al., 2016). Also, we provide a detailed landscape analysis in the under-parameterized regime, which to the best of our knowledge is the first analysis of this kind.

Although we focus on matrix sensing in this paper, it has been revealed in a line of works that the implicit regularization effect may vary for different models, including deep matrix factorization (Arora et al., 2019) and nonlinear ReLU/LeakyReLU networks (Lyu et al., 2021; Timor et al., 2022). Also, it is shown in Woodworth et al. (2020) that different initialization scales can lead to distinct inductive bias and affect the generalization and optimization behaviors. All these results indicate that we need further studies to comprehensively understand gradient-based optimization methods from the generalization aspect.

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

# Appendix

## Table of Contents

The appendix is organized as follows: in Appendix A we present a number of results that will be used for later proof. Appendix B sketches the main idea for proving our main results. Appendix C is devoted to a rigorous proof of Lemma B.1 ,with some auxiliary lemmas proved in Appendix D. In Appendix E we analyze the landscape of low-rank matrix sensing and prove our results in Section 5. These results are then used in Appendix F to prove Theorem 4.1. Finally, Appendix G studies the landscape of rank-1 matrix sensing, which enjoys a strongly convex property, as we mentioned in Section 5 without proof.

## A    PRELIMINARIES

In this section, we present some useful results that is needed in subsequent analysis.

### A.1    THE RIP CONDITION AND ITS PROPERTIES

In this subsection, we collect a few useful properties of the RIP condition, which we recall below:

**Definition A.1** *We say that the measurement $\mathcal{A}$ satisfies the $(\delta, r)$-RIP condition if for all matrices $\boldsymbol{Z} \in \mathbb{R}^{d \times d}$ with $\mathrm{rank}\,(\boldsymbol{Z}) \leq r$, we have*

$$(1 - \delta)\|\boldsymbol{Z}\|_F^2 \leq \|\mathcal{A}(\boldsymbol{Z})\|_2^2 \leq (1 - \delta)\|\boldsymbol{Z}\|_F^2.$$

The key intuition behind RIP is that $\mathcal{A}^*\mathcal{A} \approx I$, where $\mathcal{A}^* : \boldsymbol{v} \mapsto \frac{1}{\sqrt{m}} \sum_{i=1}^m v_i \boldsymbol{A}_i$ is the adjoint of $\mathcal{A}$. This intuition is made rigorous by the following proposition:

**Proposition A.1** *(Stöger & Soltanolkotabi, 2021, Lemma 7.3) Suppose that $\mathcal{A}$ satisfies $(r, \delta)$-RIP, then for all symmetric matrix $Z$ of rank $\leq r - 1$, we have*

$$\|(\mathcal{A}^*\mathcal{A} - \boldsymbol{I})\boldsymbol{Z}\| \leq \sqrt{r}\delta\|\boldsymbol{Z}\|.$$

### A.2    MATRIX ANALYSIS

The following lemma is a direct corollary of Proposition A.1 and will be frequently used in our proof.

**Lemma A.1** *Suppose that the measurement $\mathcal{A}$ satisfies $(\delta, 2r_* + 1)$-RIP condition, then for all matrices $\boldsymbol{U} \in \mathbb{R}^{d \times r}$ such that $\mathrm{rank}\,(\boldsymbol{U}) \leq r_*$, we have*

$$\left\|(\mathcal{A}^*\mathcal{A} - \boldsymbol{I})\left(\boldsymbol{X}\boldsymbol{X}^\top - \boldsymbol{U}\boldsymbol{U}^\top\right)\right\| \leq \delta\sqrt{r_*}\left(\|\boldsymbol{X}\|^2 + \|\boldsymbol{U}\|^2\right).$$

In our proof we will frequently make use of the Weyl's inequality for singular values:

**Lemma A.2 (Weyl's inequality)** *Let $\boldsymbol{A}, \Delta \in \mathbb{R}^{d \times d}$ be two matrices, then for all $1 \leq k \leq d$, we have*

$$|\sigma_k(\boldsymbol{A}) - \sigma_k(\boldsymbol{A} + \Delta)| \leq \|\Delta\|.$$

We will also need the Wedin's sin theorem for singular value decomposition:

**Lemma A.3** *(Wedin, 1972, Section 3) Define $R(\cdot)$ to be the column space of a matrix. Suppose that matrices $\boldsymbol{B} = \boldsymbol{A} + \boldsymbol{T}$, $\boldsymbol{A}_1$, $\boldsymbol{B}_1$ are the top-$s$ components in the SVD of $\boldsymbol{A}$ and $\boldsymbol{B}$ respectively, and $\boldsymbol{A}_0 = \boldsymbol{A} - \boldsymbol{A}_1, \boldsymbol{B}_0 = \boldsymbol{B} - \boldsymbol{B}_1$. If $\delta = \sigma_{\min}(\boldsymbol{B}_1) - \sigma_{\max}(\boldsymbol{A}_0) > 0$, then we have*

$$\|\sin\Theta\left(R(\boldsymbol{A}_1), R(\boldsymbol{B}_1)\right)\| \leq \frac{\|T\|}{\delta}$$

*where $\Theta(\cdot, \cdot)$ denotes the angle between two subspaces.*

Equipped with Lemma A.1, we can have the following characterization of the eigenvalues of $M$ (recall that $M = \mathcal{A}^*\mathcal{A}(\boldsymbol{X}\boldsymbol{X}^\top)$):

**Lemma A.4** *Let* $M := \mathcal{A}^*\mathcal{A}(XX^\top)$ *and* $M = \sum_{k=1}^d \hat{\sigma}_k^2 \hat{v}_k \hat{v}_k^\top$ *be the eigen-decomposition of* $M$. *For* $1 \le i \le d$ *we have*

$$\left| \sigma_i^2 - \hat{\sigma}_i^2 \right| \le \delta \|X\|^2.$$

***Proof*** **:** By Weyl's inequality we have

$$\left| \sigma_i^2 - \hat{\sigma}_i^2 \right| \le \left\| M - XX^\top \right\| \le \delta \|X\|^2$$

as desired. □

### A.3 OPTIMIZATION

**Lemma A.5** *Suppose that a smooth function* $f \in \mathbb{R}^m \mapsto \mathbb{R}$ *with minimum value* $f^* > -\infty$ *satisfies the following conditions with some* $\epsilon > 0$:

*(1).* $\lim_{\|x\| \to +\infty} f(x) = +\infty$.

*(2).* *There exists an open subset* $S \subset \mathbb{R}^m$ *such that the set* $S^*$ *of global minima of* $f$ *is contained in* $S$, *and for all stationary points* $x$ *of* $f$ *in* $\mathbb{R}^m - S$, *we have* $f(x) - f^* \ge 2\epsilon$. *Moreover, we also have* $f(x) - f^* \ge 2\epsilon$ *on* $\partial S$.

*Then we have*

$$\{x \in \mathbb{R}^m : f(x) - f^* \le \epsilon\} \subset S.$$

***Proof*** **:** Let $x^*$ be the minimizer of $f$ on $\mathbb{R}^m - S$. By condition (1) we can deduce that $x^*$ always exists. Moreover, since any local minimizer of a function defined on a compact set must either be a stationary point or lie on the boundary of its domain, we can see that either $x^* \in \partial S$ or $\nabla f(x^*) = 0$ holds. By condition (2), either cases would imply that $f(x^*) - f^* \ge 2\epsilon$, as desired. □

**Lemma A.6** *Let* $\{x_k\}, \{y_k\} \subset \mathbb{R}^n$ *be two sequences generated by* $x_{k+1} = x_k - \mu \nabla f(x_k)$ *and* $y_{k+1} = y_k - \mu \nabla f(y_k)$. *Suppose that* $\|x_k\| \le B$ *and* $\|y_k\| \le B$ *for all* $k$ *and* $f$ *is* $L$-*smooth in* $\{x \in \mathbb{R}^n : \|x\| \le B\}$, *then we have*

$$\|x_k - y_k\| \le (1 + \mu L)^k \|x_0 - y_0\|.$$

***Proof*** **:** The update rule implies that

$$\begin{aligned} \|x_{k+1} - y_{k+1}\| &= \|x_k - y_k - \mu \nabla f(x_k) + \mu \nabla f(y_k)\| \\ &\le \|x_k - y_k\| + \mu \|\nabla f(x_k) - f(y_k)\| \\ &\le (1 + \mu L) \|x_k - y_k\| \end{aligned}$$

which yields the desired inequality. □

### A.4 PROOF OF PROPOSITION 3.2

Proposition 3.2 immediately follows from the following result:

**Proposition A.2** *(Rudelson & Vershynin, 2009) Suppose that all entries of* $U \in \mathbb{R}^{d \times r}$ *are independently drawn from* $\mathcal{N}\left(0, \frac{1}{\sqrt{r}}\right)$ *and* $\rho = \epsilon \frac{\sqrt{r} - \sqrt{s-1}}{\sqrt{r}}$, *then* $\sigma_{\min}\left(V_{X_s}^\top U\right) \ge \rho$ *with probability at least* $1 - e^{-cr} - (C\epsilon)^{r-s+1}$. *Here* $c, C > 0$ *are universal constants.*

By Proposition A.2, we have

$$\mathbb{P}\left[ \exists 1 \le s \le r \wedge r_* \text{ s.t. } \sigma_{\min}\left(V_{X_s}^\top U\right) < \frac{2\epsilon}{r} \right]$$

$$\le \sum_{s=1}^{r \wedge r_*} \mathbb{P}\left[ \sigma_{\min}\left(V_{X_s}^\top U\right) < \epsilon \frac{\sqrt{r} - \sqrt{s-1}}{\sqrt{r}} \right]$$

$$\le \sum_{s=1}^{r \wedge r_*} \left( e^{-cr} + (C\epsilon)^{r-s+1} \right) \le r \left( e^{-cr} + C\epsilon \right)$$

which concludes the proof of Proposition 3.2.

# B  MAIN IDEA FOR THE PROOF OF THEOREM 4.1

In this section, we briefly introduce our main ideas for proving Theorem 4.1. Motivated by Stöger & Soltanolkotabi (2021), we decompose the matrix $U_t$ into a parallel component and an orthogonal component. Specifically, we write

$$U_t = \underbrace{U_t W_t W_t^\top}_{\text{parallel component}} + \underbrace{U_t W_{t,\perp} W_{t,\perp}^\top}_{\text{orthogonal component}}, \tag{8}$$

where $W_t := W_{V_{X_s}^\top U_t} \in \mathbb{R}^{\hat{r} \times s}$ is the matrix consisting of the right singular vectors of $V_{X_s}^\top U_t$ (Definition 3.1) and $W_{t,\perp} \in \mathbb{R}^{\hat{r} \times (\hat{r}-s)}$ is an orthogonal complement of $W_t$. Our goal is to prove that at some time $t$, we have $V_{X_s}^\top (U_t U_t^\top - X_s X_s^\top) \approx 0$ and $\|U_t W_{t,\perp}\| \approx 0$. As we will see later, these imply that $\|U_t U_t^\top - X_s X_s^\top\| \approx 0$. The remaining part of this section is organized as follows: in Appendix B.1 we give a heuristic explanation for considering (8), and in Appendix B.2, we present our proof outline.

**Additional Notations.**  Let $V_{X_s,\perp} \in \mathbb{R}^{d \times (d-s)}$ be an orthogonal complement of $V_{X_s} \in \mathbb{R}^{d \times s}$. Let $\Sigma_s = \text{diag}(\sigma_1, \ldots, \sigma_s)$ and $\Sigma_{s,\perp} = \text{diag}(\sigma_{s+1}, \ldots, \sigma_d)$. We use $\Delta_t := (\mathcal{A}^*\mathcal{A} - I)(XX^\top - U_t U_t^\top)$ to denote the vector consisting of measurement errors for $XX^\top - U_t U_t^\top$.

## B.1  HEURISTIC EXPLANATIONS OF THE DECOMPOSITION

A simple and intuitive approach for showing the implicit low rank bias is to directly analyze the growth of $V_{X_s}^\top U_t$ versus $V_{X_s,\perp}^\top U_t$. Ideally, the former grows faster than the latter, so that GD only learns the components in $X_s$.

By the update rule of GD (3),

$$V_{X_s,\perp}^\top U_{t+1} = V_{X_s,\perp}^\top \left[ I + \mu \mathcal{A}^*\mathcal{A}(XX^\top - U_t U_t^\top) \right] U_t$$

$$= \underbrace{V_{X_s,\perp}^\top \left[ I + \mu XX^\top - \mu U_t U_t^\top \right] U_t}_{=:G_{t,1}} + \mu \underbrace{V_{X_s,\perp}^\top \Delta_t U_t}_{=:G_{t,2}}$$

$$= G_{t,1} + \mu G_{t,2}.$$

For the first term $G_{t,1}$, we have

$$G_{t,1} = (I + \mu \Sigma_{s,\perp}^2) V_{X_s,\perp}^\top U_t - \mu V_{X_s,\perp}^\top U_t U_t^\top U_t$$

$$= (I + \mu \Sigma_{s,\perp}^2) V_{X_s,\perp}^\top U_t (I - \mu U_t U_t^\top) + \mathcal{O}(\mu^2),$$

where the last term $\mathcal{O}(\mu^2)$ is negligible when $\mu$ is sufficiently small. Since $\|\Sigma_{s,\perp}\| = \sigma_{s+1}$, the spectral norm of $G_{t,1}$ can be bounded by

$$\|G_{t,1}\| \le \|I + \mu \Sigma_{s,\perp}^2\| \cdot \|V_{X_s,\perp}^\top U_t\| \cdot \|I - \mu U_t U_t^\top\| + \mathcal{O}(\mu^2)$$

$$\le (1 + \mu \sigma_{s+1}^2) \|V_{X_s,\perp}^\top U_t\| + \mathcal{O}(\mu^2).$$

However, the main difference with the full-observation case (Jiang et al., 2022) is the second term $G_{t,2} := V_{X_s,\perp}^\top \Delta_t U_t$. Since the measurement errors $\Delta_t$ are small but arbitrary, it is hard to compare this term with $V_{X_s,\perp}^\top U_{t+1}$. As a result, we cannot directly bound the growth of $\|V_{X_s,\perp}^\top U_t\|$.

However, the aforementioned problem disappears if we turn to bound the growth of $\|V_{X_s,\perp}^\top U_{t+1} W_{t,\perp}\|$. To see this, first we deduce the following by repeatedly using $V_{X_s}^\top U_t W_{t,\perp} = 0$ due to the definition of $W_{t,\perp}$.

$$G_{t,1} W_{t,\perp} = V_{X_s,\perp}^\top \left[ I + \mu XX^\top - \mu U_t U_t^\top \right] U_t W_{t,\perp}$$

$$= V_{X_s,\perp}^\top (I + \mu XX^\top) U_t W_{t,\perp} - \mu V_{X_s,\perp}^\top U_t U_t^\top U_t W_{t,\perp}$$

$$= (I + \mu \Sigma_{s,\perp}^2) V_{X_s,\perp}^\top U_t W_{t,\perp} - \mu V_{X_s,\perp}^\top U_t (W_t W_t^\top + W_{t,\perp} W_{t,\perp}^\top) U_t^\top U_t W_{t,\perp}$$

$$= (I + \mu \Sigma_{s,\perp}^2) V_{X_s,\perp}^\top U_t W_{t,\perp} (I - \mu W_{t,\perp}^\top U_t^\top U_t W_{t,\perp})$$

$$\quad - \mu V_{X_s,\perp}^\top U_t W_t W_t^\top U_t^\top U_t W_{t,\perp} + \mathcal{O}(\mu^2),$$

$$G_{t,2}W_{t,\perp} = V_{X_s,\perp}^\top \Delta_t U_t W_{t,\perp} = V_{X_s,\perp}^\top \Delta_t V_{X_s,\perp} V_{X_s,\perp}^\top U_t W_{t,\perp},$$

So we have the following recursion:

$$V_{X_s,\perp}^\top U_{t+1} W_{t,\perp} = (I + \mu \Sigma_{s,\perp}^2 + \mu V_{X_s,\perp}^\top \Delta_t V_{X_s,\perp}) V_{X_s,\perp}^\top U_t W_{t,\perp} (I - \mu W_{t,\perp}^\top U_t^\top U_t W_{t,\perp})$$
$$- \mu V_{X_s,\perp}^\top U_t W_t W_t^\top U_t^\top U_t W_{t,\perp} + \mathcal{O}(\mu^2),$$

We further note that

$$V_{X_s,\perp}^\top U_{t+1} W_{t+1,\perp} = V_{X_s,\perp}^\top U_{t+1} W_t W_t^\top W_{t+1,\perp} + V_{X_s,\perp}^\top U_{t+1} W_{t,\perp} W_{t,\perp}^\top W_{t+1,\perp}, \quad (9)$$

which establishes the relationship between $V_{X_s,\perp}^\top U_{t+1} W_{t,\perp}$ and $V_{X_s,\perp}^\top U_{t+1} W_{t+1,\perp}$. To complete the proof we need to prove the following:

- The minimal eigenvalue of the *parallel component* $U_t W_t W_t^\top$ grows at a linear rate with speed strictly faster than $\sigma_{s+1}$.
- The term $\left\| V_{X_s,\perp}^\top V_{U_t W_t} \right\| \ll 1$, which implies that the first term in (9) is negligible.

## B.2 PROOF OUTLINE

An important intermediate step of our proof is the following result:

**Lemma B.1** *Under Assumptions 3.1 and 3.2, if the initialization scale $\alpha$ is sufficiently small, then for all $1 \le s \le \hat{r} \wedge r_*$ there exists a time $T_\alpha^s \in \mathbb{Z}_+$ such that*

$$\left\| X_s X_s^\top - U_{T_\alpha^s} U_{T_\alpha^s}^\top \right\|_F \le \kappa^2 \sqrt{r_*} \|X\|^2 \delta.$$

We begin with the *spectral alignment phase*, where we can make the following approximation

$$U_{t+1} \approx (I + \mu M) U_t \quad (10)$$

since $U_t$ is initially small. At some time $t = T_0$, $U_t$ would become approximately aligned with the first $s$ components, as long as there is a positive gap between the $s$-th and $(s+1)$-th largest eigenvalues of $M$. The choice of $T_0$ is subject to a trade-off such that the alignment takes effect while (10) does not induce large error.

We then enter the second phase which we call the *parallel matching* phase, in which the parallel components grow to a constant magnitude and is well-matched with the ground-truth (the orthogonal components remain small). Specifically, for small constants $c_i, 1 \le i \le 3$, we show that the followings are true in this phase:

(1). $\sigma_{\min}(V_{X_s}^\top U_t)$ grows exponentially fast until it reaches $c \cdot \sigma_s$ for some constant $c$ at some time $T_s$. Specifically, we have
$$\sigma_{\min}(V_{X_s}^\top U_{t+1} W_t) \ge \sigma_{\min}(V_{X_s}^\top U_t) \left(1 + \mu \sigma_s^2 - c_1 - \mu \sigma_{\min}^2(V_{X_s}^\top U_t)\right).$$

(2). When $t \le T_s$, the growth speed of $\|U_t W_{t,\perp}\|$ is slower than $\sigma_{\min}^2(V_{X_s}^\top U_t)$:
$$\|U_{t+1} W_{t+1,\perp}\| \le \left(1 + \mu \sigma_{s+1}^2 + c_2\right) \|U_t W_{t,\perp}\|.$$

(3). $\left\| V_{X_s,\perp}^\top V_{U_t W_t} \right\| \le c_3$ remains true until $t = T_s$.

These statements will be proven by induction. At $t = T_s$, we have $\sigma_{\min}(V_{X_s}^\top U_t) = \Theta(1)$, and we enter the *refinement phase* in which we show that the quantity $\left\| V_{X_s}^\top \left(U_t U_t^\top - X_s X_s^\top\right) \right\|_F$ decrease exponentially until it reaches $\mathcal{O}(\delta)$. Note that in general GD would not converge to an $o(1)$-neighbourhood of $X_s X_s^\top$ as the initialization scale $\alpha \to 0$, because $X_s X_s^\top$ is not the rank-$s$ minima of the RIP loss. As a result, in the refinement phase we can only expect to obtain $\left\| U_t U_t^\top - X_s X_s^\top \right\|_F \le \text{poly}(r) \cdot \delta$.

To conclude the proof of Theorem 4.1, we prove in Section 5 that the landscape of the matrix sensing loss with rank-$s$ parameterization, though non-convex, satisfies a local Polyak-Lojasiewicz (PL) condition within a neighborhood of *constant* radius. As a result, for sufficiently small $\delta$, GD converges linearly to global minima with good initialization. We then show that within a time period after the refinement phase, the GD trajectory is close to the trajectory of well-initialized GD for rank-$s$ parameterized matrix sensing. The length of this period goes to infinity when $\alpha \to 0$, thereby implying that GD finds the rank-$s$ minimizer with $o(1)$ error. The details are given in Appendix F.

## C  PROOF OF LEMMA B.1

In this section, we give the full proof of Lemma B.1, with some additional technical lemmas left to Appendix D. Appendices C.1 and C.2 are devoted to analyzing the spectral phase and parallel improvement phase, respectively. Appendix C.3 uses induction to characterize the low-rank GD trajectory in the parallel improvement phase. In Appendix C.4 we study the refinement phase, which allows us to derive Lemma B.1.

### C.1  THE SPECTRAL PHASE

Starting from a small $U_0 = \alpha U, \alpha \ll 1$, we first enter the spectral phase where GD behaves similar to power iteration. As in Stöger & Soltanolkotabi (2021), we refer to this phase as the spectral phase. Specifically, we have in the spectral phase that

$$U_{t+1} = \left(I + \mu\left(\mathcal{A}^*A\right)\left(XX^\top - U_tU_t^\top\right)\right)U_t \approx \left(I + \mu\left(\mathcal{A}^*A\right)\left(XX^\top\right)\right)U_t.$$

The approximation holds with high accuracy as long as $\|U_t\| \ll 1$. Moreover we have $M := \left(\mathcal{A}^*A\right)\left(XX^\top\right) \approx XX^\top$ by the RIP condition; when $\delta$ is sufficiently small, we can still ensure a positive eigen-gap of $M$. As a result, with small initialization $U_t$ would become approximately aligned with the top eigenvector $u_1$ of $M$. Since $\|M - XX^\top\| = \mathcal{O}(\delta\sqrt{r_*})$ by Proposition A.1, we have $\|u_1 - v_1\| = \mathcal{O}(\delta\sqrt{r_*})$ so that $\|V_{X_s}^\top V_{U_tW_t}\| = \mathcal{O}(\delta\sqrt{r_*})$. This proves the base case for the induction.

Formally, we define $M = \mathcal{A}^*\mathcal{A}(XX^\top)$, $Z_t = (I + \mu M)^t$ and $\widetilde{U}_t = Z_tU_0$. Suppose that $M = \sum_{i=1}^{\text{rank}(M)} \hat{\sigma}_i^2 \hat{v}_i\hat{v}_i^\top$ is the spectral decomposition of $M$. We additionally define $M_s = \sum_{i=1}^{\min\{s,\text{rank}(M)\}} \hat{\sigma}_i^2 \hat{v}_i\hat{v}_i^\top$. By Lemma A.4 and $\delta\sqrt{r_*} \le 10^{-3}\kappa$ as stated in Lemma B.1, we have $\hat{\sigma}_s \ge \sigma_s - 0.01\tau$ and $\hat{\sigma}_{s+1} \ge \sigma_{s+1} + 0.01\tau$, where $\tau = \sigma_s - \sigma_{s+1} > 0$. Additionally, let $L_t$ be the span of the top-$s$ left singular vectors of $U_t$. We make the following assumption on the initialization, which holds with high probability when it is i.i.d. Gaussian:

**Assumption C.1** *The matrix $V_{M_s}^\top U \in \mathbb{R}^{s\times r}$ has full row-rank i.e. $\rho = \sigma_{\min}\left(V_{M_s}^\top U\right) > 0$.*

Let

$$t^\star := \min\left\{i \in \mathbb{N} : \left\|\widetilde{U}_{i-1} - U_{i-1}\right\| > \left\|\widetilde{U}_{i-1}\right\|\right\},$$

the following lemma bounds the error of approximating $U_t$ via $\widetilde{U}_t$:

**Lemma C.1** *(Stöger & Soltanolkotabi, 2021, Lemma 8.1) Suppose that $\mathcal{A}$ satisfies the rank-1 RIP with constant $\delta_1$. For all integers $t$ such that $1 \le t \le t^\star$ it holds that*

$$\|E_t\| = \left\|U_t - \widetilde{U}_t\right\| \le 4\hat{\sigma}_1^{-2}\alpha^3 r_* (1+\delta_1)\left(1 + \mu\hat{\sigma}_1^2\right)^{3t} \|U\|^3. \tag{11}$$

**Corollary C.1** *We have*

$$t^* \ge \frac{\log\alpha^{-1} + \frac{1}{2}\log\frac{\rho\hat{\sigma}_1^2}{4(1+\delta_1)r_*}}{\log\left(1 + \mu\hat{\sigma}_1^2\right)}.$$

***Proof :*** By Lemma C.1 we have

$$\|E_t\| \le 4\hat{\sigma}_1^{-2}\alpha^3 r_* (1+\delta_1)\left(1 + \mu\hat{\sigma}_1^2\right)^{3t} \|U\|^3.$$

for all $t \le t^*$. On the other hand, we have

$$\begin{aligned}
\|\widetilde{U}_t\| &= \alpha\left\|(I + \mu M)^t U\right\| \\
&\ge \alpha(1+\mu\hat{\sigma}_1^2)^t\left\|\hat{v}_1\hat{v}_1^\top U\right\| \\
&\ge \left(1 + \mu\hat{\sigma}_1^2\right)^t\alpha\rho.
\end{aligned}$$

Thus, it follows from $\|E_{t^*}\| \ge \|\widetilde{U}_{t^*}\|$ that

$$\left(1 + \mu\hat{\sigma}_1^2\right)^{t^*} \ge \sqrt{\frac{\rho\hat{\sigma}_1^2}{4(1+\delta_1)r_*\|U\|^3}} \cdot \alpha^{-1} \Rightarrow t^* \ge \frac{\log\alpha^{-1} + \frac{1}{2}\log\frac{\rho\hat{\sigma}_1^2}{4(1+\delta_1)r_*}}{\log\left(1 + \mu\hat{\sigma}_1^2\right)}$$

as desired.  $\square$

**Lemma C.2** *There exists a time*

$$t = T_0 := \frac{2\log\alpha^{-1} + \log\frac{\rho\hat\sigma_1^2}{4r_*(1+\delta)}}{3\log(1+\mu\hat\sigma_1^2) - \log(1+\mu\hat\sigma_{s+1}^2)} \le t^*$$

*such that*

$$\left\| \boldsymbol{U}_t - \sum_{i=1}^s \alpha(1+\mu\hat\sigma_i^2)^t \hat{\boldsymbol{v}}_i \hat{\boldsymbol{v}}_i^\top \boldsymbol{U} \right\| \lesssim \alpha^\gamma$$

*where* $\gamma = 1 - \frac{2\log(1+\mu\hat\sigma_1^2)}{3\log(1+\mu\hat\sigma_1^2) - \log(1+\mu\hat\sigma_{s+1}^2)}$.

**Proof :** It's easy to check that $T_0 \le t^*$ by applying Corollary C.1.

We consider the following decomposition:

$$\left\| \boldsymbol{U}_t - \sum_{i=1}^s \alpha(1+\mu\hat\sigma_i^2)^t \hat{\boldsymbol{v}}_i \hat{\boldsymbol{v}}_i^\top \boldsymbol{U} \right\| \le \left\| \boldsymbol{U}_t - \widetilde{\boldsymbol{U}}_t \right\| + \left\| \widetilde{\boldsymbol{U}}_t - \sum_{i=1}^s \alpha(1+\mu\hat\sigma_i^2)^t \hat{\boldsymbol{v}}_i \hat{\boldsymbol{v}}_i^\top \boldsymbol{U} \right\|.$$

When $t \le t^*$, the first term can be bounded as

$$\|\boldsymbol{E}_t\| \le 4\hat\sigma_1^{-2}\alpha^3 r_* (1+\delta_1) \left(1+\mu\hat\sigma_1^2\right)^{3t}.$$

For the second term we have

$$\left\| \widetilde{\boldsymbol{U}}_t - \sum_{i=1}^s \alpha(1+\mu\hat\sigma_i^2)^t \hat{\boldsymbol{v}}_i \hat{\boldsymbol{v}}_i^\top \boldsymbol{U} \right\| \le \left\| \sum_{i=s+1}^{r_*} \alpha(1+\mu\hat\sigma_i^2)^t \hat{\boldsymbol{v}}_i \hat{\boldsymbol{v}}_i^\top \boldsymbol{U} \right\| \le \alpha \left(1+\mu\hat\sigma_{s+1}^2\right)^t.$$

In particular, the definition of $T_0$ implies that

$$\left\| \boldsymbol{U}_t - \sum_{i=1}^s \alpha(1+\mu\hat\sigma_i^2)^t \hat{\boldsymbol{v}}_i \hat{\boldsymbol{v}}_i^\top \boldsymbol{U} \right\| \lesssim \alpha^\gamma$$

as desired. □

We conclude this section with the following lemma, which states that initially the parallel component $\boldsymbol{U}_t\boldsymbol{W}_t$ would grow much faster than the noise term, and would become well-aligned with $\boldsymbol{X}_s$.

**Lemma C.3** *The following inequalities hold for $t = T_0$ when $\alpha \lesssim \rho^{-4\kappa}$ is sufficiently small:*

$$\|\boldsymbol{U}_t\| \le \|\boldsymbol{X}\| \tag{12a}$$

$$\sigma_{\min}\left(\boldsymbol{U}_{T_0}\boldsymbol{W}_{T_0}\right) \ge \rho \cdot \text{poly}(r_*)^{-1} \cdot \alpha^{1 - \frac{2\log(1+\mu\hat\sigma_s^2)}{3\log(1+\mu\hat\sigma_1^2) - \log(1+\mu\hat\sigma_{s+1}^2)}} \tag{12b}$$

$$\|\boldsymbol{U}_{T_0}\boldsymbol{W}_{T_0,\perp}\| \le \text{poly}(r_*) \cdot \alpha^{1 - \frac{2\log(1+\mu\hat\sigma_{s+1}^2)}{3\log(1+\mu\hat\sigma_1^2) - \log(1+\mu\hat\sigma_{s+1}^2)}} \tag{12c}$$

$$\left\|\boldsymbol{V}_{\boldsymbol{X}_s,\perp}^\top \boldsymbol{V}_{\boldsymbol{U}_{T_0}\boldsymbol{W}_{T_0}}\right\| \le 200\delta \tag{12d}$$

**Proof :** We prove this lemma by applying Corollary D.1 to $t = T_0$ defined in the previous lemma.

The inequality (12a) can be directly verified by using Lemma C.2:

$$\|\boldsymbol{U}_t\| \le \alpha\left(1+\mu\hat\sigma_i^2\right)^{T_0} + \alpha^\gamma \lesssim \text{poly}(r_*) \cdot \alpha^{\gamma/3} \le \|\boldsymbol{X}\|.$$

For the remaining inequalities, we first verify that the assumption in Corollary D.1:

$$\alpha\sigma_s(\boldsymbol{Z}_t) > 10\left(\alpha\sigma_{s+1}(\boldsymbol{Z}_t) + \|\boldsymbol{E}_t\|\right). \tag{13}$$

By definition of $\boldsymbol{Z}_t$, we can see that

$$\alpha\sigma_{s+1}(\boldsymbol{Z}_{T_0}) + \|\boldsymbol{E}_{T_0}\| \le \alpha\left(1+\mu\hat\sigma_{s+1}^2\right)^{T_0} + \|\boldsymbol{E}_{T_0}\|$$

$$\lesssim \alpha^\gamma \lesssim 0.1\rho\alpha^{1 - \frac{2\log(1+\mu\hat\sigma_s^2)}{3\log(1+\mu\hat\sigma_1^2) - \log(1+\mu\hat\sigma_{s+1}^2)}}$$

$$\le 0.1\alpha\sigma_s(\boldsymbol{Z}_{T_0})$$

when $\alpha \leq \text{poly}(r_*)^{-1}$, so that (13) holds. As a result, we have

$$
\begin{aligned}
\sigma_s \left(\boldsymbol{U}_t \boldsymbol{W}_t\right) &\geqslant 0.4\alpha\sigma_s \left(\boldsymbol{Z}_t\right) \sigma_{\min} \left(\boldsymbol{V}_L^T \boldsymbol{U}\right) \\
&\geq 0.4\text{poly}(r_*)^{-1} \cdot \alpha\rho \left(1 + \mu\hat{\sigma}_s^2\right)^{T_0} \\
&\gtrsim \rho \cdot \alpha^{1 - \frac{2\log(1+\mu\hat{\sigma}_s^2)}{3\log(1+\mu\hat{\sigma}_1)-\log(1+\mu\hat{\sigma}_{s+1}^2)}} \\
\|\boldsymbol{U}_t \boldsymbol{W}_{t,\perp}\| &\leqslant 2 \left(\alpha\sigma_{s+1}^2 \left(\boldsymbol{Z}_t\right) \|\boldsymbol{U}\| + \|\boldsymbol{E}_t\|\right) \\
&\lesssim \alpha^{1 - \frac{2\log(1+\mu\hat{\sigma}_{s+1}^2)}{3\log(1+\mu\hat{\sigma}_1^2)-\log(1+\mu\hat{\sigma}_{s+1}^2)}} \\
\left\|\boldsymbol{V}_{\boldsymbol{X}_s,\perp}^T \boldsymbol{V}_{\boldsymbol{U}_t \boldsymbol{W}_t}\right\| &\leqslant 100 \left(\delta + \frac{\alpha\sigma_{s+1} \left(\boldsymbol{Z}_t\right) \|\boldsymbol{U}\| + \|\boldsymbol{E}_t\|}{\alpha\rho\sigma_s \left(\boldsymbol{Z}_t\right)}\right) \\
&\leq 100 \left(\delta\alpha^{\frac{2\log(1+\mu\hat{\sigma}_s^2)-2\log(1+\mu\hat{\sigma}_{s+1}^2)}{3\log(1+\mu\hat{\sigma}_1^2)-\log(1+\mu\hat{\sigma}_{s+1}^2)}}\right) \qquad \leq 200\delta.
\end{aligned}
\tag{14}
$$

The conclusion follows. $\qquad\qquad\square$

## C.2  THE PARALLEL IMPROVEMENT PHASE

### C.2.1  THE SIGNAL TERM

In the following we estimate $\sigma_{\min} \left(V_{\boldsymbol{X}_s}^\top \boldsymbol{U}_{t+1} \boldsymbol{W}_t\right)$. We state our main result of this section in the lemma below.

**Lemma C.4** *Suppose that* $V_{\boldsymbol{X}_s}^\top \boldsymbol{U}_t \in \mathbb{R}^{s \times r}$ *is of full rank,* $\mu < 10^{-4}\|\boldsymbol{X}\|^{-2}$, $c_3 < 10^{-3}\kappa^{-1}$, $\left\|\left(\mathcal{A}^*\mathcal{A} - I\right)\left(\boldsymbol{X}\boldsymbol{X}^\top - \boldsymbol{U}_t\boldsymbol{U}_t^\top\right)\right\| < 10^{-3}\kappa^{-1}\|\boldsymbol{X}\|^2$ *and* $\|V_{\boldsymbol{X}_s^\perp} V_{\boldsymbol{U}_t \boldsymbol{W}_t}\| \leq c_3$, *then we have*

$$
\begin{aligned}
\sigma_{\min}(\boldsymbol{V}_{\boldsymbol{X}_s}^\top \boldsymbol{U}_{t+1}) &\geq \sigma_{\min}(\boldsymbol{V}_{\boldsymbol{X}_s}^\top \boldsymbol{U}_{t+1} \boldsymbol{W}_t) \\
&\geq \left(1 + \mu \left(\sigma_s^2 - (5c_3 + 20\delta)\|\boldsymbol{X}\|^2\right) - 500\mu^2\|\boldsymbol{X}\|^4\right) \left(1 - \mu\sigma_{\min}^2(\boldsymbol{V}_{\boldsymbol{X}_s}^\top \boldsymbol{U}_t)\right) \sigma_{\min}(\boldsymbol{V}_{\boldsymbol{X}_s}^\top \boldsymbol{U}_t).
\end{aligned}
$$

*Proof* : The update rule of GD implies that

$$
\begin{aligned}
&\boldsymbol{V}_{\boldsymbol{X}_s}^\top \boldsymbol{U}_{t+1} \boldsymbol{W}_t \\
&= \boldsymbol{V}_{\boldsymbol{X}_s}^\top \left(\boldsymbol{I} + \mu(\boldsymbol{X}\boldsymbol{X}^\top - \boldsymbol{U}_t\boldsymbol{U}_t^\top) + \mu\boldsymbol{\Delta}_t\right) \boldsymbol{U}_t \boldsymbol{W}_t \\
&= \boldsymbol{V}_{\boldsymbol{X}_s}^\top \left(\boldsymbol{I} + \mu(\boldsymbol{X}_s\boldsymbol{X}_s^\top - \boldsymbol{U}_t\boldsymbol{U}_t^\top) + \mu\boldsymbol{\Delta}_t\right) \boldsymbol{U}_t \boldsymbol{W}_t &(15\text{a}) \\
&= (\boldsymbol{I} + \mu\boldsymbol{\Sigma}_s^2)\boldsymbol{V}_{\boldsymbol{X}_s}^\top \boldsymbol{U}_t \boldsymbol{W}_t - \mu\boldsymbol{V}_{\boldsymbol{X}_s}^\top \boldsymbol{U}_t\boldsymbol{U}_t^\top \boldsymbol{U}_t \boldsymbol{W}_t + \mu\boldsymbol{V}_{\boldsymbol{X}_s}^\top \boldsymbol{\Delta}_t \boldsymbol{U}_t \boldsymbol{W}_t &(15\text{b}) \\
&= (\boldsymbol{I} + \mu\boldsymbol{\Sigma}_s^2)\boldsymbol{V}_{\boldsymbol{X}_s}^\top \boldsymbol{U}_t \boldsymbol{W}_t - \mu\boldsymbol{V}_{\boldsymbol{X}}^\top \boldsymbol{U}_t\boldsymbol{U}_t^\top \boldsymbol{V}_{\boldsymbol{X}_s} \boldsymbol{V}_{\boldsymbol{X}_s}^\top \boldsymbol{U}_t \boldsymbol{W}_t - \mu\boldsymbol{V}_{\boldsymbol{X}}^\top \boldsymbol{U}_t\boldsymbol{U}_t^\top \boldsymbol{V}_{\boldsymbol{X}_s,\perp} \boldsymbol{V}_{\boldsymbol{X}_s,\perp}^\top \boldsymbol{U}_t \boldsymbol{W}_t \\
&\quad + \mu\boldsymbol{V}_{\boldsymbol{X}}^\top \boldsymbol{\Delta}_t \boldsymbol{U}_t \boldsymbol{W}_t \\
&= (\boldsymbol{I} + \mu\boldsymbol{\Sigma}_s^2)\boldsymbol{V}_{\boldsymbol{X}_s}^\top \boldsymbol{U}_t \boldsymbol{W}_t(\boldsymbol{I} - \mu\boldsymbol{W}_t^\top \boldsymbol{U}_t^\top \boldsymbol{V}_{\boldsymbol{X}_s} \boldsymbol{V}_{\boldsymbol{X}_s}^\top \boldsymbol{U}_t \boldsymbol{W}_t) + \mu\boldsymbol{V}_{\boldsymbol{X}_s}^\top \boldsymbol{\Delta}_t \boldsymbol{U}_t \boldsymbol{W}_t \\
&\quad - \mu\boldsymbol{V}_{\boldsymbol{X}}^\top \boldsymbol{U}_t\boldsymbol{U}_t^\top \boldsymbol{V}_{\boldsymbol{X}_s,\perp} \boldsymbol{V}_{\boldsymbol{X}_s,\perp}^\top \boldsymbol{U}_t \boldsymbol{W}_t + \mu^2\boldsymbol{\Sigma}_s^2\boldsymbol{V}_{\boldsymbol{X}_s}^\top \boldsymbol{U}_t \boldsymbol{W}_t\boldsymbol{W}_t^\top \boldsymbol{U}_t^\top \boldsymbol{V}_{\boldsymbol{X}_s} \boldsymbol{V}_{\boldsymbol{X}_s}^\top \boldsymbol{U}_t \boldsymbol{W}_t &(15\text{c})
\end{aligned}
$$

where (15a) follows from $\boldsymbol{V}_{\boldsymbol{X}_s}^\top \boldsymbol{X}\boldsymbol{X}^\top = \boldsymbol{V}_{\boldsymbol{X}_s}^\top \boldsymbol{X}_s\boldsymbol{X}_s^\top + \boldsymbol{V}_{\boldsymbol{X}_s}^\top \boldsymbol{X}_{s,\perp}\boldsymbol{X}_{s,\perp}^\top$ and $\boldsymbol{V}_{\boldsymbol{X}_s}^\top \boldsymbol{X}_{s,\perp} = 0$; (15b) follows from $\boldsymbol{V}_{\boldsymbol{X}_s}^\top \boldsymbol{X}_s\boldsymbol{X}_s^\top = \boldsymbol{V}_{\boldsymbol{X}_s}^\top \boldsymbol{V}_{\boldsymbol{X}_s}\boldsymbol{\Sigma}_s\boldsymbol{V}_{\boldsymbol{X}_s}^\top = \boldsymbol{\Sigma}_s\boldsymbol{V}_{\boldsymbol{X}_s}^\top$, and (15c) follows from $\boldsymbol{V}_{\boldsymbol{X}_s}^\top \boldsymbol{U}_t = \boldsymbol{V}_{\boldsymbol{X}_s}^\top \boldsymbol{U}_t \boldsymbol{W}_t\boldsymbol{W}_t^\top + \boldsymbol{V}_{\boldsymbol{X}_s}^\top \boldsymbol{U}_t \boldsymbol{W}_{t,\perp}\boldsymbol{W}_{t,\perp}^\top = \boldsymbol{V}_{\boldsymbol{X}_s}^\top \boldsymbol{U}_t \boldsymbol{W}_t\boldsymbol{W}_t^\top$ by definition of $\boldsymbol{W}_t$ and $\boldsymbol{W}_{t,\perp}$.

We now relate the last three terms in (15c) to $\boldsymbol{V}_{\boldsymbol{X}_s}^\top \boldsymbol{U}_t \boldsymbol{W}_t$. Since $\boldsymbol{V}_{\boldsymbol{X}_s}^\top \boldsymbol{U}_t \boldsymbol{W}_t$ is assumed to be invertible, so is $\boldsymbol{V}_{\boldsymbol{X}_s}^\top \boldsymbol{V}_{\boldsymbol{U}_t \boldsymbol{W}_t}, \boldsymbol{\Sigma}_{\boldsymbol{U}_t \boldsymbol{W}_t}$ and $\boldsymbol{W}_{\boldsymbol{U}_t \boldsymbol{W}_t}$, thus we have

$$
\begin{aligned}
\boldsymbol{U}_t \boldsymbol{W}_t &= \boldsymbol{U}_t \boldsymbol{W}_t(\boldsymbol{V}_{\boldsymbol{X}_s}^\top \boldsymbol{U}_t \boldsymbol{W}_t)^{-1}\boldsymbol{V}_{\boldsymbol{X}_s}^\top \boldsymbol{U}_t \boldsymbol{W}_t \\
&= \boldsymbol{U}_t \boldsymbol{W}_t \left(\boldsymbol{V}_{\boldsymbol{X}_s}^\top \boldsymbol{V}_{\boldsymbol{U}_t \boldsymbol{W}_t}\boldsymbol{\Sigma}_{\boldsymbol{U}_t \boldsymbol{W}_t}\boldsymbol{W}_{\boldsymbol{U}_t \boldsymbol{W}_t}^\top\right)^{-1} \boldsymbol{V}_{\boldsymbol{X}_s}^\top \boldsymbol{U}_t \boldsymbol{W}_t &(16) \\
&= \boldsymbol{V}_{\boldsymbol{U}_t \boldsymbol{W}_t} \left(\boldsymbol{V}_{\boldsymbol{X}_s}^\top \boldsymbol{V}_{\boldsymbol{U}_t \boldsymbol{W}_t}\right)^{-1} \boldsymbol{V}_{\boldsymbol{X}_s}^\top \boldsymbol{U}_t \boldsymbol{W}_t.
\end{aligned}
$$

Plugging (16) into the second and third terms of (15) and re-arranging, we deduce that

$$
\begin{aligned}
&V_{X_s}^\top U_{t+1} W_t \\
&= \left(I + \mu(\Sigma_s^2 + P_1 + P_2)\right) V_{X_s}^\top U_t W_t (I - \mu W_t^\top U_t^\top V_{X_s} V_{X_s}^\top U_t W_t) \\
&\quad + \mu^2 \left(\Sigma_s^2 + P_1 + P_2\right) V_{X_s}^\top U_t W_t W_t^\top U_t^\top V_{X_s} V_{X_s}^\top U_t W_t \\
&= \left[I + \mu\left(\Sigma_s^2 + P_1 + P_2\right) + \mu^2\left(\Sigma_s^2 + P_1 + P_2\right) V_{X_s}^\top U_t W_t W_t^\top U_t^\top V_{X_s} \left(I - \mu V_{X_s}^\top U_t W_t W_t^\top U_t^\top V_{X_s}\right)^{-1}\right] \cdot \\
&\quad V_{X_s}^\top U_t W_t (I - \mu W_t^\top U_t^\top V_{X_s} V_{X_s}^\top U_t W_t)
\end{aligned}
\tag{17}
$$

where we use the equation $A = (I - \mu A A^\top)^{-1} A (I - \mu A^\top A)$ with $A = V_{X_s}^\top U_t W_t$ (when $\mu < \frac{1}{9\|X\|^2}$, $I - \mu A A^\top$ is invertible), and

$$
\begin{aligned}
P_1 &= V_{X_s}^\top U_t U_t^\top V_{X_s,\perp} V_{X_s,\perp}^\top V_{U_t W_t} \left(V_{X_s}^\top V_{U_t W_t}\right)^{-1} \\
P_2 &= V_{X_s}^\top \Delta_t V_{U_t W_t} \left(V_{X_s}^\top V_{U_t W_t}\right)^{-1}
\end{aligned}
\tag{18}
$$

By assumption we have

$$
\sigma_{\min}\left(V_{X_s}^\top V_{U_t W_t}\right) \geq \sqrt{1 - \left\|V_{X_s,\perp}^\top V_{U_t W_t}\right\|^2} \geq \frac{1}{2},
$$

so that

$$
\|P_1\| \leq \left\|V_{X_s}^\top U_t U_t^\top V_{X_s,\perp}\right\| \cdot \left\|V_{X_s,\perp}^\top V_{U_t W_t}\right\| \cdot \left\|\left(V_{X_s}^\top V_{U_t W_t}\right)^{-1}\right\| \leq 5 c_3 \|X\|^2 \leq 5\|X\|^2
\tag{19}
$$

and by our assumption we have

$$
\|P_2\| \leq \left\|\left(V_{X_s}^\top V_{U_t W_t}\right)^{-1}\right\| \cdot \|\Delta_t\| \leq 10^{-2} \kappa^{-1} r^{-\frac{1}{2}}.
\tag{20}
$$

Moreover, note that $\|\Sigma_s\|^2 = \|X\|^2$, and when $\mu < \frac{1}{10\|X\|^2}$ we have $\left\|\left(I - \mu V_{X_s}^\top U_t W_t W_t^\top U_t^\top V_{X_s}\right)^{-1}\right\| < 2$. Thus

$$
\left\|\left(\Sigma_s^2 + P_1 + P_2\right) V_{X_s}^\top U_t W_t W_t^\top U_t^\top V_{X_s} \left(I - \mu V_{X_s}^\top U_t W_t W_t^\top U_t^\top V_{X_s}\right)^{-1}\right\| \leq 500\|X\|^4.
$$

The equation (18) implies that

$$
\begin{aligned}
&\sigma_{\min}(V_{X_s}^\top U_{t+1} W_t) \\
&\geq \sigma_{\min}\left(I + \mu\Sigma_s^2 + P_1 + P_2 + \left(\Sigma_s^2 + P_1 + P_2\right) V_{X_s}^\top U_t W_t W_t^\top U_t^\top V_{X_s} \left(I - \mu V_{X_s}^\top U_t W_t W_t^\top U_t^\top V_{X_s}\right)^{-1}\right) \cdot \\
&\quad \sigma_{\min}\left(V_{X_s}^\top U_t W_t (I - \mu W_t^\top U_t^\top V_{X_s} V_{X_s}^\top U_t W_t)\right) \\
&\geq \left(1 + \mu\sigma_{\min}^2(\Sigma_s) - \mu\|P_1\| - \mu\|P_2\| - 500\mu^2\|X\|^4\right) \sigma_{\min}(V_{X_s}^\top U_t)\left(1 - \mu\sigma_{\min}^2(V_{X_s}^\top U_t)\right) \\
&= \left(1 + \mu\sigma_s^2 - \mu\|P_1\| - \mu\|P_2\| - 500\mu^2\|X\|^4\right) \sigma_{\min}(V_{X_s}^\top U_t)\left(1 - \mu\sigma_{\min}^2(V_{X_s}^\top U_t)\right)
\end{aligned}
$$

Recall that $P_1$ and $P_2$ are bounded in (19) and (20) respectively, so we have that

$$
\begin{aligned}
&\sigma_{\min}(V_{X_s}^\top U_{t+1}) \\
&\geq \sigma_{\min}(V_{X_s}^\top U_{t+1} W_t) \\
&\geq \left(1 + \mu\left(\sigma_s^2 - (5c_3 + 20\delta)\|X\|^2\right) - 500\mu^2\|X\|^4\right)\left(1 - \mu\sigma_{\min}^2(V_{X_s}^\top U_t)\right) \sigma_{\min}(V_{X_s}^\top U_t).
\end{aligned}
$$

The conclusion follows. $\qquad\square$

The corollaries below immediately follow from Lemma C.4.

**Corollary C.2** *Under the conditions in Lemma C.4, if* $\sigma_{\min}^2(V_{X_s}^\top U_t) < 0.3(\sigma_s^2 - \sigma_{s+1}^2) = 0.3\kappa^{-1}\|X\|^2$, *then we have*

$$
\sigma_{\min}(V_{X_s}^\top U_{t+1}) \geq \left(1 + 0.5\mu(\sigma_s^2 + \sigma_{s+1}^2)\right) \sigma_{\min}(V_{X_s}^\top U_t).
$$

**Corollary C.3** *Under the conditions in Lemma C.4, if*

$$
\sigma_{\min}^2(V_{X_s}^\top U_t) \leq \sigma_s^2 - 50(c_3 + \delta\sqrt{r})\|X\|^2,
$$

*then we have that* $\sigma_{\min}(V_{X_s}^\top U_{t+1}) \geq \sigma_{\min}(V_{X_s}^\top U_t).$

### C.2.2 THE NOISE TERM

In this section we turn to analyze the noise term. The main result of this section is presented in the following:

**Lemma C.5** *Suppose that $V_{X_s}^\top U_{t+1} W_t \in \mathbb{R}^{s \times r}$ is of full rank,*

$$\|V_{X_s, \perp} V_{U_t W_t}\| \le c_3 < 10^{-3} \kappa^{-1}$$

*and*

$$\left\| (\mathcal{A}^* \mathcal{A} - I)(XX^\top - U_t U_t^\top) \right\| \le 10^{-3} \kappa^{-1} c_3 \|X\|^2,$$

*then we have*

$$\|U_{t+1} W_{t+1, \perp}\| \le \left( 1 + \mu \sigma_{s+1}^2 + 30\mu \|X\|^2 c_3 + 0.1 \mu^2 \|X\|^4 \right) \|U_t W_{t, \perp}\|.$$

***Proof :*** By the definition of $W_{t, \perp}$, we have $V_{X_s}^\top U_t W_{t, \perp} = 0$, thus $\|U_t W_{t, \perp}\| = \left\| V_{X_s, \perp}^\top U_t W_{t, \perp} \right\|$. The latter can be decomposed as follows:

$$V_{X_s, \perp}^\top U_{t+1} W_{t+1, \perp} = \underbrace{V_{X_s, \perp}^\top U_{t+1} W_t W_t^\top W_{t+1, \perp}}_{=(a)} + \underbrace{V_{X_s, \perp}^\top U_{t+1} W_{t, \perp} W_{t, \perp}^\top W_{t+1, \perp}}_{=(b)}.$$

In the following, we are going to show that the term (a) is bounded by $c \cdot \mu$ where $c$ is a small constant, while (b) grows linearly with a slow speed.

*Bounding summand (a).* Since

$$0 = V_{X_s}^\top U_{t+1} W_{t+1, \perp} = V_{X_s}^\top U_{t+1} W_t W_t^\top W_{t+1, \perp} + V_{X_s}^\top U_{t+1} W_{t, \perp} W_{t, \perp}^\top W_{t+1, \perp}$$

by definition, we have

$$W_t^\top W_{t+1, \perp} = -\left( V_{X_s}^\top U_{t+1} W_t \right)^{-1} V_{X_s}^\top U_{t+1} W_{t, \perp} W_{t, \perp}^\top W_{t+1, \perp}. \tag{21}$$

Thus the summand (a) can be rewritten as follows:

$$V_{X_s, \perp}^\top U_{t+1} W_t W_t^\top W_{t+1, \perp}$$

$$= -V_{X_s, \perp}^\top U_{t+1} W_t \left( V_{X_s}^\top U_{t+1} W_t \right)^{-1} V_{X_s}^\top U_{t+1} W_{t, \perp} W_{t, \perp}^\top W_{t+1, \perp} \tag{22a}$$

$$= -V_{X_s, \perp}^\top U_{t+1} W_t \left( V_{X_s}^\top V_{U_{t+1} W_t} \Sigma_{U_{t+1} W_t} W_{U_{t+1} W_t} \right)^{-1} V_{X_s}^\top U_{t+1} W_{t, \perp} W_{t, \perp}^\top W_{t+1, \perp}$$

$$= -V_{X_s, \perp}^\top V_{U_{t+1} W_t} \left( V_{X_s}^\top V_{U_{t+1} W_t} \right)^{-1} V_{X_s}^\top U_{t+1} W_{t, \perp} W_{t, \perp}^\top W_{t+1, \perp} \tag{22b}$$

$$= -V_{X_s, \perp}^\top V_{U_{t+1} W_t} \left( V_{X_s}^\top V_{U_{t+1} W_t} \right)^{-1} V_{X_s}^\top \left( I + \mu \mathcal{A}^* \mathcal{A} \left( XX^\top - U_t U_t^\top \right) \right) U_t W_{t, \perp} W_{t, \perp}^\top W_{t+1, \perp}$$

$$= -\mu V_{X_s, \perp}^\top V_{U_{t+1} W_t} \left( V_{X_s}^\top V_{U_{t+1} W_t} \right)^{-1} V_{X_s}^\top \left[ \left( XX^\top - U_t U_t^\top \right) + \Delta_t \right] U_t W_{t, \perp} W_{t, \perp}^\top W_{t+1, \perp} \tag{22c}$$

$$= \mu V_{X_s, \perp}^\top V_{U_{t+1} W_t} \left( V_{X_s}^\top V_{U_{t+1} W_t} \right)^{-1} V_{X_s}^\top \left[ U_t U_t^\top - \Delta_t \right] U_t W_{t, \perp} W_{t, \perp}^\top W_{t+1, \perp}$$

$$= \mu V_{X_s, \perp}^\top V_{U_{t+1} W_t} \left( V_{X_s}^\top V_{U_{t+1} W_t} \right)^{-1} M_1 V_{X_s, \perp}^\top U_t W_{t, \perp} W_{t, \perp}^\top W_{t+1, \perp},$$

where $M_1 = V_{X_s}^\top \left[ U_t U_t^\top V_{X_s, \perp} - \Delta_t V_{X_s, \perp} \right]$. In (22), (22a) follows from (21), (22b) holds since $\Sigma_{U_{t+1} W_t} W_{U_{t+1} W_t}^\top \in \mathbb{R}^{s \times s}$ is invertible, and in (22c) we use $V_{X_s}^\top U_t W_{t, \perp} = 0$. It follows that

$$\|(a)\| \le \mu \left\| V_{X_s, \perp}^\top V_{U_{t+1} W_t} \right\| \cdot \left\| \left( V_{X_s}^\top V_{U_{t+1} W_t} \right)^{-1} \right\| \|M_1\| \left\| V_{X_s, \perp}^\top U_t W_{t, \perp} \right\|. \tag{23}$$

By Lemma D.4 we have $\left\| V_{X_s, \perp}^\top V_{U_{t+1} W_t} \right\| \le 0.01$, which implies that

$$\left\| \left( V_{X_s}^\top V_{U_{t+1} W_t} \right)^{-1} \right\| = \sigma_{\min}^{-1} \left( V_{X_s}^\top V_{U_{t+1} W_t} \right) = \left( 1 - \left\| V_{X_s, \perp}^\top V_{U_{t+1} W_t} \right\|^2 \right)^{-\frac{1}{2}} \ge \frac{1}{2}. \tag{24}$$

Lastly, we bound $M_1$ as follows:

$$\|M_1\| \leq \|V_{X_s}^\top U_t U_t^\top V_{X_s,\perp}\| + \|(\mathcal{A}^*\mathcal{A} - I)(XX^\top - U_t U_t^\top)\|$$
$$\leq \|V_{X_s}^\top U_t W_t\| \cdot \|V_{X_s,\perp}^\top U_t W_t\| + 10^{-3}\kappa^{-1} c_3 \|X\|^2 \tag{25}$$
$$\leq 10\|X\|^2 c_3.$$

where the second inequality follows from our assumption on $\|(\mathcal{A}^*\mathcal{A} - I)(XX^\top - U_t U_t^\top)\|$. Combining (23), (24) and (25) yields

$$\|(a)\| \leq 20\mu\|X\|^2 c_3 \|U_t W_{t,\perp}\|.$$

*Bounding summand (b).* This is the main component in the error term. We'll see that although this term can grow exponentially fast, the growth speed is slower than the minimal eigenvalue of the parallel component.

We have

$$V_{X_s,\perp}^\top U_{t+1} W_{t,\perp}$$
$$= V_{X_s,\perp}^\top \left[I + \mu(XX^\top - U_t U_t^\top) + \mu(\mathcal{A}^*\mathcal{A} - I)(XX^\top - U_t U_t^\top)\right] U_t W_{t,\perp} \tag{26a}$$

$$= \left(I + \mu\Sigma_{s,\perp}^2 - \mu V_{X_s,\perp}^\top U_t U_t^\top V_{X_s,\perp} + \mu \underbrace{V_{X_s,\perp}^\top \Delta_t V_{X_s,\perp}}_{=:M_2}\right) V_{X_s,\perp}^\top U_t W_{t,\perp} \tag{26b}$$

$$= \left(I + \mu\Sigma_{s,\perp}^2 - \mu V_{X_s,\perp}^\top U_t W_t W_t^\top U_t^\top V_{X_s,\perp} + \mu M_2\right) V_{X_s,\perp}^\top U_t W_{t,\perp} \left(I - \mu W_{t,\perp}^\top U_t^\top U_t W_{t,\perp}\right) \tag{26c}$$

$$+ \mu^2 \left(\Sigma_{s,\perp}^2 - V_{X_s,\perp}^\top U_t W_t W_t^\top U_t^\top V_{X_s,\perp} + M_2\right) V_{X_s,\perp}^\top U_t W_{t,\perp} W_{t,\perp}^\top U_t^\top U_t W_{t,\perp} \tag{26d}$$

where we recall that $\Sigma_{s,\perp}^2 = \mathrm{diag}(\sigma_{s+1}^2, \cdots, \sigma_r^2, 0, \cdots, 0) \in \mathbb{R}^{(d-s)\times(d-s)}$. In (26), (26a) follows from the update rule of GD, (26b) is obtained from $V_{X_s,\perp}^\top XX^\top = \Sigma_{s,\perp}^2 V_{X_s,\perp}^\top$ and $U_t W_{t,\perp} = V_{X_s} V_{X_s}^\top U_t W_{t,\perp} + V_{X_s,\perp} V_{X_s,\perp}^\top U_t W_{t,\perp} = V_{X_s,\perp} V_{X_s,\perp}^\top U_t W_{t,\perp}$, and lastly in (26d) we use

$$V_{X_s,\perp}^\top U_t U_t^\top V_{X_s,\perp} V_{X_s,\perp}^\top U_t W_{t,\perp}$$
$$= V_{X_s,\perp}^\top U_t W_t W_t^\top U_t^\top V_{X_s,\perp} V_{X_s,\perp}^\top U_t W_{t,\perp} + V_{X_s,\perp}^\top U_t W_{t,\perp} W_{t,\perp}^\top U_t^\top V_{X_s,\perp} V_{X_s,\perp}^\top U_t W_{t,\perp}$$
$$= V_{X_s,\perp}^\top U_t W_t W_t^\top U_t^\top V_{X_s,\perp} V_{X_s,\perp}^\top U_t W_{t,\perp} + V_{X_s,\perp}^\top U_t W_{t,\perp} W_{t,\perp}^\top U_t^\top U_t W_{t,\perp}.$$

It follows that

$$\|V_{X_s,\perp}^\top U_{t+1} W_{t,\perp}\|$$
$$\leq \left(\|I - \mu V_{X_s,\perp}^\top U_t W_t W_t^\top U_t^\top V_{X_s,\perp}\| + \mu\|\Sigma_{s,\perp}\|^2 + \mu\|M_2\|\right) \|V_{X_s,\perp}^\top U_t W_{t,\perp}\| \left(I - \mu\|V_{X_s,\perp}^\top U_t W_{t,\perp}\|^2\right)$$
$$+ \mu^2 \|U_t W_{t,\perp}\|^3 \left(\sigma_{s+1}^2 + \|U_t\|^2 + 10^{-3}\kappa^{-1} c_3\|X\|^2\right)$$
$$\leq \left(1 + \mu\sigma_{s+1}^2 + \mu\|M_2\|\right) \|U_t W_{t,\perp}\| \left(I - \mu\|U_t W_{t,\perp}\|^2\right) + 0.1\mu^2\|X\|^4 \|U_t W_{t,\perp}\|$$
$$\leq \|U_t W_{t,\perp}\| \left(1 + \mu\sigma_{s+1}^2 + 10\mu\delta\sqrt{r}\|X\|^2 + 0.1\mu^2\|X\|^4\right)$$

To summarize, we have

$$\|U_{t+1} W_{t+1,\perp}\| \leq \left(1 + \mu\sigma_{s+1}^2 + 30\mu\|X\|^2 c_3 + 0.1\mu^2\|X\|^4\right) \|U_t W_{t,\perp}\|$$

as desired. □

To bound the growth speed of the orthogonal component, we need to show that the quantity $\left\|V_{X_s,\perp}^\top V_{U_t W_t}\right\|$ remains small. The following lemma serves to complete an induction step from $t$ to $t+1$:

**Lemma C.6** *Suppose that* $\|V_{X_s,\perp} V_{U_t W_t}\| \leq c_3$ *and* $\|U_t W_{t,\perp}\| \leq c_4$, *with* $\max\{c_3, c_4\|X\|^{-1}\} \leq 10^{-3}\kappa^{-1}$ *and* $\|(\mathcal{A}^*\mathcal{A} - I)(XX^\top - U_t U_t^\top)\| \leq 10^{-3}\kappa^{-1} c_3\|X\|^2$ *(where $\kappa$ is the condition number defined in Section 3.1) and $\mu \leq 10^{-4}\kappa^{-1}\|X\|^{-2} c_3$, then we have* $\|V_{X_s,\perp} V_{U_{t+1} W_{t+1}}\| \leq c_3$.

**Proof :** Let $\boldsymbol{M}_t = \mathcal{A}^*\mathcal{A}(\boldsymbol{X}\boldsymbol{X}^\top - \boldsymbol{U}_t\boldsymbol{U}_t^\top)$, so the update rule of GD implies that

$$
\begin{aligned}
\boldsymbol{U}_{t+1}\boldsymbol{W}_{t+1} &= (\boldsymbol{I} + \mu\boldsymbol{M}_t)\boldsymbol{U}_t\boldsymbol{W}_{t+1} \\
&= (\boldsymbol{I} + \mu\boldsymbol{M}_t)\left(\boldsymbol{U}_t\boldsymbol{W}_t\boldsymbol{W}_t^\top\boldsymbol{W}_{t+1} + \boldsymbol{U}_t\boldsymbol{W}_{t,\perp}\boldsymbol{W}_{t,\perp}^\top\boldsymbol{W}_{t+1}\right) \\
&= (\boldsymbol{I} + \mu\boldsymbol{M}_t)\left(\boldsymbol{V}_{\boldsymbol{U}_t\boldsymbol{W}_t}\boldsymbol{V}_{\boldsymbol{U}_t\boldsymbol{W}}^\top\boldsymbol{U}_t\boldsymbol{W}_t\boldsymbol{W}_t^\top\boldsymbol{W}_{t+1} + \boldsymbol{U}_t\boldsymbol{W}_{t,\perp}\boldsymbol{W}_{t,\perp}^\top\boldsymbol{W}_{t+1}\right) \\
&= \underbrace{(\boldsymbol{I} + \mu\boldsymbol{M}_t)(\boldsymbol{I} + \boldsymbol{P})\boldsymbol{V}_{\boldsymbol{U}_t\boldsymbol{W}_t}}_{:=\hat{\boldsymbol{Z}}}\,\boldsymbol{V}_{\boldsymbol{U}_t\boldsymbol{W}_t}^\top\boldsymbol{U}_t\boldsymbol{W}_t\boldsymbol{W}_t^\top\boldsymbol{W}_{t+1},
\end{aligned}
$$

where

$$
\boldsymbol{P} = \boldsymbol{U}_t\boldsymbol{W}_{t,\perp}\boldsymbol{W}_{t,\perp}^\top\boldsymbol{W}_{t+1}\left(\boldsymbol{V}_{\boldsymbol{U}_t\boldsymbol{W}_t}^\top\boldsymbol{U}_t\boldsymbol{W}_t\boldsymbol{W}_t^\top\boldsymbol{W}_{t+1}\right)^{-1}\boldsymbol{V}_{\boldsymbol{U}_t\boldsymbol{W}_t}^\top
$$

and $\boldsymbol{V}_{\boldsymbol{U}_t\boldsymbol{W}_t}^\top\boldsymbol{U}_t\boldsymbol{W}_t\boldsymbol{W}_t^\top\boldsymbol{W}_{t+1}$ is invertible since $\boldsymbol{V}_{\boldsymbol{U}_t\boldsymbol{W}_t}^\top\boldsymbol{U}_t\boldsymbol{W}_t$ is invertible by our assumption that $\boldsymbol{V}_{\boldsymbol{X}_s}^\top\boldsymbol{U}_t$ is of full rank and $\operatorname{rank}(\boldsymbol{U}_t\boldsymbol{W}_t) \geq \operatorname{rank}\left(\boldsymbol{V}_{\boldsymbol{X}_s}^\top\boldsymbol{U}_t\boldsymbol{W}_t\right) = \operatorname{rank}\left(\boldsymbol{V}_{\boldsymbol{X}_s}^\top\boldsymbol{U}_t\right) = s$, and $\boldsymbol{W}_t^\top\boldsymbol{W}_{t+1}$ is invertible by Lemma D.6 and the assumptions on $\mu$ and $\left\|(\mathcal{A}^*\mathcal{A} - \boldsymbol{I})(\boldsymbol{X}\boldsymbol{X}^\top - \boldsymbol{U}_t\boldsymbol{U}_t^\top)\right\|$.

The key observation here is that because the (square) matrix $\boldsymbol{V}_{\boldsymbol{U}_t\boldsymbol{W}_t}^\top\boldsymbol{U}_t\boldsymbol{W}_t\boldsymbol{W}_t^\top\boldsymbol{W}_{t+1}$ is invertible, so that the column space of $\boldsymbol{U}_{t+1}\boldsymbol{W}_{t+1}$ is the same as that of $\hat{\boldsymbol{Z}}$. Following the line of proof of (Stöger & Soltanolkotabi, 2021, Lemma 9.3) (for completeness, we provide details in Lemma D.7), we deduce that

$$
\begin{aligned}
&\left\|\boldsymbol{V}_{\boldsymbol{X}_s,\perp}^\top\boldsymbol{V}_{\boldsymbol{U}_{t+1}\boldsymbol{W}_{t+1}}\right\| = \left\|\boldsymbol{V}_{\boldsymbol{X}_s,\perp}^\top\boldsymbol{V}_{\hat{\boldsymbol{Z}}}\boldsymbol{W}_{\hat{\boldsymbol{Z}}}^\top\right\| \\
&\leq \left\|\boldsymbol{V}_{\boldsymbol{X}_s,\perp}^\top\left[\left(\boldsymbol{I} + \boldsymbol{B} - \frac{1}{2}\boldsymbol{V}_{\boldsymbol{U}_t\boldsymbol{W}_t}\boldsymbol{V}_{\boldsymbol{U}_t\boldsymbol{W}_t}^\top\left(\boldsymbol{B} + \boldsymbol{B}^\top\right)\right)\boldsymbol{V}_{\boldsymbol{U}_t\boldsymbol{W}_t} - \boldsymbol{B}\boldsymbol{V}_{\boldsymbol{U}_t\boldsymbol{W}_t}\boldsymbol{V}_{\boldsymbol{U}_t\boldsymbol{W}_t}^\top\left(\boldsymbol{B} + \boldsymbol{B}^\top\right)\boldsymbol{V}_{\boldsymbol{U}_t\boldsymbol{W}_t} + \boldsymbol{D}\right]\right\| \\
&\leq \left\|\boldsymbol{V}_{\boldsymbol{X}_s,\perp}^\top\left(\boldsymbol{I} + \boldsymbol{B} - \frac{1}{2}\boldsymbol{V}_{\boldsymbol{U}_t\boldsymbol{W}_t}\boldsymbol{V}_{\boldsymbol{U}_t\boldsymbol{W}_t}^\top\left(\boldsymbol{B} + \boldsymbol{B}^\top\right)\right)\boldsymbol{V}_{\boldsymbol{U}_t\boldsymbol{W}_t}\right\| + 2\|\boldsymbol{B}\|^2 + \|\boldsymbol{D}\|
\end{aligned}
\tag{27}
$$

where $\boldsymbol{B} = (\boldsymbol{I} + \mu\boldsymbol{M}_t)(\boldsymbol{I} + \boldsymbol{P}) - \boldsymbol{I}$ and $\|\boldsymbol{D}\| \leq 100\|\boldsymbol{B}\|^2$. By assumption we have

$$
\begin{aligned}
\|\boldsymbol{P}\| &\leq \frac{\|\boldsymbol{U}_t\boldsymbol{W}_{t,\perp}\|\,\|\boldsymbol{W}_{t,\perp}\boldsymbol{W}_{t+1}\|}{\sigma_{\min}(\boldsymbol{U}_t\boldsymbol{W}_t)\sigma_{\min}(\boldsymbol{W}_t^\top\boldsymbol{W}_{t+1})} \\
&\leq 2\left\|\boldsymbol{W}_{t,\perp}\boldsymbol{W}_{t+1}\right\|,
\end{aligned}
$$

so that

$$
\begin{aligned}
&\left\|\boldsymbol{B} - \mu(\boldsymbol{X}\boldsymbol{X}^\top - \boldsymbol{U}_t\boldsymbol{U}_t^\top)\right\| \\
&\leq \mu\|\boldsymbol{M}_t - (\boldsymbol{X}\boldsymbol{X}^\top - \boldsymbol{U}_t\boldsymbol{U}_t^\top)\| + \|\boldsymbol{P}\| + \mu\|\boldsymbol{M}_t\|\|\boldsymbol{P}\| \\
&\leq \mu\left\|(\mathcal{A}^*\mathcal{A} - \boldsymbol{I})(\boldsymbol{X}\boldsymbol{X}^\top - \boldsymbol{U}_t\boldsymbol{U}_t^\top)\right\| + 2\left\|\boldsymbol{W}_{t,\perp}\boldsymbol{W}_{t+1}\right\| + 4\mu\|\boldsymbol{X}\|^2\left\|\boldsymbol{W}_{t,\perp}\boldsymbol{W}_{t+1}\right\| \\
&\leq \mu\left\|(\mathcal{A}^*\mathcal{A} - \boldsymbol{I})(\boldsymbol{X}\boldsymbol{X}^\top - \boldsymbol{U}_t\boldsymbol{U}_t^\top)\right\| + 6\left\|\boldsymbol{W}_{t,\perp}\boldsymbol{W}_{t+1}\right\| \\
&\leq 18\mu\left(10\mu\|\boldsymbol{X}\|^3 + c_4\right)c_3\|\boldsymbol{X}\| + 7\mu\left\|(\mathcal{A}^*\mathcal{A} - \boldsymbol{I})(\boldsymbol{X}\boldsymbol{X}^\top - \boldsymbol{U}_t\boldsymbol{U}_t^\top)\right\| \\
&\leq 18\mu\left(10\mu\|\boldsymbol{X}\|^3 + c_4\right)c_3\|\boldsymbol{X}\| + 0.01\mu\kappa^{-1}c_3\|\boldsymbol{X}\|^2
\end{aligned}
\tag{28}
$$

where we use Lemma D.6 to bound $\left\|\boldsymbol{W}_{t,\perp}^\top\boldsymbol{W}_{t+1}\right\|$. Let $\boldsymbol{B}_1 = \mu(\boldsymbol{X}\boldsymbol{X}^\top - \boldsymbol{U}_t\boldsymbol{U}_t^\top)$ and $\boldsymbol{R}_1 = \boldsymbol{V}_{\boldsymbol{X}_s,\perp}^\top\left(\boldsymbol{I} + \boldsymbol{B}_1 - \boldsymbol{V}_{\boldsymbol{U}_t\boldsymbol{W}_t}\boldsymbol{V}_{\boldsymbol{U}_t\boldsymbol{W}_t}^\top\boldsymbol{B}_1\right)\boldsymbol{V}_{\boldsymbol{U}_t\boldsymbol{W}_t}$, then we have

$$
\begin{aligned}
\boldsymbol{R}_1 &= \boldsymbol{V}_{\boldsymbol{X}_s,\perp}^\top\left(\boldsymbol{I} + \mu\left(\boldsymbol{I} - \boldsymbol{V}_{\boldsymbol{U}_t\boldsymbol{W}_t}\boldsymbol{V}_{\boldsymbol{U}_t\boldsymbol{W}_t}^\top\right)\left(\boldsymbol{X}\boldsymbol{X}^\top - \boldsymbol{U}_t\boldsymbol{U}_t^\top\right)\right)\boldsymbol{V}_{\boldsymbol{U}_t\boldsymbol{W}_t} \\
&= \left(\boldsymbol{I} + \mu\boldsymbol{\Sigma}_{s,\perp}^2\right)\boldsymbol{V}_{\boldsymbol{X}_s,\perp}^\top\boldsymbol{V}_{\boldsymbol{U}_t\boldsymbol{W}_t}\left(\boldsymbol{I} - \mu\boldsymbol{V}_{\boldsymbol{U}_t\boldsymbol{W}_t}^\top\boldsymbol{X}\boldsymbol{X}^\top\boldsymbol{V}_{\boldsymbol{U}_t\boldsymbol{W}_t}\right) \\
&\quad - \mu\boldsymbol{V}_{\boldsymbol{X}_s,\perp}^\top\left(\boldsymbol{I} - \boldsymbol{V}_{\boldsymbol{U}_t\boldsymbol{W}_t}\boldsymbol{V}_{\boldsymbol{U}_t\boldsymbol{W}_t}^\top\right)\boldsymbol{U}_t\boldsymbol{W}_{t,\perp}\boldsymbol{W}_{t,\perp}^\top\boldsymbol{U}_t^\top\boldsymbol{V}_{\boldsymbol{X}_s,\perp}\boldsymbol{V}_{\boldsymbol{X}_s,\perp}^\top\boldsymbol{V}_{\boldsymbol{U}_t\boldsymbol{W}_t} \\
&\quad + \mu^2\boldsymbol{\Sigma}_{s,\perp}^2\boldsymbol{V}_{\boldsymbol{X}_s,\perp}^\top\boldsymbol{V}_{\boldsymbol{U}_t\boldsymbol{W}_t}\boldsymbol{V}_{\boldsymbol{U}_t\boldsymbol{W}_t}^\top\boldsymbol{X}\boldsymbol{X}^\top\boldsymbol{V}_{\boldsymbol{U}_t\boldsymbol{W}_t}.
\end{aligned}
\tag{29}
$$

By Weyl's inequality (cf. Lemma A.2) and our assumption on $c_3$,

$$
\begin{aligned}
\sigma_{\min}\left(\boldsymbol{V}_{\boldsymbol{U}_t \boldsymbol{W}_t}^\top \boldsymbol{X} \boldsymbol{X}^\top \boldsymbol{V}_{\boldsymbol{U}_t \boldsymbol{W}_t}\right) &\geq \sigma_{\min}\left(\boldsymbol{V}_{\boldsymbol{U}_t \boldsymbol{W}_t}^\top \boldsymbol{X}_s \boldsymbol{X}_s^\top \boldsymbol{V}_{\boldsymbol{U}_t \boldsymbol{W}_t}\right) - \left\|\boldsymbol{V}_{\boldsymbol{U}_t \boldsymbol{W}_t}^\top \boldsymbol{X}_{s,\perp} \boldsymbol{X}_{s,\perp}^\top \boldsymbol{V}_{\boldsymbol{U}_t \boldsymbol{W}_t}\right\|^2 \\
&\geq \sigma_{\min}\left(\boldsymbol{V}_{\boldsymbol{U}_t \boldsymbol{W}_t}^\top \boldsymbol{X}_s \boldsymbol{X}_s^\top \boldsymbol{V}_{\boldsymbol{U}_t \boldsymbol{W}_t}\right) - \sigma_{s+1}^2 \left\|\boldsymbol{V}_{\boldsymbol{X}_s,\perp}^\top \boldsymbol{V}_{\boldsymbol{U}_t \boldsymbol{W}_t}\right\|^2 \\
&\geq \sigma_s^2 \left\|\boldsymbol{V}_{\boldsymbol{U}_t \boldsymbol{W}_t}^\top \boldsymbol{V}_{\boldsymbol{X}_s}\right\|^2 - \sigma_{s+1}^2 c_3^2 \\
&= \sigma_s^2 - (\sigma_s^2 + \sigma_{s+1}^2) c_3^2 > \frac{1}{2}\left(\sigma_s^2 + \sigma_{s+1}^2\right).
\end{aligned}
$$

So we have

$$
\|\boldsymbol{R}_1\| \leq \left(1 - \frac{\mu}{2}(\sigma_s^2 - \sigma_{s+1}^2)\right) \left\|\boldsymbol{V}_{\boldsymbol{X}_s,\perp}^\top \boldsymbol{V}_{\boldsymbol{U}_t \boldsymbol{W}_t}\right\| + 3\mu\|\boldsymbol{X}\| c_3 c_4 + \mu^2 \|\boldsymbol{X}\|^4.
$$

It thus follows from (27) that

$$
\begin{aligned}
&\left\|\boldsymbol{V}_{\boldsymbol{X}_s^\perp}^\top \boldsymbol{V}_{\boldsymbol{U}_{t+1} \boldsymbol{W}_{t+1}}\right\| \\
&\leq \|\boldsymbol{R}_1\| + 2\|\boldsymbol{B} - \boldsymbol{B}_1\| + 102\|\boldsymbol{B}\|^2 \\
&\leq \left(1 - \frac{\mu}{2}(\sigma_s^2 - \sigma_{s+1}^2)\right) \left\|\boldsymbol{V}_{\boldsymbol{X}_s,\perp}^\top \boldsymbol{V}_{\boldsymbol{U}_t \boldsymbol{W}_t}\right\| + 40\mu c_3 c_4 \|\boldsymbol{X}\| + 0.02\mu\kappa^{-1} c_3 \|\boldsymbol{X}\|^2 + 10^3 \mu^2 \|\boldsymbol{X}\|^4.
\end{aligned}
$$

Since $\left\|\boldsymbol{V}_{\boldsymbol{X}_s,\perp}^\top \boldsymbol{V}_{\boldsymbol{U}_t \boldsymbol{W}_t}\right\| \leq c_3$, it follows from our assumption on $c_3, c_4$ and $\mu$ that $\left\|\boldsymbol{V}_{\boldsymbol{X}_s^\perp}^\top \boldsymbol{V}_{\boldsymbol{U}_{t+1} \boldsymbol{W}_{t+1}}\right\| \leq c_3$ as well, which concludes the proof. $\qquad\square$

### C.3 INDUCTION

Let

$$
\widetilde{T}_\alpha^s = \min\left\{t \geq 0 : \sigma_{\min}^2\left(\boldsymbol{V}_{\boldsymbol{X}_s}^\top \boldsymbol{U}_{\alpha,t+1}\right) > 0.3\left(\sigma_s^2 - \sigma_{s+1}^2\right) =: \tau_s\right\}.
$$

In this section, we show that when $T_0 \leq t < \widetilde{T}_\alpha^s$, the parallel component grows exponentially faster than the orthogonal component. We prove this via induction and the base case is already shown in Lemma C.3.

**Lemma C.7** Let $\max\{c_3, c_4\|\boldsymbol{X}\|^{-1}\} \leq 10^{-3}\kappa^{-1}$, $\delta \leq 10^{-4}\kappa^{-1} r_*^{-\frac{1}{2}} c_3$ and $\mu \leq 10^{-4}\kappa^{-1}\|\boldsymbol{X}\|^{-2}$. Then the following holds for all $T_0 \leq t < \widetilde{T}_{\alpha,s}$ as long as $\alpha \leq \mathrm{poly}(r)^{-1}$:

$$
\sigma_{\min}\left(\boldsymbol{V}_{\boldsymbol{X}_s}^\top \boldsymbol{U}_{t+1}\right) \geq \sigma_{\min}\left(\boldsymbol{V}_{\boldsymbol{X}_s}^\top \boldsymbol{U}_{t+1} \boldsymbol{W}_t\right) \geq \left(1 + 0.5\mu\left(\sigma_s^2 + \sigma_{s+1}^2\right)\right) \sigma_{\min}\left(\boldsymbol{V}_{\boldsymbol{X}_s}^\top \boldsymbol{U}_{\alpha,t}\right) \quad (30\text{a})
$$

$$
\|\boldsymbol{U}_{t+1} \boldsymbol{W}_{t+1,\perp}\| \leq \min\left\{\left(1 + \mu\left(0.4\sigma_s^2 + 0.6\sigma_{s+1}^2\right)\right)\|\boldsymbol{U}_t \boldsymbol{W}_{t,\perp}\|, c_4\right\} \quad (30\text{b})
$$

$$
\left\|\boldsymbol{V}_{\boldsymbol{X}_s,\perp}^\top \boldsymbol{V}_{\boldsymbol{U}_{t+1} \boldsymbol{W}_{t+1}}\right\| \leq c_3. \quad (30\text{c})
$$

$$
\mathrm{rank}\left(\boldsymbol{V}_{\boldsymbol{X}_s}^\top \boldsymbol{U}_{t+1}\right) = \mathrm{rank}\left(\boldsymbol{V}_{\boldsymbol{X}_s}^\top \boldsymbol{U}_{t+1} \boldsymbol{W}_t\right) = s. \quad (30\text{d})
$$

**Proof :** The base case $t = T_0$ is already proved in (12). Now suppose that the lemma holds for $t$, we now show that it holds for $t + 1$ as well.

To begin with, we bound the term $\|\boldsymbol{\Delta}_t\|$ as follows:

$$
\begin{aligned}
\|\boldsymbol{\Delta}_t\| &= \left\|(\mathcal{A}^*\mathcal{A} - \boldsymbol{I})(\boldsymbol{X}\boldsymbol{X}^\top - \boldsymbol{U}_t \boldsymbol{U}_t^\top)\right\| \\
&\leq \left\|(\mathcal{A}^*\mathcal{A} - \boldsymbol{I})(\boldsymbol{X}\boldsymbol{X}^\top - \boldsymbol{U}_t \boldsymbol{W}_t \boldsymbol{W}_t^\top \boldsymbol{U}_t^\top)\right\| + \left\|(\mathcal{A}^*\mathcal{A} - \boldsymbol{I})\boldsymbol{U}_t \boldsymbol{W}_{t,\perp} \boldsymbol{W}_{t,\perp}^\top \boldsymbol{U}_t^\top\right\| \\
&\leq 10\delta\sqrt{r_*}\|\boldsymbol{X}\|^2 + \delta\left\|\boldsymbol{U}_t \boldsymbol{W}_{t,\perp} \boldsymbol{W}_{t,\perp}^\top \boldsymbol{U}_t^\top\right\|_* \\
&\leq 10\delta\sqrt{r_*}\|\boldsymbol{X}\|^2 + \delta\sqrt{d}\left(1 + \mu(0.4\sigma_s^2 + 0.6\sigma_{s+1}^2)\right)^{t - T_0} \|\boldsymbol{U}_{T_0} \boldsymbol{W}_{T_0,\perp}\|
\end{aligned}
$$

By induction hypothesis, it's easy to see that

$$
\frac{\sigma_{\min}\left(\boldsymbol{V}_{\boldsymbol{X}_s}^\top \boldsymbol{U}_t\right)}{\|\boldsymbol{U}_t \boldsymbol{W}_{t,\perp}\|} \geq \frac{\sigma_{\min}\left(\boldsymbol{V}_{\boldsymbol{X}_s}^\top \boldsymbol{U}_{T_0}\right)}{\|\boldsymbol{U}_{T_0} \boldsymbol{W}_{T_0,\perp}\|} \geq \mathrm{poly}(r) \cdot \alpha^{-\gamma_s} \quad (31)
$$

where

$$\gamma_s = \frac{2\left(\log\left(1 + \mu\hat{\sigma}_s^2\right) - \log\left(1 + \mu\hat{\sigma}_{s+1}^2\right)\right)}{3\log\left(1 + \mu\hat{\sigma}_1^2\right) - \log\left(1 + \mu\hat{\sigma}_{s+1}^2\right)} \geq \frac{1}{4\kappa}.$$

Since we must have $\sigma_{\min}\left(\boldsymbol{V}_{\boldsymbol{X}_s}^\top \boldsymbol{U}_t\right) \leq 0.3\tau = \mathcal{O}(1)$ by definition of $\widetilde{T}_{\alpha,s}$, it follows that $\|\boldsymbol{U}_t \boldsymbol{W}_{t,\perp}\| \leq \text{poly}(r)\alpha^{\frac{1}{4\kappa}}$, so for sufficiently small $\alpha \leq \text{poly}(r)^{-1}$, $\|\boldsymbol{\Delta}_t\| \leq 11\delta\sqrt{r}\|\boldsymbol{X}\|^2$ holds.

The above inequality combined with our assumption on $\delta$ implies that the conditions on $\|\boldsymbol{\Delta}_t\|$ in Lemmas C.4 to C.6 hold. We now show that (30a) to (30d) hold for $t + 1$, which completes the induction step.

First, since $t < \widetilde{T}_\alpha^s$, we have $\sigma_{\min}\left(\boldsymbol{V}_{\boldsymbol{X}_s}^\top \boldsymbol{U}_{t+1}\right) \leq \tau$. Moreover, the induction hypothesis implies that $\left\|\boldsymbol{V}_{\boldsymbol{X}_s,\perp}^\top \boldsymbol{V}_{\boldsymbol{U}_{t-1}\boldsymbol{W}_{t-1}}\right\| \leq c_3$ and that $\boldsymbol{V}_{\boldsymbol{X}_s}^\top \boldsymbol{U}_{\alpha,t}$ is of full rank. Thus the conditions of Corollary C.2 are all satisfied, and we deduce that (30a) holds.

Second, the assumptions on $c_3, c_4$ and $\delta$, combined with Lemma C.5, immediately implies

$$\|\boldsymbol{U}_{t+1}\boldsymbol{W}_{t+1,\perp}\| \leq \left(1 + \mu\left(0.4\sigma_s^2 + 0.6\sigma_{s+1}^2\right)\right)\|\boldsymbol{U}_t \boldsymbol{W}_{t,\perp}\|.$$

As a result, similar to (31) we observe that

$$\frac{\sigma_{\min}\left(\boldsymbol{V}_{\boldsymbol{X}_s}^\top \boldsymbol{U}_{t+1}\right)}{\|\boldsymbol{U}_{t+1}\boldsymbol{W}_{t+1,\perp}\|} \geq \frac{\sigma_{\min}\left(\boldsymbol{V}_{\boldsymbol{X}_s}^\top \boldsymbol{U}_{T_0}\right)}{\|\boldsymbol{U}_{T_0}\boldsymbol{W}_{T_0,\perp}\|} \geq \text{poly}(r) \cdot \alpha^{-\frac{1}{4\kappa}}.$$

Since $\sigma_{\min}\left(\boldsymbol{V}_{\boldsymbol{X}_s}^\top \boldsymbol{U}_{t+1}\right) \leq \|\boldsymbol{X}\|$, when $\alpha < \text{poly}(r)^{-1}$ we must have that $\|\boldsymbol{U}_{t+1}\boldsymbol{W}_{t+1,\perp}\| \leq c_4$.

Finally, Lemma C.6 implies that (30c) is true, and (30d) follows from our application of Lemma C.4. This concludes the proof. $\square$

## C.4 THE REFINEMENT PHASE AND CONCLUDING THE PROOF OF LEMMA B.1

We have shown that the parallel component $\sigma_{\min}\left(\boldsymbol{V}_{\boldsymbol{X}_s}^\top \boldsymbol{U}_{t+1}\right)$ grows exponentially faster than the orthogonal component $\|\boldsymbol{U}_t \boldsymbol{W}_{t,\perp}\|$. In this section, we characterize the GD dynamics *after* $\widetilde{T}_\alpha^s$. We begin with the following lemma, which is straightforward from the proof of Lemma C.7.

**Lemma C.8** *The following inequality holds when $\alpha \leq \text{poly}(r)^{-1}$ is sufficiently small:*

$$\left\|\boldsymbol{U}_{\widetilde{T}_s}\boldsymbol{W}_{\widetilde{T}_s,\perp}\right\| \leq \text{poly}(r) \cdot \alpha^{\frac{1}{4\kappa}}.$$

The following lemma states that in a certain time period after $\widetilde{T}_\alpha^s$, the parallel and orthogonal components still behave similarly to the second (parallel improvement) phase.

**Lemma C.9** *There exists $\tilde{t}_\alpha^s = \Theta\left(\log\frac{1}{\alpha}\right)$ when $\alpha \to 0$ (here we omit the dependence of $\widetilde{T}_\alpha^s$ on $\alpha$ for simplicity) such that when $0 \leq t - \widetilde{T}_\alpha^s \leq \tilde{t}_\alpha^s$, we have*

$$\sigma_{\min}\left(\boldsymbol{V}_{\boldsymbol{X}_s}^\top \boldsymbol{U}_t\right) \geq \sigma_{\min}\left(\boldsymbol{U}_t \boldsymbol{W}_t\right) \geq 0.3\tau \tag{32a}$$

$$\|\boldsymbol{U}_t \boldsymbol{W}_t\| \leq \left(1 + \mu(0.4\sigma_s^2 + 0.6\sigma_{s+1}^2)\right)^{t - \widetilde{T}_\alpha^s}\left\|\boldsymbol{U}_{\widetilde{T}_\alpha^s}\boldsymbol{W}_{\widetilde{T}_\alpha^s}\right\| \tag{32b}$$

$$\|\boldsymbol{V}_{\boldsymbol{X}_s,\perp}\boldsymbol{V}_{\boldsymbol{U}_t \boldsymbol{W}_t}\| \leq c_3. \tag{32c}$$

***Proof*** : We choose $\tilde{t}_\alpha^s = \min\left\{t \geq 0 : \|\boldsymbol{U}_{t+1}\boldsymbol{W}_{t+1,\perp}\|^2 \leq c_5\right\}$ where

$$c_5 = 10^{-4}d^{-\frac{1}{2}}\kappa^{-1}c_3\|\boldsymbol{X}\|^2 \tag{33}$$

We prove (32) by induction. The proof follows the idea of Lemma C.7, except that we need to bound $\|\boldsymbol{\Delta}_t\|$ in each induction step. Concretely, suppose that (32) holds at time $t$, then

$$\begin{aligned}
\|\boldsymbol{\Delta}_t\| &= \left\|(\mathcal{A}^*\mathcal{A} - \boldsymbol{I})(\boldsymbol{X}\boldsymbol{X}^\top - \boldsymbol{U}_t \boldsymbol{U}_t^\top)\right\| \\
&\leq \left\|(\mathcal{A}^*\mathcal{A} - \boldsymbol{I})(\boldsymbol{X}\boldsymbol{X}^\top - \boldsymbol{U}_t \boldsymbol{W}_t \boldsymbol{W}_t^\top \boldsymbol{U}_t^\top)\right\| + \left\|(\mathcal{A}^*\mathcal{A} - \boldsymbol{I})\boldsymbol{U}_t \boldsymbol{W}_{t,\perp}\boldsymbol{W}_{t,\perp}^\top \boldsymbol{U}_t^\top\right\| \\
&\leq 10\delta\sqrt{r_*}\|\boldsymbol{X}\|^2 + \delta\left\|\boldsymbol{U}_t \boldsymbol{W}_{t,\perp}\boldsymbol{W}_{t,\perp}^\top \boldsymbol{U}_t^\top\right\|_* \\
&\leq 10\delta\sqrt{r_*}\|\boldsymbol{X}\|^2 + \delta c_5\sqrt{d} \leq 10^{-3}\kappa^{-1}c_3\|\boldsymbol{X}\|^2
\end{aligned} \tag{34}$$

where we used the definition of $c_5$ in the last step. As a result, we can apply the conclusion of Lemmas C.4 to C.6 which implies that (32) holds for $t + 1$. Finally, combining Lemma C.8 and (32b) yields $\widetilde{T}_\alpha^s = \omega(1)$. $\qquad\square$

We now present the main result of this section:

**Lemma C.10** *Suppose that* $0 \leq t - \widetilde{T}_\alpha^s \leq \widetilde{t}_\alpha^s$, $\left\|\boldsymbol{V}_{\boldsymbol{X}_s,\perp}^\top \boldsymbol{V}_{\boldsymbol{U}_t \boldsymbol{W}_t}\right\| \leq c_3$ *and the conditions on* $c_3, c_4, \delta$ *and* $\mu$ *in Lemma C.7 hold, then we have*

$$\left\|\boldsymbol{V}_{\boldsymbol{X}_s}^\top (\boldsymbol{X}\boldsymbol{X}^\top - \boldsymbol{U}_{t+1}\boldsymbol{U}_{t+1}^\top)\right\|_F$$
$$\leq \left(1 - \frac{1}{2}\mu\tau\right) \left\|\boldsymbol{V}_{\boldsymbol{X}_s}^\top \left(\boldsymbol{X}\boldsymbol{X}^\top - \boldsymbol{U}_t\boldsymbol{U}_t^\top\right)\right\|_F + 20\mu\|\boldsymbol{X}\|^4 (\delta + 5c_3) + 2000\mu^2\sqrt{r}\|\boldsymbol{X}\|^6.$$

*where we recall that* $\tau = \min_{1 \leq s \leq \hat{r} \wedge r_*} \sigma_s^2 - \sigma_{s+1}^2 > 0$.

**Proof** : The update of GD implies that

$$\boldsymbol{X}\boldsymbol{X}^\top - \boldsymbol{U}_{t+1}\boldsymbol{U}_{t+1}^\top$$
$$= \boldsymbol{X}\boldsymbol{X}^\top - (\boldsymbol{I} + \mu\boldsymbol{M}_t)\boldsymbol{U}_t\boldsymbol{U}_t^\top (\boldsymbol{I} + \mu\boldsymbol{M}_t)$$
$$= \underbrace{\left(\boldsymbol{I} - \mu\boldsymbol{U}_t\boldsymbol{U}_t^\top\right) \left(\boldsymbol{X}\boldsymbol{X}^\top - \boldsymbol{U}_t\boldsymbol{U}_t^\top\right) \left(\boldsymbol{I} - \mu\boldsymbol{U}_t\boldsymbol{U}_t^\top\right)}_{=(i)} + \mu \underbrace{\boldsymbol{\Delta}_t\boldsymbol{U}_t\boldsymbol{U}_t^\top}_{=(ii)}$$
$$+ \underbrace{\mu\boldsymbol{U}_t\boldsymbol{U}_t^\top \boldsymbol{\Delta}_t}_{=(iii)} + \mu^2 \left(\mathcal{E}_{t,1} + \mathcal{E}_{t,2}\right),$$

where

$$\left\|\boldsymbol{V}_{\boldsymbol{X}_s}^\top \mathcal{E}_{t,1}\right\|_F = \left\|\boldsymbol{V}_{\boldsymbol{X}_s}^\top \boldsymbol{U}_t\boldsymbol{U}_t \left(\boldsymbol{X}\boldsymbol{X}^\top - \boldsymbol{U}_t\boldsymbol{U}_t^\top\right) \boldsymbol{U}_t\boldsymbol{U}_t^\top\right\|_F$$
$$\leq \sqrt{r_*} \left\|\boldsymbol{V}_{\boldsymbol{X}_s}^\top \boldsymbol{U}_t\boldsymbol{U}_t \left(\boldsymbol{X}\boldsymbol{X}^\top - \boldsymbol{U}_t\boldsymbol{U}_t^\top\right) \boldsymbol{U}_t\boldsymbol{U}_t^\top\right\|_2$$
$$\leq 10^3\sqrt{r_*}\|\boldsymbol{X}\|^2$$

and

$$\left\|\boldsymbol{V}_{\boldsymbol{X}_s}^\top \mathcal{E}_{t,2}\right\|_F = \left\|\boldsymbol{V}_{\boldsymbol{X}_s}^\top \left[(\mathcal{A}^*\mathcal{A}) \left(\boldsymbol{X}\boldsymbol{X}^\top - \boldsymbol{U}_t\boldsymbol{U}_t^\top\right)\right] \boldsymbol{U}_t\boldsymbol{U}_t^\top \left[(\mathcal{A}^*\mathcal{A}) \left(\boldsymbol{X}\boldsymbol{X}^\top - \boldsymbol{U}_t\boldsymbol{U}_t^\top\right)\right]\right\|_F$$
$$\leq \sqrt{r_*} \left\|\left[(\mathcal{A}^*\mathcal{A}) \left(\boldsymbol{X}\boldsymbol{X}^\top - \boldsymbol{U}_t\boldsymbol{U}_t^\top\right)\right] \boldsymbol{U}_t\boldsymbol{U}_t^\top \left[(\mathcal{A}^*\mathcal{A}) \left(\boldsymbol{X}\boldsymbol{X}^\top - \boldsymbol{U}_t\boldsymbol{U}_t^\top\right)\right]\right\|$$
$$\leq 10^3\sqrt{r_*}\|\boldsymbol{X}\|^2.$$

Note that we would like to bound $\left\|\boldsymbol{V}_{\boldsymbol{X}_s}^\top \left(\boldsymbol{X}\boldsymbol{X}^\top - \boldsymbol{U}_{t+1}\boldsymbol{U}_{t+1}^\top\right)\right\|$. We deal with the above three terms separately. For the first term, we have

$$\left\|\boldsymbol{V}_{\boldsymbol{X}_s}^\top \left(\boldsymbol{I} - \mu\boldsymbol{U}_t\boldsymbol{U}_t^\top\right) \left(\boldsymbol{X}\boldsymbol{X}^\top - \boldsymbol{U}_t\boldsymbol{U}_t^\top\right) \left(\boldsymbol{I} - \mu\boldsymbol{U}_t\boldsymbol{U}_t\right)\right\|_F$$
$$= \left\|\boldsymbol{V}_{\boldsymbol{X}_s}^\top \left(\boldsymbol{I} - \mu\boldsymbol{U}_t\boldsymbol{U}_t^\top\right) \boldsymbol{V}_{\boldsymbol{X}_s}\boldsymbol{V}_{\boldsymbol{X}_s}^\top \left(\boldsymbol{X}\boldsymbol{X}^\top - \boldsymbol{U}_t\boldsymbol{U}_t^\top\right) \left(\boldsymbol{I} - \mu\boldsymbol{U}_t\boldsymbol{U}_t^\top\right)\right\|_F$$
$$+ \left\|\boldsymbol{V}_{\boldsymbol{X}_s}^\top \left(\boldsymbol{I} - \mu\boldsymbol{U}_t\boldsymbol{U}_t^\top\right) \boldsymbol{V}_{\boldsymbol{X}_s,\perp}\boldsymbol{V}_{\boldsymbol{X}_s,\perp}^\top \left(\boldsymbol{X}\boldsymbol{X}^\top - \boldsymbol{U}_t\boldsymbol{U}_t^\top\right) \left(\boldsymbol{I} - \mu\boldsymbol{U}_t\boldsymbol{U}_t^\top\right)\right\|_F$$
$$\leq \left\|\boldsymbol{I} - \mu\boldsymbol{V}_{\boldsymbol{X}_s}^\top \boldsymbol{U}_t\boldsymbol{U}_t^\top \boldsymbol{V}_{\boldsymbol{X}_s}\right\| \left\|\boldsymbol{V}_{\boldsymbol{X}_s}^\top \left(\boldsymbol{X}\boldsymbol{X}^\top - \boldsymbol{U}_t\boldsymbol{U}_t^\top\right)\right\| + \mu \left\|\boldsymbol{V}_{\boldsymbol{X}_s}^\top \boldsymbol{U}_t\boldsymbol{U}_t^\top \boldsymbol{V}_{\boldsymbol{X}_s,\perp}\boldsymbol{V}_{\boldsymbol{X}_s,\perp}^\top \left(\boldsymbol{X}\boldsymbol{X}^\top - \boldsymbol{U}_t\boldsymbol{U}_t^\top\right)\right\|_F$$

$$\tag{35a}$$

$$\leq \left(1 - \mu\sigma_{\min}^2(\boldsymbol{U}_t\boldsymbol{W}_t)\sigma_{\min}^2\left(\boldsymbol{V}_{\boldsymbol{X}_s}^\top \boldsymbol{V}_{\boldsymbol{U}_t\boldsymbol{W}_t}\right)\right) \left\|\boldsymbol{V}_{\boldsymbol{X}_s}^\top \left(\boldsymbol{X}\boldsymbol{X}^\top - \boldsymbol{U}_t\boldsymbol{U}_t^\top\right)\right\|_F + 100\mu\|\boldsymbol{X}\|^4 c_3 \quad \text{(35b)}$$

$$\leq \left(1 - \frac{1}{2}\mu\tau\right) \left\|\boldsymbol{V}_{\boldsymbol{X}_s}^\top \left(\boldsymbol{X}\boldsymbol{X}^\top - \boldsymbol{U}_t\boldsymbol{U}_t^\top\right)\right\|_F + 100\mu\|\boldsymbol{X}\|^4 c_3, \quad \text{(35c)}$$

where in (35a) we use $\left\|\boldsymbol{I} - \mu\boldsymbol{U}_t\boldsymbol{U}_t^\top\right\| \leq 1$, (35b) follows from

$$\sigma_{\min}\left(\boldsymbol{V}_{\boldsymbol{X}_s}^\top \boldsymbol{U}_t\boldsymbol{U}_t^\top \boldsymbol{V}_{\boldsymbol{X}_s}\right) = \sigma_{\min}\left(\boldsymbol{V}_{\boldsymbol{X}_s}^\top \boldsymbol{U}_t\boldsymbol{W}_t\boldsymbol{W}_t^\top \boldsymbol{U}_t^\top \boldsymbol{V}_{\boldsymbol{X}_s}\right) \geq \sigma_{\min}\left(\boldsymbol{V}_{\boldsymbol{X}_s}^\top \boldsymbol{U}_t\boldsymbol{W}_t\right)^2$$
$$\geq \sigma_{\min}^2(\boldsymbol{U}_t\boldsymbol{W}_t)\sigma_{\min}^2\left(\boldsymbol{V}_{\boldsymbol{X}_s}^\top \boldsymbol{V}_{\boldsymbol{U}_t\boldsymbol{W}_t}\right)$$

and

$$\left\|\boldsymbol{V}_{\boldsymbol{X}_s}^\top \boldsymbol{U}_t\boldsymbol{U}_t^\top \boldsymbol{V}_{\boldsymbol{X}_s,\perp}\right\| = \left\|\boldsymbol{V}_{\boldsymbol{X}_s}^\top \boldsymbol{U}_t\boldsymbol{W}_t\boldsymbol{W}_t^\top \boldsymbol{U}_t^\top \boldsymbol{V}_{\boldsymbol{X}_s,\perp}\right\| \leq \|\boldsymbol{U}_t\|^2 \left\|\boldsymbol{V}_{\boldsymbol{X}_s,\perp}^\top \boldsymbol{U}_t\boldsymbol{W}_t\right\| \leq c_3\|\boldsymbol{U}_t\|^2,$$

and lastly (35c) is obtained from

$$\sigma_{\min}^2\left(\boldsymbol{V}_{\boldsymbol{X}_s}^\top \boldsymbol{V}_{\boldsymbol{U}_t \boldsymbol{W}_t}\right) \geq 1 - \left\|\boldsymbol{V}_{\boldsymbol{X}_s,\perp}^\top \boldsymbol{V}_{\boldsymbol{U}_t \boldsymbol{W}_t}\right\|^2 \geq 1 - c_3^2.$$

For the second and the third terms, we have

$$\left\|\left[\left(\boldsymbol{I} - \mathcal{A}^*\mathcal{A}\right)\left(\boldsymbol{X}\boldsymbol{X}^\top - \boldsymbol{U}_t\boldsymbol{U}_t\right)\right]\boldsymbol{U}_t\boldsymbol{U}_t^\top + \boldsymbol{U}_t\boldsymbol{U}_t^\top\left[\left(\boldsymbol{I} - \mathcal{A}^*\mathcal{A}\right)\left(\boldsymbol{X}\boldsymbol{X}^\top - \boldsymbol{U}_t\boldsymbol{U}_t^\top\right)\right]\right\| \\ \leq 0.1\kappa^{-1}c_3\|\boldsymbol{X}\|^4 \tag{36}$$

where we use the estimate in (34). Combining (35) and (36) yields

$$\left\|\boldsymbol{V}_{\boldsymbol{X}_s}^\top(\boldsymbol{X}\boldsymbol{X}^\top - \boldsymbol{U}_{t+1}\boldsymbol{U}_{t+1}^\top)\right\| \\ \leq \left(1 - \frac{1}{2}\mu\tau\right)\left\|\boldsymbol{V}_{\boldsymbol{X}_s}^\top\left(\boldsymbol{X}\boldsymbol{X}^\top - \boldsymbol{U}_t\boldsymbol{U}_t^\top\right)\right\| + 200\mu\|\boldsymbol{X}\|^4 c_3 + 110\mu^2\sqrt{r_*}\|\boldsymbol{X}\|^6.$$

$\square$

To apply the result of Lemma C.10, we need to verify that $\|\boldsymbol{V}_{\boldsymbol{X}_s,\perp}\boldsymbol{V}_{\boldsymbol{U}_t\boldsymbol{W}_t}\| \leq c_3$ still holds when $t \geq \widetilde{T}_\alpha^s$. In fact, this is true as long as $t - \widetilde{T}_\alpha^s \leq \mathcal{O}\left(\log\frac{1}{\alpha}\right)$.

We are now ready to present our first main result, which states that with small initialization, GD would visit the $\mathcal{O}(\delta)$-neighbourhood of the rank-$s$ minimizer of the full observation loss i.e. $\boldsymbol{X}_s\boldsymbol{X}_s^\top$.

**Theorem C.1** When $\alpha < \text{poly}(r_*)^{-1}$ and $\delta = 10^{-4}\kappa^{-1}c_3$ with $c_3 < 10^{-3}\kappa^{-1}r_*^{-\frac{1}{2}}$, there exists a time $t = \hat{T}_\alpha^s \in \mathbb{Z}_+$ such that

$$\|\boldsymbol{X}_s\boldsymbol{X}_s^\top - \boldsymbol{U}_t\boldsymbol{U}_t^\top\| \leq 10^3\tau^{-2}\|\boldsymbol{X}\|^6\sqrt{r_*}c_3.$$

*Proof* : First, observe that for all $t \geq 0$,

$$\left\|\boldsymbol{X}_s\boldsymbol{X}_s^\top - \boldsymbol{U}_t\boldsymbol{U}_t^\top\right\|_F \leq \left\|\left(\boldsymbol{X}_s\boldsymbol{X}_s^\top - \boldsymbol{U}_t\boldsymbol{U}_t^\top\right)\boldsymbol{V}_{\boldsymbol{X}_s}\boldsymbol{V}_{\boldsymbol{X}_s}^\top\right\|_F + \left\|\boldsymbol{U}_t\boldsymbol{U}_t^\top\boldsymbol{V}_{\boldsymbol{X}_s^\perp}\boldsymbol{V}_{\boldsymbol{X}_s^\perp}^\top\right\|_F \\ \leq \left\|\left(\boldsymbol{X}_s\boldsymbol{X}_s^\top - \boldsymbol{U}_t\boldsymbol{U}_t^\top\right)\boldsymbol{V}_{\boldsymbol{X}_s}\boldsymbol{V}_{\boldsymbol{X}_s}^\top\right\|_F + \left\|\boldsymbol{V}_{\boldsymbol{X}_s^\perp}^\top\boldsymbol{U}_t\boldsymbol{U}_t^\top\boldsymbol{V}_{\boldsymbol{X}_s^\perp}\right\|_F \\ \leq \left\|\boldsymbol{V}_{\boldsymbol{X}_s}^\top\left(\boldsymbol{X}_s\boldsymbol{X}_s^\top - \boldsymbol{U}_t\boldsymbol{U}_t^\top\right)\right\|_F + \sqrt{r_*}\left\|\boldsymbol{V}_{\boldsymbol{X}_s^\perp}^\top\boldsymbol{U}_t\boldsymbol{W}_t\right\|^2 + \sqrt{d}\left\|\boldsymbol{V}_{\boldsymbol{X}_s^\perp}^\top\boldsymbol{U}_t\boldsymbol{W}_{t,\perp}\right\|^2 \\ \leq \left\|\boldsymbol{V}_{\boldsymbol{X}_s}^\top\left(\boldsymbol{X}_s\boldsymbol{X}_s^\top - \boldsymbol{U}_t\boldsymbol{U}_t^\top\right)\right\| + 9\sqrt{r_*}\|\boldsymbol{X}\|^2\left\|\boldsymbol{V}_{\boldsymbol{X}_s,\perp}\boldsymbol{V}_{\boldsymbol{U}_t\boldsymbol{W}_t}\right\|^2 + \sqrt{d}\|\boldsymbol{U}_t\boldsymbol{W}_{t,\perp}\|^2. \tag{37}$$

We set $\delta = 10^{-3}\|\boldsymbol{X}\|^{-2}\tau c_3$ and

$$\hat{T}_\alpha^s = \widetilde{T}_\alpha^s - \frac{\log\left(10^{-2}\|\boldsymbol{X}\|^{-2}\tau c_3^{-1}\right)}{\log\left(1 - \frac{1}{2}\mu\tau\right)}, \tag{38}$$

then for small $\alpha$ we have $\hat{T}_\alpha^s \leq \widetilde{T}_\alpha^s + \widetilde{T}_\alpha^s$ (defined in Lemma C.9). Hence for $\widetilde{T}_\alpha^s \leq t < \hat{T}_\alpha^s$ we always have $\|\boldsymbol{V}_{\boldsymbol{X}_s,\perp}\boldsymbol{V}_{\boldsymbol{U}_t\boldsymbol{W}_t}\| \leq c_3$. By Lemma C.10 and the choice of $c_3$ and $\delta$, we have for $\widetilde{T}_\alpha^s \leq t < T_s$ that

$$\left\|\boldsymbol{V}_{\boldsymbol{X}_s}^\top\left(\boldsymbol{X}\boldsymbol{X}^\top - \boldsymbol{U}_{t+1}\boldsymbol{U}_{t+1}^\top\right)\right\|_F \leq \left(1 - \frac{1}{2}\mu\tau_s\right)\left\|\boldsymbol{V}_{\boldsymbol{X}_s}^\top\left(\boldsymbol{X}\boldsymbol{X}^\top - \boldsymbol{U}_t\boldsymbol{U}_t^\top\right)\right\|_F + 150\mu\|\boldsymbol{X}\|^4\sqrt{r_*}c_3$$

which implies that

$$\left\|\boldsymbol{V}_{\boldsymbol{X}_s}^\top\left(\boldsymbol{X}\boldsymbol{X}^\top - \boldsymbol{U}_{T_s}\boldsymbol{U}_{T_s}^\top\right)\right\|_F \leq 400\tau^{-1}\|\boldsymbol{X}\|^4\sqrt{r_*}c_3.$$

Meanwhile, by Lemma C.9 we have $\|\boldsymbol{U}_t\boldsymbol{W}_{t,\perp}\| \leq c_5$ and $\left\|\boldsymbol{V}_{\boldsymbol{X}_s,\perp}^\top\boldsymbol{V}_{\boldsymbol{U}_t\boldsymbol{W}_t}\right\| \leq c_3$ at $t = \hat{T}_\alpha^s$. Plugging into (37) yields

$$\left\|\boldsymbol{X}_s\boldsymbol{X}_s^\top - \boldsymbol{U}_{\hat{T}_\alpha^s}\boldsymbol{U}_{\hat{T}_\alpha^s}^\top\right\|_F \leq 400\tau^{-1}\|\boldsymbol{X}\|^4\sqrt{r_*}c_3 + 9\|\boldsymbol{X}\|^2 c_3^2\sqrt{r_*} + c_5^2\sqrt{d}.$$

By definition of $c_3$ and $c_5$ we deduce that $\left\|\boldsymbol{X}_s\boldsymbol{X}_s^\top - \boldsymbol{U}_{\hat{T}_\alpha^s}\boldsymbol{U}_{\hat{T}_\alpha^s}^\top\right\|_F \leq 10^3\tau^{-2}\|\boldsymbol{X}\|^6\sqrt{r_*}c_3$, as desired. $\square$

**Corollary C.4** *There exists a constant $C_1 = C_1(\kappa, r_*)$ such that*

$$\max_{0 \leq t \leq \hat{T}_\alpha^s} \|U_t W_{t,\perp}\| \leq C_1 \alpha^{\frac{1}{4\kappa}}.$$

**Proof** : The case of $\widetilde{T}_\alpha^s$ directly follows from Lemma C.8. For $t > \widetilde{T}_\alpha^s$, we know from Lemma C.9 that

$$\|U_t W_{t,\perp}\| \leq \left\|U_{\widetilde{T}_\alpha^s} W_{\widetilde{T}_\alpha^s,\perp}\right\| \cdot \left(1 + \mu \sigma_s^2\right)^{\hat{T}_\alpha^s - \widetilde{T}_\alpha^s}.$$

By (38), the second term is a constant independent of $\alpha$, so the conclusion follows. □

## D   AUXILIARY RESULTS FOR PROVING LEMMA B.1

This section contains a collection of auxiliary results that are used in the previous section.

### D.1   THE SPECTRAL PHASE

In the section, we provide auxiliary results for the analysis in the spectral phase.

Recall that $N_t = (I + \mu M)^t$ and $U_t = \widetilde{U}_t + E_t = N_t U_0 + E_t$ and $U_0 = \alpha U$. Also recall that $M = \sum_{i=1}^{\text{rank}(M)} \hat{\lambda}_i \hat{v}_i \hat{v}_i^\top$; we additionally define $M_s = \sum_{i=1}^{\min\{s, \text{rank}(M)\}} \hat{\lambda}_i \hat{v}_i \hat{v}_i^\top$. Similarly, let $L_t$ be the span of the top-$s$ left singular vectors of $U_t$. The following lemma shows that power iteration would result in large eigengap of $U_t$.

**Lemma D.1** *Let $\rho = \sigma_{\min}\left(V_{M_s}^\top U\right) > 0$, then the following three inequalities hold, given that the denominator of the third is positive.*

$$\sigma_s(U_t) \geq \alpha \left(\rho \sigma_s\left(\hat{Z}_t\right) - \sigma_{s+1}\left(\hat{Z}_t\right) \|U\|\right) - \|E_t\|, \tag{39a}$$

$$\sigma_{s+1}(U_t) \leq \alpha \sigma_{s+1}\left(\hat{Z}_t\right) \|U\| + \|E_t\|, \tag{39b}$$

$$\left\|V_{M_s^\perp}^\top V_{L_t}\right\| \leq \frac{\alpha \sigma_{s+1}\left(\hat{Z}_t\right) \|U\| + \|E_t\|}{\alpha \rho \sigma_s\left(\hat{Z}_t\right) - 2\left(\alpha \sigma_{s+1}\left(\hat{Z}_t\right) \|U\| + \|E_t\|\right)}. \tag{39c}$$

**Proof** : By Weyl's inequality we have

$$\begin{aligned}
\sigma_{s+1}(U_t) &= \sigma_{s+1}\left((1 + \mu M)^t U_0\right) + \|E_t\| \\
&= \alpha \sigma_{s+1}\left((1 + \mu M)^t U\right) + \|E_t\| \\
&\leq \alpha \sigma_{s+1}\left((1 + \mu M_s)^t U\right) + \alpha \left\|\left[(1 + \mu M)^t - (1 + \mu M_s)^t\right] U\right\| + \|E_t\| \\
&\leq \alpha (1 + \mu \hat{\lambda}_{s+1})^t \|U\| + \|E_t\|.
\end{aligned}$$

Thus (39b) holds. Similarly,

$$\begin{aligned}
\sigma_s(U_t) &\geq \alpha \sigma_s\left(N_t V_{M_s} V_{M_s}^\top U\right) - \alpha(1 + \mu \hat{\lambda}_{s+1})^t \|U\| - \|E_t\| \\
&\geq \alpha \sigma_s\left(N_t V_{M_s}\right) \sigma_{\min}\left(V_{M_s}^\top U\right) - \alpha(1 + \mu \hat{\lambda}_{s+1})^t \|U\| - \|E_t\| \\
&\geq \alpha \rho(1 + \mu \hat{\lambda}_s)^t - \alpha(1 + \mu \hat{\lambda}_{s+1})^t \|U\| - \|E_t\|.
\end{aligned}$$

Finally, note that we can write

$$\alpha(1 + \mu M_s)^t U = V_{M_s} \underbrace{(1 + \mu \Sigma_{M_s})^t V_{M_s}^\top U}_{\text{invertible}},$$

so that the subspace spanned by the left singular vectors of $\alpha(1 + \mu M_s)^t U$ coincides with the column span of $V_{M_s}$. Since $L_t$ is the span of top-$s$ left singular vectors of $U_t$, we apply Wedin's sin theorem (Wedin, 1972) and deduce (39c). □

The next lemma relates the quantities studied in Lemma D.1 with those that are needed in the induction. The proof is the same as (Stöger & Soltanolkotabi, 2021, Lemma 8.4), so we omit it here.

**Lemma D.2** *Suppose that* $\left\|V_{\boldsymbol{X}_s,\perp}^\top V_{L_t}\right\| \leq 0.1$ *for some* $t \geq 1$. *Then it holds that*

$$\sigma_s\left(\boldsymbol{U}_t\boldsymbol{W}_t\right) \geq \frac{1}{2}\sigma_s\left(\boldsymbol{U}_t\right), \tag{40a}$$

$$\left\|V_{\boldsymbol{X}_s,\perp}^\top V_{\boldsymbol{U}_t\boldsymbol{W}_t}\right\| \leq 10\left\|V_{\boldsymbol{X}_s,\perp}^\top V_{L_t}\right\|, \tag{40b}$$

$$\left\|\boldsymbol{U}_t\boldsymbol{W}_{t,\perp}\right\| \leq 2\sigma_{s+1}\left(\boldsymbol{U}_t\right). \tag{40c}$$

Combining the above two lemmas, we directly obtain the following corollary:

**Corollary D.1** *Suppose that* $\alpha\sigma_s(\boldsymbol{N}_t) > 10\left(\alpha\sigma_{s+1}(\boldsymbol{N}_t) + \|\boldsymbol{E}_t\|\right)$, *then we have that*

$$\sigma_s\left(\boldsymbol{U}_t\boldsymbol{W}_t\right) \geq 0.4\alpha\sigma_{r_\star}\left(\hat{\boldsymbol{Z}}_t\right)\sigma_{\min}\left(\boldsymbol{V}_L^\top\boldsymbol{U}\right)$$

$$\left\|\boldsymbol{U}_t\boldsymbol{W}_{t,\perp}\right\| \leq 2\left(\alpha\sigma_{s+1}\left(\hat{\boldsymbol{Z}}_t\right)\|\boldsymbol{U}\| + \|\boldsymbol{E}_t\|\right) \tag{41}$$

$$\left\|V_{\boldsymbol{X}_s,\perp}^\top V_{\boldsymbol{U}_t\boldsymbol{W}_t}\right\| \leq 100\left(\delta + \frac{\alpha\sigma_{s+1}\left(\boldsymbol{Z}_t\right)\|\boldsymbol{U}\| + \|\boldsymbol{E}_t\|}{\alpha\rho\sigma_s\left(\boldsymbol{Z}_t\right)}\right)$$

### D.2 The parallel improvement phase

In the section, we provide auxiliary results for the analysis in the parallel improvement phase.

**Lemma D.3** *(Stöger & Soltanolkotabi, 2021, Lemma 9.4) For sufficiently small* $\mu$ *and* $\delta$, *suppose that* $\|\boldsymbol{U}_t\| \leq 3\|\boldsymbol{X}\|$, *then we also have* $\|\boldsymbol{U}_{t+1}\| \leq 3\|\boldsymbol{X}\|$.

**Lemma D.4** *Under the assumptions in Lemma C.5, we have*

$$\left\|V_{\boldsymbol{X}_s,\perp}^\top V_{\boldsymbol{U}_{t+1}\boldsymbol{W}_t}\right\| \leq 2\left(c_3 + 10\mu\|\boldsymbol{X}\|^2\right) \leq 0.01.$$

***Proof*** : The proof of this lemma is essentially the same as (Stöger & Soltanolkotabi, 2021, Lemma B.1), and we omit it here. $\square$

**Lemma D.5** *Under the assumptions in Lemma C.6, we have*

$$\sigma_{\min}\left(V_{\boldsymbol{X}_s}^\top\boldsymbol{U}_{t+1}\right) \geq \frac{1}{2}\sigma_{\min}(\boldsymbol{U}_t\boldsymbol{W}_t).$$

***Proof*** : We have

$$\sigma_{\min}\left(V_{\boldsymbol{X}_s}^\top\boldsymbol{U}_{t+1}\right) \geq \sigma_{\min}\left(V_{\boldsymbol{X}_s}^\top\boldsymbol{U}_{t+1}\boldsymbol{W}_t\right)$$
$$= \sigma_{\min}\left(V_{\boldsymbol{X}_s}^\top\left(\boldsymbol{I} + \mu\boldsymbol{M}_t\right)\boldsymbol{U}_t\boldsymbol{W}_t\right)$$
$$\geq \sigma_{\min}\left(V_{\boldsymbol{X}_s}^\top\left(\boldsymbol{I} + \mu\boldsymbol{M}_t\right)V_{\boldsymbol{U}_t\boldsymbol{W}_t}\right)\cdot\sigma_{\min}\left(V_{\boldsymbol{U}_t\boldsymbol{W}_t}^\top\boldsymbol{U}_t\boldsymbol{W}_t\right)$$
$$\geq \left[\sigma_{\min}\left(V_{\boldsymbol{X}_s}^\top V_{\boldsymbol{U}_t\boldsymbol{W}_t}\right) - \mu\|\boldsymbol{M}_t\|\right]\cdot\sigma_{\min}(\boldsymbol{U}_t\boldsymbol{W}_t)$$
$$\geq \left(\sqrt{1-c_3^2} - 10\mu\|\boldsymbol{X}\|^2\right)\sigma_{\min}(\boldsymbol{U}_t\boldsymbol{W}_t) \geq \frac{1}{2}\sigma_{\min}(\boldsymbol{U}_t\boldsymbol{W}_t)$$

where the last step follows from

$$\sigma_{\min}\left(V_{\boldsymbol{X}_s}^\top V_{\boldsymbol{U}_t\boldsymbol{W}_t}\right)^2 \geq 1 - \left\|V_{\boldsymbol{X}_s,\perp}^\top V_{\boldsymbol{U}_t\boldsymbol{W}_t}\right\|^2 \geq 1 - c_3^2.$$

The conclusion follows. $\square$

**Lemma D.6** *Under the assumptions in Lemma C.6, we have*

$$\left\|\boldsymbol{W}_{t,\perp}^\top\boldsymbol{W}_{t+1}\right\| \leq 3\mu\left(10\mu\|\boldsymbol{X}\|^2 + c_4\right)c_3\|\boldsymbol{X}\| + \mu\left\|(\mathcal{A}^*\mathcal{A}-\boldsymbol{I})(\boldsymbol{X}\boldsymbol{X}^\top - \boldsymbol{U}_t\boldsymbol{U}_t^\top)\right\|.$$

***Proof*** : The proof roughly follows (Stöger & Soltanolkotabi, 2021, Lemma B.3), but we include it here for completeness.

Since $V_{X_s}^\top U_{t+1} = V_{t+1}\Sigma_{t+1}W_{t+1}$ and $V_{t+1}\Sigma_{t+1} \in \mathbb{R}^{s \times s}$ is invertible, we have

$$\left\| W_{t,\perp}^\top W_{t+1} \right\| = \left\| W_{t,\perp}^\top U_{t+1}^\top V_{X_s} \left( V_{X_s}^\top U_{t+1} U_{t+1}^\top V_{X_s} \right)^{-\frac{1}{2}} \right\|.$$

Since

$$
\begin{aligned}
&V_{X_s}^\top U_{t+1} W_{t,\perp} \\
&= V_{X_s}^\top \left( I + \mu \mathcal{A}^* \mathcal{A}(XX^\top - U_t U_t^\top) \right) U_t W_{t,\perp} \\
&= V_{X_s}^\top \left( I + \mu(XX^\top - U_t U_t^\top) \right) U_t W_{t,\perp} + \mu V_{X_s}^\top \Delta_t U_t W_{t,\perp} \\
&= -\mu V_{X_s}^\top U_t U_t^\top U_t W_{t,\perp} + \mu V_{X_s}^\top \Delta_t U_t W_{t,\perp} & (42a) \\
&= -\mu V_{X_s}^\top U_t W_t W_t^\top U_t^\top U_t W_{t,\perp} + \mu V_{X_s}^\top \Delta_t U_t W_{t,\perp} & (42b) \\
&= -\mu \underbrace{V_{X_s}^\top U_t W_t W_t^\top U_t^\top V_{X_s,\perp} V_{X_s,\perp}^\top U_t W_{t,\perp}}_{=:K_1} + \mu \underbrace{V_{X_s}^\top \Delta_t U_t W_{t,\perp}}_{:=K_2} & (42c)
\end{aligned}
$$

where (42a) follows from $V_{X_s}^\top XX^\top U_t W_{t,\perp} = \Sigma_s V_{X_s}^\top U_t W_{t,\perp} = 0$, and in (42b) and (42c) we use $V_{X_s}^\top U_t W_{t,\perp} = 0$.

For $K_1$, note that

$$
\left\| \left( V_{X_s}^\top U_{t+1} U_{t+1}^\top V_{X_s} \right)^{-\frac{1}{2}} V_{X_s}^\top U_t \right\|
$$

$$
\leq \left\| \left( V_{X_s}^\top U_{t+1} U_{t+1}^\top V_{X_s} \right)^{-\frac{1}{2}} V_{X_s}^\top U_{t+1} \right\| + \mu \left\| \left( V_{X_s}^\top U_{t+1} U_{t+1}^\top V_{X_s} \right)^{-\frac{1}{2}} V_{X_s}^\top \mathcal{A}^* \mathcal{A}(XX^\top - U_t U_t^\top) U_t \right\|
$$

$$
\leq 1 + 10\mu \|X\|^3 \sigma_{\min}^{-1} \left( V_{X_s}^\top U_{t+1} \right)
$$

so that

$$
\left\| \left( V_{X_s}^\top U_{t+1} U_{t+1}^\top V_{X_s} \right)^{-\frac{1}{2}} K_1 \right\| \leq \left[ 1 + 10\mu \|X\|^3 \sigma_{\min}^{-1} \left( V_{X_s}^\top U_{t+1} \right) \right] \left\| V_{X_s,\perp}^\top U_t W_t \right\| \left\| U_t W_{t,\perp} \right\|.
$$

Plugging into (42), we deduce that

$$
\begin{aligned}
&\left\| W_{t,\perp}^\top W_{t+1} \right\| \\
&\leq 3\mu \left( 1 + 10\mu \|X\|^3 \sigma_{\min}^{-1} \left( V_{X_s}^\top U_{t+1} \right) \right) \left\| V_{X_s,\perp}^\top V_{U_t W_t} \right\| \|X\| \left\| U_t W_{t,\perp} \right\| \\
&\quad + \mu \sigma_{\min}^{-1} \left( V_{X_s}^\top U_{t+1} \right) \left\| U_t W_{t,\perp} \right\| \left\| (\mathcal{A}^* \mathcal{A} - I)(XX^\top - U_t U_t^\top) \right\| \\
&\leq 3\mu \left( \left\| U_t W_{t,\perp} \right\| + 10\mu \|X\|^3 \right) \left\| V_{X_s,\perp}^\top V_{U_t W_t} \right\| \|X\| \\
&\quad + \mu \sigma_{\min}^{-1} \left( V_{X_s}^\top U_{t+1} \right) \left\| U_t W_{t,\perp} \right\| \left\| (\mathcal{A}^* \mathcal{A} - I)(XX^\top - U_t U_t^\top) \right\| \\
&\leq 3\mu \left( 10\mu \|X\|^2 + c_4 \right) c_3 \|X\| + \mu \left\| (\mathcal{A}^* \mathcal{A} - I)(XX^\top - U_t U_t^\top) \right\|.
\end{aligned}
$$

where in the last step we use Lemma D.5 and the induction hypothesis which implies that $\sigma_{\min}(U_t W_t) \geq \|U_t W_{t,\perp}\|$. $\qquad\square$

**Lemma D.7** *The matrix $\hat{Z}$ defined in the proof of Lemma C.6 satisfies the following:*

$$
\hat{Z}(\hat{Z}^\top \hat{Z})^{-\frac{1}{2}} = V_{U_t W_t} + B V_{U_t W_t} - \frac{1}{2}(I + B)V_{U_t W_t} V_{U_t W_t}^\top \left( B + B^\top \right) V_{U_t W_t} - D,
$$

*where $\|D\| \leq 30\|B\|^2$.*

**Proof** : By definition of $\hat{Z}$ we have

$$
\begin{aligned}
&\hat{Z}(\hat{Z}^\top \hat{Z})^{-\frac{1}{2}} \\
&= (I + \mu M)(I + P)V_{U_t W_t} \left( V_{U_t W_t}^\top \left( I + P^\top \right) \left( I + \mu M \right)^2 (I + P)V_{U_t W_t} \right)^{-\frac{1}{2}} \\
&= (I + B)V_{U_t W_t} \left[ V_{U_t W_t}^\top \left( I + B^\top + B + B^\top B \right) V_{U_t W_t} \right]^{-\frac{1}{2}} \\
&= (I + B)V_{U_t W_t} \left[ I + \underbrace{V_{U_t W_t}^\top \left( B^\top + B + B^\top B \right) V_{U_t W_t}}_{=:\Delta} \right]^{-\frac{1}{2}}.
\end{aligned}
$$

It follows from (28) and our assumptions on $c_3$ and $c_4$ that

$$\|\boldsymbol{B}\| \le \mu \left\|\boldsymbol{X}\boldsymbol{X}^\top - \boldsymbol{U}_t\boldsymbol{U}_t^\top\right\| + 6\mu\left(c_3 c_4\|\boldsymbol{X}\| + 50\|\boldsymbol{X}\|^2\delta\right)$$
$$\le 10\mu\|\boldsymbol{X}\|^2 + 6\mu c_3\left(c_4 + 1\right)\|\boldsymbol{X}\| < 0.1$$

(note that this step is independent and does not rely on earlier derivations in the proof of Lemma C.6), so by Taylor's formula, we have

$$\left\|(\boldsymbol{I} + \Delta)^{-\frac{1}{2}} - \boldsymbol{I} + \frac{1}{2}\Delta\right\| \le 3\|\Delta\|^2.$$

Hence,

$$\left\|\hat{\boldsymbol{Z}}(\hat{\boldsymbol{Z}}^\top\hat{\boldsymbol{Z}})^{-\frac{1}{2}} - \left(\boldsymbol{V}_{\boldsymbol{U}_t\boldsymbol{W}_t} + \boldsymbol{B}\boldsymbol{V}_{\boldsymbol{U}_t\boldsymbol{W}_t} - \frac{1}{2}(\boldsymbol{I} + \boldsymbol{B})\boldsymbol{V}_{\boldsymbol{U}_t\boldsymbol{W}_t}\boldsymbol{V}_{\boldsymbol{U}_t\boldsymbol{W}_t}^\top\left(\boldsymbol{B} + \boldsymbol{B}^\top\right)\boldsymbol{V}_{\boldsymbol{U}_t\boldsymbol{W}_t}\right)\right\|$$
$$= \left\|(\boldsymbol{I} + \boldsymbol{B})\boldsymbol{V}_{\boldsymbol{U}_t\boldsymbol{W}_t}\left((\boldsymbol{I} + \Delta)^{-\frac{1}{2}} - \boldsymbol{I} + \frac{1}{2}\Delta - \frac{1}{2}\boldsymbol{V}_{\boldsymbol{U}_t\boldsymbol{W}_t}^\top\boldsymbol{B}^\top\boldsymbol{B}\boldsymbol{V}_{\boldsymbol{U}_t\boldsymbol{W}_t}\right)\right\|$$
$$\le (1 + \|\boldsymbol{B}\|)\left(3\|\Delta\|^2 + \frac{1}{2}\|\boldsymbol{B}\|^2\right) < 30\|\boldsymbol{B}\|^2$$

as desired. $\qquad\square$

## E  PROOF OF RESULTS IN SECTION 5

Theorem 1.1 states that GD approximately learns the rank-$s$ constrained minimizer of the matrix sensing loss. However, Theorem C.1 only implies that GD would get into an $\mathcal{O}(\delta)$ neighborhood of $\boldsymbol{X}_s\boldsymbol{X}_s^\top$. As a result, a more fine-grained analysis is needed. Note that it is not even clear whether the rank-$s$ minimizer of is unique. If it is not, then we may naturally ask which minimizer it converges to.

In this section, we study the landscape of under-parameterized matrix sensing problem

$$f_s(\boldsymbol{U}) = \frac{1}{2}\left\|\mathcal{A}(\boldsymbol{U}\boldsymbol{U}^\top - \boldsymbol{X}\boldsymbol{X}^\top)\right\|_2^2, \quad U \in \mathbb{R}^{d\times s}$$

and establish local convergence of gradient descent. Our key result in this section is Lemma E.6, which states a local PL condition for the matrix sensing loss. Most existing results only study the landscape of (1) in the exact- and over-parameterized case. Zhu et al. (2021) studies the landscape of under-parameterized matrix factorization problem, but they only prove a strict saddle property without asymptotic convergence rate of GD.

When the measurement satisfies the RIP condition, we can expect that the landscape of $f$ looks similar to that of the (under-parameterized) matrix factorization loss:

$$F_s(\boldsymbol{U}) = \frac{1}{2}\left\|\boldsymbol{U}\boldsymbol{U}^\top - \boldsymbol{X}\boldsymbol{X}^\top\right\|_F^2, \quad \boldsymbol{U} \in \mathbb{R}^{d\times s}$$

for some $s < r$. Recall that $\boldsymbol{X}\boldsymbol{X}^\top = \sum_{i=1}^{r_*}\sigma_i^2\boldsymbol{v}_i\boldsymbol{v}_i^\top$. The critical points of $F_s(\boldsymbol{U})$ is characterized by the following lemma:

**Lemma E.1** $\boldsymbol{U} \in \mathbb{R}^{d\times s}$ is a critical point of $F_s(\boldsymbol{U})$ if and only if there exists an orthogonal matrix $\boldsymbol{R} \in \mathbb{R}^{s\times s}$, such that all columns of $\boldsymbol{U}\boldsymbol{R}$ are in $\{\sigma_i\boldsymbol{v}_i : 1 \le i \le r_*\}$.

**Proof**: Assume WLOG that $\boldsymbol{X}\boldsymbol{X}^\top = \mathrm{diag}(\sigma_1^2, \sigma_2^2, \cdots, \sigma_r^2, 0, \cdots, 0) =: \boldsymbol{\Sigma}$. Let $\boldsymbol{U}$ be a critical point of $F_s$, then we have that $\left(\boldsymbol{U}\boldsymbol{U}^\top - \boldsymbol{X}\boldsymbol{X}^\top\right)\boldsymbol{U} = 0$. Let $\boldsymbol{W} = \boldsymbol{U}\boldsymbol{U}^\top$, then $(\boldsymbol{\Sigma} - \boldsymbol{W})\boldsymbol{W} = 0$.

Since $\boldsymbol{W}$ is symmetric, so is $\boldsymbol{W}^2$, and we obtain that $\boldsymbol{\Sigma}\boldsymbol{W}$ is also symmetric. It's then easy to see that that if $\boldsymbol{\Sigma} = \mathrm{diag}\left(\lambda_1\boldsymbol{I}_{m_1}, \cdots, \lambda_t\boldsymbol{I}_{m_t}\right)$ with $\lambda_1 > \lambda_2 > \cdots > \lambda_t \ge 0$, then $\boldsymbol{W}$ is also in block-diagonal form: $\boldsymbol{W} = \mathrm{diag}\left(\boldsymbol{W}_1, \boldsymbol{W}_2, \cdots, \boldsymbol{W}_t\right)$ where $\boldsymbol{W}_i \in \mathbb{R}^{m_i\times m_i}$. For each $1 \le i \le t$, we then have the equation $(\lambda_i\boldsymbol{I}_{m_i} - \boldsymbol{W}_i)\boldsymbol{W}_i = 0$. Hence, there exists an orthogonal matrix $\boldsymbol{R}_i$ such that $\boldsymbol{R}_i^\top\boldsymbol{W}_i\boldsymbol{R}_i$ is a diagonal matrix where the diagonal entries are either 0 or $\sqrt{\lambda_i} = \sigma_i$. Let

$R = \mathrm{diag}\,(R_1, R_2, \cdots, R_t)$, then $R^\top W R$ is diagonal and its nonzero diagonal entries form an $s$-subset of the multi-set $\{\sigma_i : 1 \le i \le r_*\}$. The conclusion follows. $\qquad\square$

In the case of $s = 1$, the global minimizers of $F_s$ are $\pm\sigma_1 v_1$, and we can show that $F_s$ is locally strongly convex around these minimizers. Therefore, we can deduce that $f$ is locally strongly-convex as well. Since our main focus is on $s > 1$, we put these details in Appendix G. When $s > 1$, $F_s(U)$ is not locally strongly-convex due to rotational invariance: if $U$ is a global minima, then so is $UR$ for any orthogonal matrix $R \in \mathbb{R}^{s \times s}$. Instead, we will establish a local PL property w.r.t the procrustes distance:

**Definition E.1** *Let $U_1, U_2 \in \mathbb{R}^{d \times s}$, we define*

$$\mathrm{dist}(U_1, U_2) = \inf \left\{ \|U_1 - U_2 R\|_F : R \in \mathbb{R}^{s \times s} \text{ is orthogonal} \right\}.$$

The following characterization of the optimal $R$ in Definition 5.1 is known in the literature (see e.g. (Tu et al., 2016, Section 5.2.1)) but we provide a proof of it for completeness.

**Lemma E.2** *Let $R$ be the orthogonal matrix which minimizes $\|U_1 - U_2 R\|_F$, then $U_1^\top U_2 R$ is positive semi-definite.*

**Proof** : Observe that

$$
\begin{aligned}
\|U_1 - U_2 R\|_F^2 &= \mathrm{tr}\left((U_1^\top - R^\top U_2^\top)(U_1 - U_2 R)\right) \\
&= \|U_1\|_F^2 + \|U_2\|_F^2 - 2\,\mathrm{tr}\left(R^\top U_2^\top U_1\right).
\end{aligned}
$$

Let $U_2^\top U_1 = A\Sigma B^\top$ be its SVD, then

$$\mathrm{tr}\left(R^\top U_2^\top U_1\right) = \mathrm{tr}\left(B^\top R^\top A\Sigma\right) \le \|B^\top R^\top A\|\,\mathrm{tr}\left(\Sigma\right) = \mathrm{tr}\left(\Sigma\right),$$

where the final step is due to orthogonality of $B^\top R^\top A \in \mathbb{R}^{s \times s}$, and equality holds if and only if $B^\top R^\top A = I$ i.e. $R = AB^\top$. Thus $U_1^\top U_2 R = B\Sigma B^\top$ is positive semi-definite. $\qquad\square$

The following lemma states that the minimizer of matrix sensing loss is also near-optimal for the matrix factorization loss. The main idea for proving this result is to utilize

**Lemma E.3** *Let $Z_s^*$ be a best rank-$s$ solution as defined in* (1), *then we have*

$$\left\|Z_s^* - XX^\top\right\|_F^2 \le \left\|X_s X_s^\top - XX^\top\right\|_F^2 + 10\delta\left\|XX^\top\right\|_F^2.$$

**Proof** : Since $U_s^*$ is the global minimizer of $f$, by the RIP property we have

$$
\begin{aligned}
\left\|XX^\top - Z_s^*\right\|_F^2 &\le (1-\delta)^{-1}\left\|\mathcal{A}\left(XX^\top - Z_s^*\right)\right\|_2^2 \\
&\le (1-\delta)^{-1}\left\|\mathcal{A}\left(XX^\top - X_s X_s^\top\right)\right\|_2^2 \\
&\le \frac{1+\delta}{1-\delta}\left\|XX^\top - X_s X_s^\top\right\|_F^2 \\
&\le \left\|XX^\top - X_s X_s^\top\right\|_F^2 + 10\delta\|XX^\top\|_F^2
\end{aligned}
$$

as desired. $\qquad\square$

The next lemma show that $F_s$ satisfies a local PL property:

**Lemma E.4** *Given $U \in \mathbb{R}^{d \times s}$ and let $R$ be an orthogonal matrix such that $\|U - X_s R\|_F = \mathrm{dist}(U, X_s)$. Suppose that $\mathrm{dist}(U, X_s) \le 0.1\|X\|^{-1}\tau$ (where we recall that $\tau = \min_{s \in [r_*]}\left(\sigma_s^2 - \sigma_{s+1}^2\right)$ is the eigengap of $XX^\top$), then we have*

$$\langle \nabla F_s(U), U - X_s R\rangle \ge 0.1\tau\,\mathrm{dist}^2(U, X_s).$$

**Proof** : Assume WLOG that $R = I$, then $U^\top X_s$ is positive semi-definite. Let $H = U - X_s$, then

$$
\begin{aligned}
\nabla F_s(U) &= (UU^\top - XX^\top)U \\
&= \left[(H + X_s)(H + X_s)^\top - XX^\top\right](H + X_s),
\end{aligned}
$$

so that

$$
\begin{aligned}
&\langle \nabla F_s(\boldsymbol{U}), \boldsymbol{U} - \boldsymbol{X}_s \rangle \\
&= \left\langle \left[ (\boldsymbol{H} + \boldsymbol{X}_s)(\boldsymbol{H} + \boldsymbol{X}_s)^\top - \boldsymbol{X}\boldsymbol{X}^\top \right] (\boldsymbol{H} + \boldsymbol{X}_s), \boldsymbol{H} \right\rangle \\
&= \operatorname{tr} \left( \boldsymbol{H}^\top \left[ (\boldsymbol{H} + \boldsymbol{X}_s)(\boldsymbol{H} + \boldsymbol{X}_s)^\top - \boldsymbol{X}\boldsymbol{X}^\top \right] \boldsymbol{H} + \boldsymbol{H}^\top \left( \boldsymbol{H}\boldsymbol{H}^\top + \boldsymbol{H}\boldsymbol{X}_s^\top + \boldsymbol{X}_s\boldsymbol{H}^\top \right) \boldsymbol{X}_s \right) \\
&\geq -\operatorname{tr} \left( \boldsymbol{H}^\top \boldsymbol{X}_{s,\perp} \boldsymbol{X}_{s,\perp}^\top \boldsymbol{H} \right) - 3\|\boldsymbol{X}\|\|\boldsymbol{H}\|_F^3 + \operatorname{tr} \left( \boldsymbol{H}^\top \boldsymbol{H}\boldsymbol{X}_s^\top \boldsymbol{X}_s \right) \quad (43\mathrm{a}) \\
&\geq \left( \sigma_s^2 - \sigma_{s+1}^2 \right) \|\boldsymbol{H}\|_F^2 - 3\|\boldsymbol{X}\|\|\boldsymbol{H}\|_F^3 \geq 0.1\tau\|\boldsymbol{H}\|_F^2 \quad (43\mathrm{b})
\end{aligned}
$$

where in (43a) we use $\operatorname{tr} \left( (\boldsymbol{H}^\top \boldsymbol{X}_s)^2 \right) \geq 0$ (since $\boldsymbol{H}^\top \boldsymbol{X}_s$ is symmetric as noticed in the beginning of the proof), and (43b) is because of

$$
\begin{aligned}
\operatorname{tr} \left( \boldsymbol{H}^\top \boldsymbol{H}\boldsymbol{X}_s^\top \boldsymbol{X}_s \right) &= \operatorname{tr} \left( \boldsymbol{H}^\top \boldsymbol{H} W_{\boldsymbol{X}_s} \boldsymbol{\Sigma}_{\boldsymbol{X}_s}^2 W_{\boldsymbol{X}_s}^\top \right) \\
&= \operatorname{tr} \left( W_{\boldsymbol{X}_s}^\top \boldsymbol{H}^\top \boldsymbol{H} W_{\boldsymbol{X}_s} \boldsymbol{\Sigma}_{\boldsymbol{X}_s}^2 \right) \\
&\geq \sigma_s^2 \operatorname{tr} \left( W_{\boldsymbol{X}_s}^\top \boldsymbol{H}^\top \boldsymbol{H} W_{\boldsymbol{X}_s} \right) = \sigma_s^2 \|\boldsymbol{H}\|_F^2
\end{aligned}
$$

and

$$
\begin{aligned}
\operatorname{tr} \left( \boldsymbol{H}^\top \boldsymbol{X}_{s,\perp} \boldsymbol{X}_{s,\perp}^\top \boldsymbol{H} \right) &= \operatorname{tr} \left( \boldsymbol{H}^\top V_{\boldsymbol{X}_{s,\perp}} \boldsymbol{\Sigma}_{s,\perp} V_{\boldsymbol{X}_{s,\perp}}^\top \boldsymbol{H} \right) \\
&\leq \|\boldsymbol{\Sigma}_{s,\perp}\| \cdot \left\| \boldsymbol{H}^\top V_{\boldsymbol{X}_{s,\perp}} \right\|_F^2 \leq \sigma_{s+1}^2 \|\boldsymbol{H}\|_F^2.
\end{aligned}
$$

$\square$

**Corollary E.1** *Under the conditions of Lemma E.4, we have $\|\nabla F_s(\boldsymbol{U})\|_F \geq 0.1\tau \operatorname{dist}(\boldsymbol{U}, \boldsymbol{X}_s)$.*

The following lemma shows that the rank-$s$ global minima of matrix sensing must lie in an $\mathcal{O}(\delta)$-neighbourhood of the minima of $F_s$.

**Lemma E.5** *Suppose that Assumption 3.1 holds. Let $\boldsymbol{U}_s^*$ be a global minimizer of $f_s$, then we have*

$$
\operatorname{dist}(\boldsymbol{U}_s^*, \boldsymbol{X}_s) \leq 40\delta\kappa\|\boldsymbol{X}\|_F.
$$

*where we recall that $\kappa = \tau^{-1}\|\boldsymbol{X}\|$ is the condition number of $\boldsymbol{X}\boldsymbol{X}^\top$.*

**Proof** : Define

$$
S = \left\{ \boldsymbol{U} \in \mathbb{R}^{d \times s} : \operatorname{dist}(\boldsymbol{U}, \boldsymbol{X}_s) \leq 0.1\kappa^{-1}\|\boldsymbol{X}\| \right\}.
$$

First we can show that $\boldsymbol{U}_s^* \in S$. The main idea is to apply Lemma A.5. Indeed, it's easy to see that

$$
\lim_{\|\boldsymbol{U}\|_F \to +\infty} F_s(\boldsymbol{U}) = +\infty.
$$

Let $\boldsymbol{U} \in \partial S$ i.e. $\operatorname{dist}^2(\boldsymbol{U}, \boldsymbol{X}_s) = 0.1\|\boldsymbol{X}\|^{-1}\tau$. Assume WLOG that $\operatorname{dist}(\boldsymbol{U}, \boldsymbol{X}_s) = \|\boldsymbol{U} - \boldsymbol{X}_s\|_F$, then by Lemma E.4 we have

$$
\begin{aligned}
F_s(\boldsymbol{U}) - F_s(\boldsymbol{X}_s) &= \int_0^1 t \left\langle \nabla F_s(t\boldsymbol{U} + (1-t)\boldsymbol{X}_s), \boldsymbol{U} - \boldsymbol{X}_s \right\rangle \mathrm{d}t \\
&\geq \int_0^1 0.1\tau t^2 \|\boldsymbol{U} - \boldsymbol{X}_s\|_F^2 \, \mathrm{d}t \\
&\geq 10^{-3}\|\boldsymbol{X}\|^{-2}\tau^3.
\end{aligned}
$$

Recall that all the stationary points of $F_s$ are characterized in Lemma E.1, so that for all $\boldsymbol{U} \notin S$ with $\nabla F_s(\boldsymbol{U}) = 0$, we have

$$
F_s(\boldsymbol{U}) - F_s^* \geq 0.5 \left( \sigma_s^4 - \sigma_{s+1}^4 \right) \geq 0.5\tau^2.
$$

On the other hand, we know from Lemma E.3 that

$$
F_s(\boldsymbol{U}_s^*) - F_s^* \leq 5\delta r\|\boldsymbol{X}\|^4 < 0.5\tau^2, \quad (44)
$$

so Lemma A.5 implies that $\boldsymbol{U}_s^* \in S$.

Inside $S$, we can apply the local PL property that we previously derived. Indeed, note that

$$\|\nabla F_s(\boldsymbol{U}_s^*)\|_F = \left\|\left(\boldsymbol{X}\boldsymbol{X}^\top - \boldsymbol{U}_s^*\left(\boldsymbol{U}_s^*\right)^\top\right)\boldsymbol{U}_s^*\right\|_F$$

$$= \left\|(\mathcal{A}^*\mathcal{A} - \boldsymbol{I})\left(\boldsymbol{X}\boldsymbol{X}^\top - \boldsymbol{U}_s^*\left(\boldsymbol{U}_s^*\right)^\top\right)\boldsymbol{U}_s^*\right\|_F$$

$$\leq \delta\left\|\boldsymbol{X}\boldsymbol{X}^\top - \boldsymbol{U}_s^*\left(\boldsymbol{U}_s^*\right)^\top\right\|_F \|\boldsymbol{U}_s^*\|$$

$$\leq 4\delta\|\boldsymbol{X}\| \cdot \left\|\boldsymbol{X}\boldsymbol{X}^\top\right\|_F.$$

Hence we have that

$$\operatorname{dist}(\boldsymbol{U}, \boldsymbol{X}_s) \leq 40\delta\tau^{-1}\|\boldsymbol{X}\|^2\|\boldsymbol{X}\|_F = 40\delta\kappa\|\boldsymbol{X}\|_F.$$

$\square$

**Corollary E.2** *Suppose that [Assumption 3.1](#) holds, then we have* $\sigma_{\min}\left(\left(\boldsymbol{U}_s^*\right)^\top\boldsymbol{U}_s^*\right) \geq \sigma_s^2 - 80\delta\kappa\|\boldsymbol{X}\|\|\boldsymbol{X}\|_F.$

**Proof :** We assume WLOG that $\|\boldsymbol{U}_s^* - \boldsymbol{X}_s\|_F = \operatorname{dist}(\boldsymbol{U}_s^*, \boldsymbol{X}_s)$ i.e. $\boldsymbol{R} = \boldsymbol{I}$ in [Definition 5.1](#). By [Lemma E.5](#), we have that

$$\left\|\left(\boldsymbol{U}_s^*\right)^\top\boldsymbol{U}_s^* - \boldsymbol{X}_s^\top\boldsymbol{X}_s\right\| \leq \left\|\left(\boldsymbol{U}_s^*\right)^\top\boldsymbol{U}_s^* - \boldsymbol{X}_s^\top\boldsymbol{X}_s\right\|$$

$$\leq \max\left\{\|\boldsymbol{U}_s^*\|, \|\boldsymbol{X}_s\|\right\} \cdot \|\boldsymbol{U}_s^* - \boldsymbol{X}_s\|$$

$$\leq 80\delta\kappa\|\boldsymbol{X}\|\|\boldsymbol{X}\|_F.$$

$\square$

**Lemma E.6** *Suppose that [Assumption 3.1](#) holds. Given* $\boldsymbol{U} \in \mathbb{R}^{d\times s}$*, let* $\boldsymbol{U}_s^* \in \mathbb{R}^{d\times s}$ *be a minimizer of* $f_s$*, and* $\boldsymbol{U}_s^*\boldsymbol{R}$ *be the rank-s minimizer which is closest to* $\boldsymbol{U}$ *(*$\boldsymbol{R} \in \mathbb{R}^{s\times s}$ *is orthogonal). When* $\|\boldsymbol{U} - \boldsymbol{U}_s^*\boldsymbol{R}\| \leq 10^{-2}\kappa^{-1}\|\boldsymbol{X}\|$*, we have*

$$\langle\nabla f_s(\boldsymbol{U}), \boldsymbol{U} - \boldsymbol{U}_s^*\boldsymbol{R}\rangle \geq 0.1\tau\operatorname{dist}(\boldsymbol{U}, \boldsymbol{U}_s^*)^2.$$

**Proof :** We assume WLOG that $\boldsymbol{R} = \boldsymbol{I}$, then $\boldsymbol{U}^\top\boldsymbol{U}_s^*$ is positive semi-definite. Let $\boldsymbol{H} = \boldsymbol{U} - \boldsymbol{U}_s^*$, then

$$\nabla f_s(\boldsymbol{U}) = (\mathcal{A}^*\mathcal{A})(\boldsymbol{U}\boldsymbol{U}^\top - \boldsymbol{X}\boldsymbol{X}^\top)\boldsymbol{U}$$

$$= (\mathcal{A}^*\mathcal{A})\left[(\boldsymbol{H} + \boldsymbol{U}_s^*)(\boldsymbol{H} + \boldsymbol{U}_s^*)^\top - \boldsymbol{X}\boldsymbol{X}^\top\right](\boldsymbol{H} + \boldsymbol{U}_s^*)$$

$$= \left[(\mathcal{A}^*\mathcal{A})\left(\boldsymbol{H}\boldsymbol{H}^\top + \boldsymbol{U}_s^*\boldsymbol{H}^\top + \boldsymbol{H}\left(\boldsymbol{U}_s^*\right)^\top\right)\right](\boldsymbol{H} + \boldsymbol{U}_s^*) - \mathcal{A}^*\mathcal{A}\left(\boldsymbol{X}\boldsymbol{X}^\top - \boldsymbol{U}_s^*\left(\boldsymbol{U}_s^*\right)^\top\right)\boldsymbol{H}$$

where we use the first-order optimality condition

$$\mathcal{A}^*\mathcal{A}\left(\boldsymbol{X}\boldsymbol{X}^\top - \boldsymbol{U}_s^*\left(\boldsymbol{U}_s^*\right)^\top\right)\boldsymbol{U}_s^* = 0.$$

Since $\|\boldsymbol{U}_s^*\| \leq 2\|\boldsymbol{X}\|$ by [Lemma E.5](#), we may thus deduce that

$$\left\|\nabla f_s(\boldsymbol{U}) - \left[\left(\boldsymbol{H}\boldsymbol{H}^\top + \boldsymbol{U}_s^*\boldsymbol{H}^\top + \boldsymbol{H}\left(\boldsymbol{U}_s^*\right)^\top\right)(\boldsymbol{H} + \boldsymbol{U}_s^*) - \left(\boldsymbol{X}\boldsymbol{X}^\top - \boldsymbol{U}_s^*\left(\boldsymbol{U}_s^*\right)^\top\right)\boldsymbol{H}\right]\right\|_F$$

$$\leq \left\|(\mathcal{A}^*\mathcal{A} - \boldsymbol{I})\left(\boldsymbol{H}\boldsymbol{H}^\top + \boldsymbol{U}_s^*\boldsymbol{H}^\top + \boldsymbol{H}\left(\boldsymbol{U}_s^*\right)^\top\right)(\boldsymbol{H} + \boldsymbol{U}_s^*)\right\|_F + \left\|(\mathcal{A}^*\mathcal{A} - \boldsymbol{I})\left(\boldsymbol{X}\boldsymbol{X}^\top - \boldsymbol{U}_s^*\left(\boldsymbol{U}_s^*\right)^\top\right)\boldsymbol{H}\right\|$$

$$\leq 50\delta\|\boldsymbol{X}\|^2\|\boldsymbol{H}\|_F$$

Hence

$$\langle\nabla f_s(\boldsymbol{U}), \boldsymbol{U} - \boldsymbol{U}_s^*\rangle$$

$$\geq \left\langle\left(\boldsymbol{H}\boldsymbol{H}^\top + \boldsymbol{U}_s^*\boldsymbol{H}^\top + \boldsymbol{H}\left(\boldsymbol{U}_s^*\right)^\top\right)(\boldsymbol{H} + \boldsymbol{U}_s^*) - \left(\boldsymbol{X}\boldsymbol{X}^\top - \boldsymbol{U}_s^*\left(\boldsymbol{U}_s^*\right)^\top\right)\boldsymbol{H}, \boldsymbol{H}\right\rangle - 50\delta\|\boldsymbol{X}\|^2\|\boldsymbol{H}\|_F^2$$

$$\geq \operatorname{tr}\left(\boldsymbol{H}(\boldsymbol{H} + \boldsymbol{U}_s^*)^\top(\boldsymbol{H} + \boldsymbol{U}_s^*)\boldsymbol{H}^\top + \boldsymbol{H}^\top\boldsymbol{U}_s^*\boldsymbol{H}^\top\boldsymbol{H} + \left(\left(\boldsymbol{U}_s^*\right)^\top\boldsymbol{H}\right)^2\right.$$

$$\left. - \boldsymbol{H}^\top\left(\boldsymbol{X}\boldsymbol{X}^\top - \boldsymbol{U}_s^*\left(\boldsymbol{U}_s^*\right)^\top\right)\boldsymbol{H}\right) - 50\delta\|\boldsymbol{X}\|^2\|\boldsymbol{H}\|_F^2$$

$$\geq \left[\sigma_{\min}\left(\left(\boldsymbol{U}_s^*\right)^\top\boldsymbol{U}_s^*\right) - \left\|\boldsymbol{X}\boldsymbol{X}^\top - \boldsymbol{U}_s^*\left(\boldsymbol{U}_s^*\right)^\top\right\| - 50\delta\|\boldsymbol{X}\|^2 - 3\|\boldsymbol{U}_s^*\|\|\boldsymbol{H}\| - \|\boldsymbol{H}\|^2\right]\|\boldsymbol{H}\|_F^2.$$

By [Corollary E.2](#) we have $\sigma_{\min}\left(\left(U_s^*\right)^\top U_s^*\right) \geq \sigma_s^2 - 80\delta\kappa\|X\|\|X\|_F$ and $\left\|XX^\top - U_s^*\left(U_s^*\right)^\top\right\| \leq \sigma_{s+1}^2 + 80\delta\kappa\|X\|_F^2$, so that

$$\langle \nabla f_s(U), U - U_s^* \rangle \geq \left(\sigma_s^2 - \sigma_{s+1}^2 - 160\delta\kappa\|X\|\|X\|_F - 50\delta\|X\|^2 - 3\|U_s^*\|\|H\| - \|H\|^2\right)\|H\|_F^2.$$

When $\delta \leq 10^{-3} r_*^{-\frac{1}{2}}\kappa^{-2}$ and $\|H\| \leq 10^{-2}\tau\|X\|^{-1}$, the above implies that $\langle \nabla f_s(U), U - U_s^* \rangle \geq 0.5\tau\|H\|_F^2$, as desired. $\qquad\square$

## F    PROOF OF THEOREM 4.1

With the landscape results in hand, we are now ready to characterize the saddle-to-saddle dynamics of GD. We first note the following proposition, with is straightforward from [Lemma C.9](#) and [Theorem C.1](#). In the following we use $U_{\alpha,t}$ to denote the $t$-th iteration of GD when initialized at $U_0 = \alpha U$.

**Proposition F.1** *There exists matrices $\bar{U}_{\alpha,t}$, $-\hat{T}_\alpha^s \leq t \leq 0$ with rank $\leq s$ such that*

$$\max_{\hat{T}_\alpha^s \leq t \leq 0} \left\|\bar{U}_{t,\alpha} - U_{\alpha,\hat{T}_\alpha^s+t}\right\|_F = \mathcal{O}\left(\alpha^{\frac{1}{2\kappa}}\right) \quad (\alpha \to 0)$$

*where $\hat{T}_\alpha^s$ is defined in [Theorem C.1](#) (we omit the dependence on $\alpha$ there) and moreover*

$$\left\|\bar{U}_0\bar{U}_0^\top - Z_s\right\| \leq 100\kappa^2\|X\|^2 r_*\delta.$$

*where $Z_s = U_s^*\left(U_s^*\right)^\top$ is the rank-s minimizer of the matrix sensing loss i.e.*

$$Z_s = \arg\min\left\{\frac{1}{2}\left\|\mathcal{A}(Z - XX^\top)\right\|_2^2 : Z \in \mathbb{R}^{d\times d} \text{ is positive semi-definite and } \text{rank}\left(Z\right) \leq s\right\}. \tag{45}$$

**Remark F.1** *We omit the dependence on $\alpha$ for simplicity of notations, when it is clear from context.*

**Proof** : It follows from [Lemma C.9](#) that $\max_{1\leq t\leq \hat{T}_\alpha^s}\|U_tW_{t,\perp}\| = O\left(\alpha^{\frac{1}{4\kappa}}\right)$. We choose $\bar{U}_t = U_{\hat{T}_\alpha^s+t}W_{\hat{T}_\alpha^s+t}W_{\hat{T}_\alpha^s+t}^\top$, then rank $\left(\bar{U}_t\right) \leq s$ and moreover by [Theorem C.1](#) we have $\left\|X_sX_s^\top - \bar{U}_0\bar{U}_0^\top\right\|_F \leq 2\kappa^2\|X\|^2\sqrt{r_*}\delta$. On the other hand, similar to [Corollary E.2](#) we have that $\left\|Z_s - X_sX_s^\top\right\|_F \leq 80\delta\kappa\|X\|_F^2$. Thus $\left\|\bar{U}_0\bar{U}_0^\top - Z_s\right\| \leq 100\kappa^2\|X\|^2 r_*\delta$ as desired. $\qquad\square$

Let $\hat{U}_{\alpha,t} = U_{\alpha,t}W_{\alpha,t} \in \mathbb{R}^{d\times s}$, then it satisfies $\hat{U}_0\hat{U}_0^\top = \bar{U}_0\bar{U}_0^\top$. The following corollary shows that $\hat{U}_{\alpha,0}$ is close to $U_s^*$ in terms of the procrutes distance.

**Corollary F.1** *We have $\text{dist}(\hat{U}_0, U_s^*) \leq 80\kappa^3 r_*^{\frac{1}{2}}\|X\|\delta$.*

**Proof** : We know from [Lemma E.5](#) that $\text{dist}(U_s^*, X_s) \leq 40\delta\kappa\|X\|_F$, so it remains to bound $\text{dist}(\hat{U}_0, X_s)$.

The proof idea is the same as that of [Lemma E.5](#), so we only provide a proof sketch here. It has been shown in the proof of [Proposition F.1](#) that

$$F_s(\hat{U}_0) := \frac{1}{2}\left\|X_sX_s^\top - \hat{U}_0\hat{U}_0^\top\right\|_F^2 \leq r_*\left\|X_sX_s^\top - \hat{U}_0\hat{U}_0^\top\right\|^2 \leq 4\kappa^4 r_*\|X\|^4\delta^2 \leq 0.5\tau^2.$$

Note that $F_s$ is the matrix factorization loss with $X_sX_s^\top$ as the ground-truth, so the local PL property (cf. [Lemma E.4](#)) still holds here, and by the same reason as [(44)](#), we deduce that $\text{dist}(\hat{U}_0, X_s) \leq 0.1\|X\|^{-1}\tau$ i.e. $\hat{U}_0$ is in the local PL region around $X_s$. Finally, it follows from the PL property that

$$\text{dist}(\hat{U}_0, X_s) \leq 10\tau^{-1}\left\|\nabla F_s(\hat{U}_0)\right\|_F \leq 10\tau^{-1}\|\hat{U}_0\|\left\|X_sX_s^\top - \hat{U}_0\hat{U}_0^\top\right\|_F \leq 40\kappa^3 r_*^{\frac{1}{2}}\|X\|\delta.$$

The conclusion follows. $\qquad\square$

Recall that matrix sensing loss satisfies a local PL property (cf. [Lemma E.6](#)). As a result, when $\delta$ is small, we can show that GD initialized at $\hat{U}_0$ converges linearly to the ground-truth.

**Lemma F.1** *Let $\hat{U}_t$ be the $t$-th iteration of GD initialized at $\hat{U}_0$. Suppose that $\delta \leq 10^{-2} r_*^{-\frac{1}{2}} \kappa^{-3}$ and $\mu \leq 10^{-3} \|X\|^{-2}$, then we have that*

$$\mathrm{dist}^2(\hat{U}_t, U_s^*) \leq (1 - 0.05\tau\mu)^t \mathrm{dist}^2(\hat{U}_0, U_s^*).$$

***Proof*** : We know from Corollary F.1 that

$$\left\| \hat{U}_0 \right\|_F \leq \|X\|_F + 40\kappa^3 r_*^{\frac{1}{2}} \|X\|\delta.$$

We will prove the following result, which immediately implies Lemma F.1: suppose that $\mathrm{dist}(\hat{U}_t, U_s^*) \leq \mathrm{dist}(\hat{U}_0, U_s^*)$, then

$$\mathrm{dist}^2(\hat{U}_{t+1}, U_s^*) \leq (1 - 0.05\tau\mu)\mathrm{dist}^2(\hat{U}_t, U_s^*). \tag{46}$$

Let $R$ be the orthogonal matrix satisfying $\|U_t - U_s^* R\|_F = \mathrm{dist}(U_t, U_s^*)$. Assume WLOG that $R = I$. We first bound the gradient $\nabla f(\hat{U}_t)$ as follows:

$$
\begin{aligned}
\left\| \nabla f(\hat{U}_t) \right\|_F &= \left\| \mathcal{A}^* \mathcal{A} \left( XX^\top - \hat{U}_t \hat{U}_t^\top \right) \hat{U}_t \right\|_F \\
&\leq \left\| \mathcal{A}^* \mathcal{A} \left( XX^\top - \hat{U}_t \hat{U}_t^\top \right) \right\| \left\| \hat{U}_t - U_s^* \right\|_F + \left\| \left( \hat{U}_t \hat{U}_t^\top - U_s^* (U_s^*)^\top \right) U_s^* \right\|_F \\
&\leq 20\|X\|^2 \left\| \hat{U}_t - U_s^* \right\|_F
\end{aligned}
\tag{47}
$$

where we use $\left\| \hat{U}_t \right\| \leq \|X\| + 40\kappa^3 r_*^{\frac{1}{2}} \|X\|\delta \leq 2\|X\|$ and the RIP property. It follows that

$$
\begin{aligned}
\mathrm{dist}^2(\hat{U}_{t+1}, U_s^*) &\leq \left\| \hat{U}_{t+1} - U_s^* R \right\|_F^2 \tag{48a} \\
&= \left\| \hat{U}_t - \mu\nabla f(\hat{U}_t) - U_s^* R \right\|_F^2 \\
&= \left\| \hat{U}_t - U_s^* R \right\|_F^2 - \mu \left\langle \nabla f(\hat{U}_t), \hat{U}_t - U_s^* R \right\rangle + \mu^2 \left\| \nabla f(\hat{U}_t) \right\|_F^2 \\
&\leq \left( 1 - 0.1\tau\mu + 400\|X\|^4\mu^2 \right) \left\| \hat{U}_t - U_s^* R \right\|_F^2 \tag{48b}
\end{aligned}
$$

where (48a) follows from the definition of dist, and (48b) is due to Lemma E.6 and (47). Finally, (46) follows from the condition $\mu \leq 10^{-3}\kappa^{-2}$. $\qquad\square$

We are now ready to complete the proof of Theorem 4.1.

**Theorem F.1 (Restatement of Theorem 4.1)** *Under Assumptions 3.1 and 3.2, consider GD (3) with learning rate $\mu \leq \frac{1}{10^3\|Z^*\|}$ and initialization $U_{\alpha,0} = \alpha\bar{U}$ for solving the matrix sensing problem (1). There exists a universal constant $c > 0$, a constant $C$ (depending on $\hat{r}$ and $\kappa$) and a sequence of time points $T_\alpha^1 < T_\alpha^2 < \cdots < T_\alpha^{\hat{r} \wedge r_*}$ such that for all $1 \leq s \leq \hat{r} \wedge r_*$, the following holds when $\alpha = \mathcal{O}\left( (\rho r_*)^{-c\kappa} \right)$:*

$$\left\| U_{\alpha, T_\alpha^s} U_{\alpha, T_\alpha^s}^\top - Z_s^* \right\|_F \leq C\alpha^{\frac{1}{10\kappa}}. \tag{49}$$

*where we recall that $Z_s^*$ is the best rank-$s$ solution defined in Definition 1.1. Moreover, GD follows an incremental learning procedure: for all $1 \leq s \leq \hat{r} \wedge r_*$, we have $\lim_{\alpha \to 0} \max_{1 \leq t \leq T_\alpha^s} \sigma_{s+1}(U_{\alpha,t}) = 0$, where $\sigma_i(A)$ denotes the $(s+1)$-th largest singular value of a matrix $A$.*

***Proof*** : Recall that $\left\| U_{\hat{T}_\alpha^s} - \bar{U}_0 \right\|_F = o(1)$ ($\alpha \to 0$) where $\hat{T}_\alpha^s$ is defined in Proposition F.1; we omit the dependence on $\alpha$ to simplify notations. We also note that by the update of GD, we have $\bar{U}_t \bar{U}_t^\top = \hat{U}_t \hat{U}_t^\top$ for all $t \geq 0$.

By Lemma F.1, we have that $\mathrm{dist}^2(\hat{U}_t, U_s^*) \leq (1 - 0.05\tau\mu)^t \mathrm{dist}^2(\hat{U}_0, U_s^*)$ and, in particular, $\left\| \hat{U}_t \right\| \leq 2\|X\|$ for all $t$. Thus $\left\| \bar{U}_t \right\| \leq 2\|X\|$ as well. Moreover, recall that $\|U_t\| \leq 3\|X\|$ for all

$t$. It's easy to see that that the matrix sensing loss $f$ is $L$-smooth in $\{U \in \mathbb{R}^{d \times r} : \|U\| \leq 3\|X\|\}$ for some constant $L = \mathcal{O}(\|X\|^2)$, so it follows from Lemma A.6 that

$$\left\|U_{\hat{T}_\alpha^s + t} - \bar{U}_t\right\|_F \leq (1 + \mu L)^t \left\|U_{\hat{T}_\alpha^s} - \bar{U}_0\right\|_F.$$

On the other hand, since $\text{dist}^2(\hat{U}_t, U_s^*) \leq (1 - 0.05\tau\mu)^t \text{dist}^2(\hat{U}_0, U_s^*)$, we can deduce that

$$\begin{aligned}
\left\|U_{\hat{T}_\alpha^s + t} U_{\hat{T}_\alpha^s + t}^\top - Z_s\right\|_F &\leq \left\|U_{\hat{T}_\alpha^s + t} U_{\hat{T}_\alpha^s + t}^\top - \bar{U}_t \bar{U}_t^\top\right\|_F + \left\|\bar{U}_t \bar{U}_t^\top - U_s^* (U_s^*)^\top\right\|_F \\
&= \left\|U_{\hat{T}_\alpha^s + t} U_{\hat{T}_\alpha^s + t}^\top - \bar{U}_t \bar{U}_t^\top\right\|_F + \left\|\hat{U}_t \hat{U}_t^\top - U_s^* (U_s^*)^\top\right\|_F \\
&\leq 3\|X\| \left(\left\|U_{\hat{T}_\alpha^s + t} - \bar{U}_t\right\|_F + \text{dist}(\hat{U}_t, U_s^*)\right) \\
&\leq 3\|X\| \left((1 + \mu L)^t \left\|U_{\hat{T}_\alpha^s} - \bar{U}_0\right\|_F + (1 - 0.05\tau\mu)^{\frac{t}{2}} \text{dist}^2(\hat{U}_0, U_s^*)\right)
\end{aligned}$$

Since when $\alpha \to 0$, $\left\|U_{\hat{T}_\alpha^s} - \bar{U}_0\right\|_F = O(\alpha^{\frac{1}{4\kappa}})$, it's easy to see that there exists a time $t = t_\alpha^s$ so that we have $\max_{-\hat{T}_\alpha^s \leq t \leq t_\alpha^s} \left\|U_{\hat{T}_\alpha^s + t} - \bar{U}_t\right\|_F = \mathcal{O}\left(\alpha^{\frac{c_1}{\kappa^2}}\right)$ and $\left\|U_{\hat{T}_\alpha^s + t} U_{\hat{T}_\alpha^s + t}^\top - Z_s\right\|_F = \mathcal{O}\left(\alpha^{\frac{c_1}{\kappa^2}}\right)$ as well, where $c_1$ is a universal constant. Let $T_\alpha^s = \hat{T}_\alpha^s + t_\alpha^s$, then $\left\|U_{T_\alpha^s} U_{T_\alpha^s}^\top - Z_s\right\|_F = o(1)$ holds. Recall that $\text{rank}(U_t) \leq s$, so that $\max_{0 \leq t \leq T_\alpha^s} \sigma_{s+1}(U_t) = o(1)$. Finally, for all $0 \leq s < \hat{r} \wedge r_*$, we need to show that $T_\alpha^s < T_\alpha^{s+1}$. Indeed, by Corollary E.2 and the Assumption 3.1 we have $\sigma_{s+1}^2(U_{T_\alpha^s}) \geq \sigma_{s+1}(Z_{s+1}) - o(1) \geq 0.5\sigma_{s+1}^2$, so that $T_\alpha^{s+1} > T_\alpha^s$, as desired. $\qquad\square$

# G  THE LANDSCAPE OF MATRIX SENSING WITH RANK-1 PARAMETERIZATION

In this section, we establish a local strong-convexity result Lemma G.2 for rank-1 parameterized matrix sensing. This result is stronger than the PL-condition we established for general ranks, though the latter is sufficient for our analysis.

**Lemma G.1** *Define the full-observation loss with rank-1 parameterization*

$$g_1(u) = \frac{1}{4} \left\|uu^T - XX^T\right\|_F^2.$$

*Then the global minima of $g_1$ are $u^* = \sigma_1 v_1$ and $-u^*$. Moreover, suppose that $g(u) - g(u^*) \leq 0.5\tau_1$ where $\tau_1 = \sigma_1^2 - \sigma_2^2$ is the eigengap, then we must have*

$$\|u - u^*\|^2 \leq 20\tau_1^{-1}(g_1(u) - g_1(u^*)).$$

**Proof** : We can assume WLOG that $XX^T = \text{diag}(\sigma_1^2, \cdots, \sigma_{r_*}^2, 0, \cdots, 0)$. Then

$$g_1(u) = \frac{1}{4}\left(\|u\|_2^4 - 2\sum_{i=1}^s \sigma_i^2 u_i^2 + \|X^T X\|_F^2\right) \tag{50a}$$

$$\geq \frac{1}{4}\left(\|u\|_2^4 - 2\sigma_1^2\|u\|_2^2 + \|X^T X\|_F^2\right) \tag{50b}$$

$$\geq \frac{1}{4}\left(\|X^T X\|_F^2 - \sigma_1^4\right) \tag{50c}$$

where equality holds if and only if $u_2 = \cdots = u_d = 0$ and $\|u\|^2 = \sigma_1^2$ i.e. $u = \pm\sigma_1 e_1$. Moreover, suppose that $g_1(u) - g_1(u^*) \leq 0.5\tau_1$, it follows from (50b) that $\tau_1 \sum_{i=2}^d u_i^2 \leq 2(g_1(u) - g_1(u^*))$ which implies that $\sum_{i=2}^d u_i^2 \leq 2\tau_1^{-1}(g_1(u) - g_1(u^*))$. Also (50c) yields $\left|\|u\|^2 - \sigma_1^2\right| \leq 4\sqrt{g_1(u) - g_1(u^*)}$. Assume WLOG that $u_1 > 0$, then we have

$$\|u - \sigma_1 e_1\|^2 \leq \sigma_1^{-2}\left(u_1^2 - \sigma_1^2\right)^2 + \sum_{i=2}^d u_i^2$$

$$\leq 20\tau_1^{-1}(g_1(u) - g_1(u^*)).$$

$\qquad\square$

**Lemma G.2** *Let*

$$f_1(\boldsymbol{u}) = \frac{1}{4} \left\| \mathcal{A} \left( \boldsymbol{u}\boldsymbol{u}^T - \boldsymbol{X}\boldsymbol{X}^T \right) \right\|_2^2, \quad \boldsymbol{u} \in \mathbb{R}^d.$$

*Suppose that $\delta \leq 10^{-3} \|\boldsymbol{X}\|^{-2} \tau_1$, then there exists constants $a_1$ and $\iota$, such that $f_1$ is locally $\iota$-strongly convex in $\mathcal{B}_1 = \mathcal{B}(\sigma_1 v_1, a_1) \subset \mathbb{R}^d$. Furthermore, there is a unique global minima of $f_1$ inside $\mathcal{B}_1$.*

**_Proof_** : Recall that we defined the full observation loss $g_1(\boldsymbol{u}) = \frac{1}{4} \left\| \boldsymbol{u}\boldsymbol{u}^T - \boldsymbol{X}\boldsymbol{X}^T \right\|_F^2$. Let $h_1 = f_1 - g_1$, then

$$\begin{aligned}
\left\| \nabla^2 h_1(\boldsymbol{u}) \right\| &= \frac{1}{2} \left\| (\mathcal{A}^* \mathcal{A} - \boldsymbol{I}) (\boldsymbol{u}\boldsymbol{u}^T - \boldsymbol{X}\boldsymbol{X}^T) + 2 (\mathcal{A}^* \mathcal{A} - I) \boldsymbol{u}\boldsymbol{u}^T \right\| \\
&\leq \delta \left( 2\|\boldsymbol{u}\|^2 + \|\boldsymbol{X}\|^2 \right).
\end{aligned}$$

When $\|\boldsymbol{u} - \sigma_1 v_1\|^2 \leq 0.1 \min \left\{ \sigma_1^2, \tau_1 \right\}$ (recall $\tau_1 = \sigma_1^2 - \sigma_2^2$),

$$\sigma_{\min} \left( \nabla^2 g_1(\boldsymbol{u}) \right) = \frac{1}{2} \sigma_{\min} \left( \|\boldsymbol{u}\|^2 I + 2\boldsymbol{u}\boldsymbol{u}^T - \boldsymbol{X}\boldsymbol{X}^T \right) \geq 0.4 \tau_1.$$

Hence we have

$$\sigma_{\min} \left( \nabla^2 f_1(\boldsymbol{u}) \right) \geq \left( \nabla^2 g_1(\boldsymbol{u}) \right) - \left\| \nabla^2 h_1(\boldsymbol{u}) \right\| \geq 0.4 \tau_1 - 4\|\boldsymbol{X}\|^2 \delta \geq 0.2 \tau_1,$$

i.e. strong-convexity holds for $a_1^2 = 0.1 \min \left\{ \sigma_1^2, \tau_1 \right\}$ and $\iota = 0.2 \tau_1$.

Let $\boldsymbol{u}^*$ be a global minima of $f_1$, then we must have $\|\boldsymbol{u}^*\| \leq 2\|\boldsymbol{X}\|$ (otherwise $f_1(\boldsymbol{u}) > f_1(0)$). We can thus deduce that

$$\begin{aligned}
g_1(\boldsymbol{u}^*) &\leq f_1(\boldsymbol{u}^*) + \frac{1}{4} \left| \langle \boldsymbol{u}\boldsymbol{u}^T - \boldsymbol{X}\boldsymbol{X}^T, (\mathcal{A}^* \mathcal{A} - I)(\boldsymbol{u}\boldsymbol{u}^T - \boldsymbol{X}\boldsymbol{X}^T) \rangle \right| \\
&\leq f_1(\boldsymbol{u}) + 10\delta \|\boldsymbol{X}\|^2 \leq g_1(\boldsymbol{u}) + 20\delta \|\boldsymbol{X}\|^2.
\end{aligned}$$

It follows from Lemma G.1 and our assumption on $\delta$ that $\min \left\{ \|\boldsymbol{u}^* - \sigma_1 v_1\|^2, \|\boldsymbol{u}^* + \sigma_1 v_1\|^2 \right\} \leq \frac{1}{2} a_1^2$. Moreover, by strong convexity, there exists only one global minima in $\mathcal{B}_1$, which concludes the proof. $\qquad \square$

