# OpenReview forum: "Understanding Incremental Learning of Gradient Descent: A Fine-grained analysis of Matrix Sensing"
_ICLR.cc/2023/Conference — Submitted to ICLR 2023_

### Official Review · Reviewer_7KXn · 2022-10-27

**Confidence:** 3
**Correctness:** 2
**Technical Novelty And Significance:** 2
**Empirical Novelty And Significance:** 2
**Recommendation:** 3

**Clarity, Quality, Novelty And Reproducibility:**

Clarity:

I found the paper quite difficult to read. The paper tries to write informal theorems that are supposed to help understand the main theorem, but it is marginally easier to read, and the notation in section 1 & and the start of Section 3 are undefined. This leads to an unnecessary waste of space. The results are also written in an extremely confusing manner. I do not think it is ready for publication in its current state.

Quality:

The proofs are technical, but they are based on prior insights with a finer analysis. The proofs are hard to read and there are simplifying assumptions in the appendix, such as eq 10.

Novelty:
The stated problem is novel and so are the theorem statements.

**Strength And Weaknesses:**

Weakesses:
- I'm very confused about the results, and I don't see how the authors claim the incremental learning aspect. Li et al 2018 showed that once you start from $U_\alpha$ for a small enough $\alpha$, you will reach the rank-$r^*$ solution quickly (where $r^*$ is the true rank). Theorem 4.1 shows that for each $s \leq r^*$ , there exists a time $T_{\alpha, s}$, such that gradient descent will come close to the rank-$s$ matrix that minimizes the L2-loss on the measurements (it is also not true that the loss will continue to decrease after this hitting time). But there is no guarantee that this hitting time $T_{\alpha, s}$ is an increasing function in $s$ -- it is not necessary that it learns the rank-1 minimizer, then the rank-2 minimizer, and so on until the rank $r^*$ minimizer. Another difficulty is that each of the ranks require an initialization from a sufficiently small $\alpha$, and hence I do not see why finding a rank-1 solution should help with finding the rank-2 solution.

- This is related to the above point, but the proof of Lemma 4.1 makes no sense. If $\tilde{U}_0$ is supposed to have all eigenvalues $=\alpha$, then how does the proof sketch of Lemma 4.1 satisfy this? Also, there is no proof of Lemma 4.1 in the appendix.

- What is the value of $r$ in theorem 4.1 and 4.2 ? I thought it satisfies $ r \geq r^*$, but then in corollary 4.1 the authors say $r \leq r^*$ and use results from section 5.

- I don't understand how you can claim incremental learning for the underparameterized setting. The theorem in section 5 says nothing about local minima or saddle points, and Theorems 4.1 and 4.2 are defined for overparameterized setting.


Miscellaneous:
- In Theorem 1.1, the step size $\mu$ needs to be redefined, else it's meaning is lost as the gradient step is not in a block environment.

- In Assumption 3.2, Proposition, what is $V_{X_s}$ ? This has not been defined

- In Lemma 4.1, what is $W_{\alpha, T_\alpha}$?

**Summary Of The Paper:**

This paper analyzes the dynamics of gradient descent on overparameterized and underparameterized matrix sensing. The authors show that gradient descent learns solutions with increasing rank -- i.e., it finds the minimizer of the rank-1 matrices that best approximate the given measurements, then increases its rank and learns the best rank-2 approximation, and so on until the true rank is achieved.

For the case of underparameterized matrix sensing, the authors show that the L2-loss function satisfies a local PL inequality, which guarantees a unique global minimum when fitting a rank-s matrix to satisfy the given measurements. However, there is no guarantee that gradient descent will not get at some other stationary point.

**Summary Of The Review:**

The stated problem is interesting, but the results are written in an extremely confusing manner. Overparameterized and underparameterized matrix sensing are arbitrarily exchanged between Theorem 4.1, 4.2, and corollary 4.1. The incremental learning aspect is not rigorously shown, theorems 4.1 and 4.2 are for a fixed rank $s$, and not for all $s=1,\cdots, r$.

---

> ### Author Response · Authors · 2022-11-19
> **Response to Reviewer 7KXn**
>
> We would like to thank the reviewer for providing helpful suggestions and raising concerns about our paper. Specifically, the reviewer asks whether our results hold in the under-parameterized setting, and how our main results can imply the incremental learning procedure of GD. The following is our response to the questions of the reviewer.
>
> **Question 1:Can GD learn high-rank minimizer before it learns low-rank minimizer?**
>
> No, we explicitly state in the revision (see Theorem 4.1) that GD learns the ground truth in increasing ranks i.e. $T_{\alpha,s}<T_{\alpha,s+1}$. The proof sketch is provided on Page 6. At a high level, this is because $U_{\alpha,t}$ stays approximately rank $\leq s$ until $T_{\alpha,s}$. As a result, it would not get close to $Z_{s+1}$, whose $(s+1)$-th largest eigenvalue is $\Theta(1)$.
>
> In the original version, we state our results only for a fixed rank $s$ because they hold under weaker conditions: we only require that the $s$-th largest eigenvalue of the ground-truth $Z$ is larger than the $s+1$ eigenvalue. However, when all eigenvalues are different, the above analysis holds for all $s$ and we thus have the whole incremental learning procedure. The complete incremental learning procedure is not reflected in the original version. We thank the reviewer for catching this issue and it has now been fixed.
>
> **Question 2: How are the theorems related to the under-parameterized setting?**
>
> Our main result (see Theorem 4.1) holds for both the over- and the under-parameterized setting. In Proposition 4.1 of the revised version, we explicitly state our result in the under-parameterized setting i.e. GD would converge to the best low-rank solution of the matrix sensing loss.
>
> **Comment: Some notations are unclear.** We thank the reviewer for pointing out some confusing notations, and we have fixed them in the revised version. $W_{\alpha,t}$ the same as $W_t$ and $W_t$ is defined via singular value decomposition in the proof sketch of Lemma 4.1 (on Page 5). In most parts of this paper we omit the dependence on $\alpha$ and simply write $W_{t}$. $V_{X_s}$ is defined in the notation part at the beginning of Sec.3.

---

### Official Review · Reviewer_rmE5 · 2022-10-27

**Confidence:** 2
**Correctness:** 3
**Technical Novelty And Significance:** 3
**Empirical Novelty And Significance:** 2
**Recommendation:** 5

**Clarity, Quality, Novelty And Reproducibility:**

The paper is mostly easy to read but has many typos (see above). Some details of the experiment is missing.




**Strength And Weaknesses:**

Strength
- The paper provides an analysis on behaviour of gradient flow for a non-convex problem of matrix sensing. The analysis also holds for low rank matrix with rank > 1, unlike previous results which was limited to rank = 1.



Weakness

- The analysis limited to the noiseless case without a discussion of the challenges of the extending it to the noiseless case. In the noisy case, is the expected behaviour that $||U_{\alpha, T_\alpha} U_{\alpha, T_\alpha}^\top - Z_s|| \approx 0$ for small $\alpha$? The experiments are limited to noiseless setting as well. Is the recovery behaviour of GD recovering higher rank matrix with iterations present in the noisy case as well?

- Unclear notation
     -  In Assumption 3.2, where is $V^\top_{X_S}$ defined?
     - In equation 3, $\mu$ is missing. Also, $M_t$ is used without referring to where it is defined.

- Typos
     - In corollary 4.1, is $s=r$ ?
     - In abstract, in complete pharse: "we that...."
     - In page 2, rank$(Z) = r_* \ll d $ should be rank$(Z^*) = r_* \ll d $
     - In proposition 3.1, what is an asymmetric matrix? Is it supposed to be symmetric instead?
     - In equation 4, I believe $U_t$ should be $U_T$ instead.
     - In Theorem 1, please specify the minimization variable.
    - Theorem 5.1 is for the case when $\delta = 0$. Is the condition $\delta \leq 0.01...$ in (2) needed?

- Unclear experiment
     - Is figure 1 for the overparametrized case? What is r? Also, what is the behaviour of the relative loss for larger iteration for figure 1(a). Does the relative error for $r^* = 5$ go to zero?
     - In figure 2, was GD run for 1000 iterations or 500 iterations? Figure shows 500 but in section 6.2, T = 1000.

General comment
- In Thm 1.1, does $T_\alpha$ depend on s? If so, is it true that $T_{\alpha, s+1} \geq T_{\alpha, s}$?
- In page 1, the authors state attribute incremental learning as the reason for why overparametrized neural networks do not overfit? Are there other reasons such as early stopping (which the results in the papers seems to suggest to be true in the under parametrized case)? If so, please adjust the phrasing accordingly.


**Summary Of The Paper:**

This paper studies gradient descent (GD) for low-rank, PSD matrix sensing problem. The authors show that, with RIP sensing matrices and small initialization, GD exhibits an incremental learning behaviour, i.e. its trajectory goes through a series of low-rank solutions with increasing rank. In the under-parameterized regime, GD eventually converges to a global minimizer under rank constraint; in the over-parameterized regime, GD (with possibly early stopping) finds the true solution as loss goes to 0. Some numerical experiments are then provided to illustrate this learning behaviour.

**Summary Of The Review:**

Although the results in the paper are interesting, the quality of the paper suffers from the many errors in the paper. The paper also only considers the noiseless case without a discussion of the noisy case (or experiments that show similar behaviour of GD can be expected in the noisy case).

---

> ### Author Response · Authors · 2022-11-19
> **Response to Reviewer rmE5**
>
> We would like to thank the reviewer for providing helpful comments and pointing out possible confusion in our paper. The reviewer points out that there are several typos, undefined notations and unspecified experimental settings. Also, the reviewer asks how the `incremental learning' procedure is reflected in our theorems.  We address the reviewer's concerns in the following.
>
> **Response to “Unclear notations”.** Both $V_{X_s}$ and $M_t$ are defined in the notation section. However, they are put after Assumption 3.2 and equation (3). In the revision, we have moved the notations Section 3.1, and in the subsequent part of this paper, we also point to Section 3.1 when we think it is necessary.
>
> **Response to “Typos”.**
> Yes, in Corollary 4.1 $s$ is actually $r$. We have fixed it in the revised version (now it is Proposition 4.1).
> We thank the reviewer for pointing out this typo in the abstract and on page 2; it is fixed in the revised version.
> The "asymmetric" in Proposition 3.1 should be symmetric instead, and $U_t$ in eq.4 should be $U_T$.
> In Theorem 1 the optimization variable is $Z$. We have specified the optimization variable in the revision.
> In Theorem 5.1, the conclusion does not depend on $\delta$. The condition on $\delta$ is only needed for Theorem 5.2, where we only have partial observations. We have deleted it.
>
> **Reviewer’s Comment 1: The analysis is limited to the noiseless case without a discussion of the challenges of extending it to the noiseless case.**
> Studying the noiseless case is standard for the purpose of understanding the simplicity bias/incremental learning of GD. Previous works such as Arora et al. (2019) [1], Gissin et al. (2019) [2] , Jiang et al. (2022) [3], all study the noiseless case. The noisy case is also interesting to study and we leave it for future work.
>
> **Reviewer’s Comment 2: Experiment Setup is unclear**
>
> We thank the reviewer for pointing out some unclear points regarding the experimental setting.
>
> Fig.1 is for the over-parameterized case where $U \in R^{50\times 50}$. The rank of ground truth is $5$ as stated in the setup. For the over-parameterized setting, the curve for rank $5$ would decrease $o(1)$ as $\alpha\to 0$, but would not go to zero for fixed $\alpha$.
> We change $1000$ to $500$ in the text.
>
> **Reviewer’s Comment 3: Does $T_{\alpha}$ depend on $s$? If so, is it increasing with $s$?**
>
> The time $T_{\alpha}$ depends on $s$. The integer $s$ is a fixed one specified in Assumption 3.2. We notice the potential confusion caused by the way that our theorems are stated. In the revised version, we state our results for all $s$ instead of a specific $s$ (see Theorem 4.1).
> It is true that $T_{\alpha,s}<T_{\alpha,s+1}$, because $U_{\alpha,t}$ stays approximately rank $\leq s$ until $T_{\alpha,s}$. It would not get close to $z_{s+1}$, which is constant distance away from the set of rank $\leq s$ matrices. This is implicit in our proof sketch of Theorem 4.1, since the orthogonal component stays $o(1)$, while the parallel component has rank $\leq s$. We explicitly state this in the revised version (see Theorem 4.1 and its proof sketch on Page 6).
>
>
> **Reviewer’s Comment 4: In Section 1, the authors attribute incremental learning as the reason for why overparametrized neural networks do not overfit, but there could be other reasons**
> We agree that there could be other reasons. We have changed the wording in Section 1 and discussed more on the related works.
>
> [1] Arora, Sanjeev, et al. "Implicit regularization in deep matrix factorization." Advances in Neural Information Processing Systems 32 (2019).
>
> [2] Gissin, Daniel, Shai Shalev-Shwartz, and Amit Daniely. "The implicit bias of depth: How incremental learning drives generalization." arXiv preprint arXiv:1909.12051 (2019).
>
> [3] Jiang, Liwei, Yudong Chen, and Lijun Ding. "Algorithmic Regularization in Model-free Overparametrized Asymmetric Matrix Factorization." arXiv preprint arXiv:2203.02839 (2022).

---

### Official Review · Reviewer_vVy7 · 2022-11-03

**Confidence:** 2
**Correctness:** 4
**Technical Novelty And Significance:** 3
**Empirical Novelty And Significance:** 4
**Recommendation:** 8

**Clarity, Quality, Novelty And Reproducibility:**

The paper is very clear.

The appendix seems quite clearly written but very dense. I have to say that unfortunately being exceptionally behind on my reviewing this year, I didn't have enough time to go through all of the details a much as I wanted to.

I love the paper though so I am hoping I get around to doing that at some point. It would be nice to expand Section B a bit. Indeed, it is quite key to understanding the deeper intuition behind the proof of Theorem 4.1.

There are also a couple of typos which are admittedly minor but make it hard for a reader given that the paper is very technical in the first place. For instance, in the second and third lines of Section B, it says " Specifically, let $V_X^{\top}U_tV_t\Sigma_tW_t^\top$ be the SVD (where we recall that $Z_s$ is defined in Theorem 1.1), then we can write..."
After comparing with the main paper, I finally reached the conclusion that the authors mean "let $V_X^{\top}U_t=V_t\Sigma_tW_t^\top$  be an SVD (i.e. the RHS is the SVD of the LHS), ....", and that this doesn't have much to do with $Z_s$ directly.
There is also an extra ".s" at the end of the introductory part of Section B.





========================Question for the authors==========


Do you think the Gradient Flow analysis would be easier and slightly less technical? Did you give it a try? It seems like the key decomposition in Section B is quite dependent on the fact that we are using a discrete GD procedure, which is a bit surprising to me.


===============================More minor comments==================
I think there is some slight inconsistency between notations such as $V_{X_s^\top}^\top$ and $V_{X_s,\top}^\top$, for instance in the beginning of equation (8).  For instance, I think after "so we can write" on the next page, there are also transposes missing. It is not that easy to get the equation below ", so plugging into (8) gives..." by doing what is said. It would be nice to expand things a bit. Maybe at least add a pointer to the bottom of page 19 in the proof of Lemma C.4. (which expands this calculation)?



On page 14, Lemma A.5, it is not clearly stated whether this lemma is proved somewhere else or if it is from a reference etc. I reached the conclusion that it is assumed obvious enough not to need proof. Indeed, I can see that the first three inequalities follow directly from Weyl's inequality. However, although it seems reasonable, the fourth and last inequality seems like it would deserve a one line explanation so the reader doesn't need to take out a sheet of paper and do calculations for her/himself.


It would always be nice to add a paragraph differentiating the (matrix) RIP property used here from the classic (vector) RIP property with references to both lines of work.

At the top of page 5 in the main, equation (4), are the transposes necessary? It seems they apply to scalars.

In Lemma 4.1 (3) It might be better to write " let $U\in\mathbb{R}^{d\times s} and LET $U_s^*$ be the global minimizer with the minimal..."





=========================minor typos==============

page 16 " however, the aforementioned problem disappear" (disappears)
also page 16 "it thus look promising...." (looks)

**Strength And Weaknesses:**

Strengths:


This is a very interesting topic and a very powerful result. Even if only the extension to the underparametrized case is novel, this is a very significant contribution.

The main paper and some "textual" parts of the appendix are written in a very reader friendly way. A substantial amount of effort is made to try to explain the ideas to the reader.  I love how the authors explain that the dynamics of GD approximate iterative power methods in the early stages, for instance.

The text of the main paper is very well polished.

I think the experiments and conclusions are very believable as well. I actually remember observing similar phenomena in experiments I was running (though I wasn't actually working on the theoretical properties of the optimization strategy).


Weaknesses:



(not serious) The actual mathematical computations are sometimes a bit difficult to follow without extra details.





**Summary Of The Paper:**

This paper studies the convergence of gradient descent on the matrix sensing problem in both the overparametrized and underparametrized regimes, with the latter one being completely new compared to existing studies. The (matrix) RIP property is assumed so that the problem is well-conditioned from the statistical perspective, leaving the problem of optimization to study. The results show that if the initialization is small enough, the learning process grows towards a rank 1 approximation of the solution, and then towards a rank 2 approximation, and so on until it approaches the rank s approximation, where s is the minimum of the ground truth rank and the explicit rank restriction in the optimization problem.

Synthetic data experiments confirm that the training procedure exhibits this behavior in practice.



**Summary Of The Review:**

This seems like a very solid paper with highly impactful results of general interest to the community. I hope I get to read the appendix more deeply in the future.

---

> ### Author Response · Authors · 2022-11-19
> **Response to reviewer vVy7**
>
> We would like to thank the reviewer for appreciating our work and providing helpful suggestions. Below is our response to the reviewer's questions and concerns.
>
> **Notation and typos.** We have followed the reviewer's advice to clarify the notations that may cause confusion. The typos that the reviewer points out are also fixed in the revised version. Given the limited amount of time for rebuttal revision, we have mainly focused on polishing the main text. We will continue to polish the appendix after the rebuttal revision deadline.
>
> **Comment 1: Does the analysis in Appendix B only work for discrete GD? Will analyzing gradient flow be easier and slightly less technical?**
> As GD brings an additional challenge that the magnitude of the step size $\mu$ needs to be controlled, we do believe an analysis for gradient flow would be slightly simpler. But we focus on GD in this paper since GD is more of practical interest.
> With infinitesimal step size, all the equations should be replaced by some ODEs with respect to the time $t$. We believe that the analysis in Appendix B can be carried out in this way, where an important change is to replace the perturbation bounds for SVD with certain bounds for the time derivatives of singular vectors/values.
>
> **Comment 2: It would be nice to expand the calculations in Appendix B:**
> Thanks for the suggestion! We have expended the calculation in Appendix B.
>
> **Comment 3: Lemma A.5 is not obvious but it does not have proof**
> We thank the reviewer for pointing out the missing proof. While  Lemma A.5 can be easily derived from Weyl’s inequality, during revision we find that we only need to use Lemmas A.1-A.4 when proving our results. So we have deleted Lemma A.5 to avoid confusion.
>
> **Comment 4: In equation (4), is the transpose for scalars necessary?**
> Sorry for the confusion. Actually it means to take the $T$-th power, not to take the transpose.

---

### Author Response · Authors · 2022-11-19
**To all reviewers and the AC**

We thank all the reviewers for their constructive suggestions for our paper. It appears that most of the reviewers' concerns are about the presentation and clarity. We have uploaded a revised version and we hope that it is clearer and easier to read.

The following are major changes that we make in the revision:

- We improve our writing by fixing all the typos and clarifying all the notations in Section 3.1 and assumptions in Section 3.2.

- We make modifications to highlight the incremental learning procedure of GD, in response to reviewers **rmE5** and **7KXn**. Although the results in the previous version are all correct, we state them for a specific rank $s$ rather than for every rank. These results rely on the weaker assumption that the $s$-th largest eigenvalue is larger than the $(s+1)$-th largest one. However, as pointed out by Reviewer **7KXn**, this may cause confusion on how incremental learning actually takes place. In the revision, our main result is presented in Theorem 4.1, where the incremental learning procedure is more explicitly characterized.

- We reorganize Section 4. We first state our main result Theorem 4.1 and then sketch its proof using Lemma 4.1 and Proposition 4.1. We hope that this roadmap would make it easier for the readers to understand our proof techniques.

---

### Decision · Program_Chairs · 2023-01-20

**Decision:**

Reject

**Justification For Why Not Higher Score:**

The paper is mostly easy to read but has many typos.
Some details of the experiment are missing.
The proofs are technical and hard to read, but they are based on prior insights with a finer analysis.
The paper tries to write informal theorems that are supposed to help understand the main theorem, but it is marginally easier to read.
The results are written in a confusing manner.

**Justification For Why Not Lower Score:**

An interesting topic and a powerful result.
The main paper is well polished. A substantial amount of effort is made to try to explain the ideas to the reader.
The authors explain that the dynamics of GD approximate iterative power methods in the early stages, which is of interest and seems new.


**Metareview: Summary, Strengths And Weaknesses:**

This paper analyzes the dynamics of gradient descent on over-parameterized and under-parameterized matrix sensing.
The authors show that gradient descent learns solutions with increasing rank, i.e., finds the minimizer of the rank-1 matrices that best approximate the given measurements, then increases its rank and learns the best rank-2 approximation, and so on until the true rank is achieved.
For under-parameterized matrix sensing, the authors show that the L2-loss function satisfies a local Polyak-Łojasiewicz inequality, which guarantees a unique global minimum when fitting a fixed rank matrix to satisfy the given measurements. However, there is no guarantee that gradient descent will not reach other stationary points.